# Multi-dimensional satellite observations of aerosol properties and aerosol types over three major urban clusters in eastern China

Yuqin Liu[1], Tao Lin[1], Juan Hong[4], Yonghong Wang[3], Lamei Shi[2], Yiyi Huang[1,9], Xian Wu[1], Hao Zhou[1], Jiahua Zhang[2], Gerrit de Leeuw[5,6,7,8]

1 Key Lab of Urban Environment and Health, Institute of Urban Environment, Chinese Academy of Sciences, Xiamen 361021, China

2 Key Laboratory of Digital Earth Sciences, The Aerospace Information Research Institute, Chinese Academy of Sciences, Beijing 100094, China

3 Department of Physics, P.O. Box 64, 00014 University of Helsinki, Helsinki, Finland

4 Institute for Environmental and Climate Research, Jinan University, Guangzhou, Guangdong 511443, China

5 Royal Netherlands Meteorological Institute (KNMI), R&D Satellite Observations, 3730AE De Bilt, The Netherlands

6 Aerospace Information Research Institute, Chinese Academy of Sciences (AirCAS), No.20 Datun Road, Chaoyang District, Beijing 100101, China

7 Nanjing University of Information Science & Technology (NUIST), School of Atmospheric Physics, No.219, Ningliu Road, Nanjing, Jiangsu, China

8 China University of Mining and Technology (CUMT), School of Environment Science and Spatial Informatics, Xuzhou, Jiangsu 221116, China

9 Coastal and Ocean Management Institute, Xiamen University, Xiamen 361102, China

*Correspondence to: Tao Lin (tlin@iue.ac.cn), Jiahua Zhang(zhangjh@radi.ac.cn)*

**Abstract.** Using fourteen years (2007-2020) of data from passive (MODIS/Aqua) and active (CALIOP/CALIPSO) satellite measurements over China, we investigate (1) the temporal and spatial variation of aerosol properties over the Beijing-Tianjin-Hebei (BTH) region, the Yangtze River Delta (YRD) and the Pearl River Delta (PRD) and (2) the vertical distribution of aerosol types and extinction coefficients for different aerosol optical depth (AOD) and meteorological conditions. The results show the different spatial patterns and seasonal variations of the AOD over the three regions. Annual time series reveal the occurrence of AOD maxima in 2011 over the YRD and in 2012 over the BTH and PRD; thereafter the AOD decreases steadily. Using the CALIOP vertical feature mask, the relative frequency of occurrence (rFO) of each aerosol type in the atmospheric column are analysed: rFOs of dust and polluted dust decrease from north to south, rFOs of clean ocean, polluted continental, clean continental and elevated smoke aerosol increase from north to south. In the vertical, the peak frequency of occurrence (FO) for each aerosol type depends on region and season and varies with AOD and meteorological conditions. In general, three distinct altitude ranges are observed with the peak FO at the surface (clean continental and clean marine aerosol), at ~1 km (polluted dust and polluted continental aerosol) and at ~3 km (elevated smoke aerosol), whereas dust aerosol may occur over the whole altitude range considered in this study (from the surface up to 8 km). The designation of the aerosol type in different height ranges may to some extend reflect the CALIOP aerosol type classification approach. Air mass trajectories indicate the different source regions for the three study areas and for the three different altitude ranges over each area. In this study nighttime CALIOP profiles are used. The comparison with daytime profiles shows substantial differences in the FO profiles with altitude which suggest effects of boundary layer dynamics and aerosol transport on the vertical distribution of aerosol types, although differences due to day/night CALIOP performance cannot be ruled out.

## 1 Introduction

An aerosol is technically defined as a suspension of fine solid or liquid particles in a gas. In the atmosphere, the air is the gas and in atmospheric research the term aerosol commonly refers to the particulate component only (Seinfeld and Pandis, 1998). In this paper, aerosol is used as a generic term for the particulate component, whereas processes are described for aerosol particles, and a group of aerosol

particles with specific properties is indicated by that property (e.g. "dust aerosol"). Aerosol particles are characterized by their diameter, chemical composition and shape (both are size-dependent) and the number of aerosol particles of each size is described by the particle size distribution. Each of these aerosol properties varies with time and space (Unger et al., 2008; Shindell et al., 2009). The chemical composition of an aerosol particle determines its hygroscopicity and thus the ability to take up or release water vapor in response to changes in relative humidity (RH). In supersaturated conditions, hygroscopic particles may be activated and become cloud condensation nuclei (CCN). At low temperatures aerosol particles can act as ice nuclei (Kanji et al., 2017). The chemical composition of the aerosol particles together with the amount of aerosol water determines the optical properties through the complex refractive index which is important for the scattering and absorption of solar radiation in the atmosphere. The effects of these processes on climate (see below) are determined by the amount and size of the aerosol particles and thus the particle size distribution. For instance, in the presence of large CCN concentrations, the amount of water vapour available is distributed over many cloud droplets which results in smaller sizes and larger cloud albedo (Twomey, 1974) and less precipitation (Rosenfeld et al., 2008). In the presence of high concentrations of aerosol particles, more solar radiation is scattered and absorbed than in the presence of low concentrations, resulting in larger extinction and less radiation reaching the surface (Quan et al., 2014; Li et al., 2017; 2018).

Due to such processes, aerosol particles have an important effect on the Earth's climate, directly by the scattering and absorption of solar radiation and indirectly by modifying cloud properties such as the size and lifetime of cloud droplets, which in turn affect cloud albedo and precipitation (Albrecht, 1989; Twomey, 1974; Andreae et al., 2004; Rosenfeld et al., 2008). Aerosol indirect effects on climate are still poorly understood, much research is done on aerosol-cloud-precipitation interaction (Rosenfeld et al., 2014; Seinfeld et al., 2016; Zhou et al., 2016; Liu et al., 2017; Saponaro et al., 2017; Guo et al., 2018). As indicated above, aerosol direct and indirect effects are strongly influenced by aerosol composition (IPCC, 2013; Rosenfeld et al., 2008; Yang et al., 2016; Massie et al., 2016). In addition, the aerosol vertical distribution is an important factor (Heese et al., 2017; Zhao et al., 2018; Pan et al., 2019) which depends on local sources and vertical mixing together with long-range transport of aerosol generated elsewhere. Also, the aerosol altitude relative to cloud layers needs to be considered (Costantino and Breon, 2013; Wang et al., 2015; Liu et al., 2017; de Graaf et al., 2019). Such information can only be obtained by airborne measurements, or by using remote sensing which provides the data for the current study. However, the data obtained from the optical instruments used for remote sensing, either ground-based or aboard satellites, does not provide sufficient information to fully constrain the aerosol properties. In particular, aerosol composition is poorly constrained and therefore at best aerosol types are retrieved based on the limited number of degrees of freedom. In this study we used aerosol types derived from observations using the Cloud-Aerosol Lidar with Orthogonal Polarization (CALIOP) aboard the Cloud-Aerosol Lidar and Infrared Pathfinder Satellite Observations (CALIPSO) (Kim et al., 2018), see Sect. 2.2.2 for detail.

Because of the strong spatial variability of aerosol properties and their vertical variation, which are hard to determine from local measurements and sparsely distributed networks, satellites are often used to study effects of aerosol on climate. Satellite-based instruments provide the aerosol optical depth (AOD, the column-integrated aerosol extinction coefficient) at the available wavelengths, as the primary parameter.

AOD is often used as a proxy for the aerosol loading and to assess the aerosol effect on radiation, clouds and precipitation (Luo et al., 2014; Tian et al., 2017; Zhao et al., 2018; Liu et al., 2017; 2018). This brief summary of aerosol properties important for climate and air quality studies shows that a systematic analysis of the temporal and spatial variations of aerosol concentrations, aerosol types and their vertical distribution is needed to better understand aerosol effects.

Many previous studies on the aerosol climatology and trends over China were conducted through ground-based remote sensing and satellite observations. Ground-based remote sensing includes the use of sun photometers which are part of networks such as AERONET (Zhang and Li, 2019), SONET (Zhang and Li, 2015; Li et al., 2019; Zhang et al., 2020), CARSNET (Che et al., 2015), hand-held sun photometers in the CARE-China network (Xin et al., 2015) and solar radiation measurements (Xu et al., 2015). Satellite observations are made using, especially, MODIS (Moderate Resolution Imaging Spectroradiometer; Song et al., 2009; Tan et al., 2015; Ma et al., 2016; He et al., 2016) but also by multi-source satellite data (Lin et al., 2010; Guo et al., 2016a; Dong et al., 2017; Zhang et al., 2017; Proestakis et al., 2018; de Leeuw et al., 2018; Sogacheva et al., 2018a,b; 2020). An analysis of global aerosol type as retrieved by MISR was presented by Kahn and Gaitley (2015). Other studies on the variation of aerosol types and vertical distribution of aerosol have been conducted through field campaigns (Schwarz et al., 2010; Kipling et al., 2013; Bauer et al., 2013; Samset et al., 2014; Wang et al., 2014; Kipling et al., 2016) and using ground-based lidars (He et al., 2008; Huang et al., 2008; Cao et al., 2013), although these have limited spatial coverage (e.g., Liu et al., 2012; Matthias et al., 2004). Since the launch of CALIOP/CALIPSO in 2006 (Winker et al., 2009), the seasonal variations of aerosol types and the aerosol vertical distribution could be examined over large spatial scales, complementary to the local point measurements using ground-based lidars. Huang et al. (2013) examined the seasonal variations of aerosol type and extinction profiles using 5-years of CALIOP observations, including aerosol extinction coefficient, aerosol type, and maximum aerosol layer top height. Guo et al. (2016a) investigated the three-dimensional (3D) structure of aerosol using the frequency of occurrence of aerosol derived from CALIOP observations over China. Tian et al. (2017) investigated the regional climatological aerosol vertical distributions and optical properties for eight representative regions over China. Zhao et al. (2018) examined the seasonal variations of aerosol column loading, vertical distribution, and aerosol types through combining datasets from multiple satellite sensors and ground-based observations during the period 2007-2016. Proestakis et al. (2018) used CALIOP data to create a 3-D climatology of dust aerosol over southeast Asia for 9 years (2007-2015), including the seasonality of dust transport pathways and dust layer heights, and dust-AOD trends. These previous studies mainly focused on the analysis of aerosol types and vertical distributions on global and regional scales. Only few studies focused on exploring the vertical distribution of different aerosol types under different aerosol conditions over China, especially over the urban clusters in eastern China. In addition, meteorology and large scale circulation have a strong effect on the vertical distribution of aerosol, which usually further complicates aerosol indirect effects (He et al., 2008; Kipling et al., 2016; Guo et al., 2016b; Li et al., 2017; Kang et al., 2019; Hou et al., 2019; 2020). With the availability of long-term (2007-2020) measurements of aerosol properties, aerosol types and vertical profiles, together with ERA-Interim reanalysis data and GDAS meteorological data, the aerosol properties and vertical profiles over eastern China can be explored.

This study aims to investigate (1) the spatial and temporal variation of the AOD and the vertical distribution of aerosol types using multiple satellite datasets; (2) the vertical distribution of aerosol types and extinction coefficients for different atmospheric aerosol loading; (3) the effect of meteorological conditions on the vertical distribution of aerosol types. The study is conducted over three major urban areas in China, i.e. the BTH (Beijing, Tianjin and Hebei), the YRD (Yangtze River Delta) and the PRD (Pearl River delta), using MODIS and CALIOP data for the years 2007-2020. In this study, nighttime data are used. To put nighttime data in context, the day/night variation of the vertical distribution of aerosol types available from CALIOP is discussed. This paper is organized as follows. Section 2 describes the study area, datasets used and data processing. Section 3 starts with a general description of the temporal and spatial variations of aerosol properties, followed by a description of the vertical distribution of aerosol types and extinction coefficients under different AOD conditions. In the latter we examine the impacts of different meteorological conditions on the vertical distribution of aerosol types. Also the difference between the vertical distributions of aerosol types during day- and night-time is discussed in Section 3. Daily air mass back trajectories are provided for the whole study period and discussed to evaluate the different source regions for the three study areas, at three altitude ranges in which different aerosol types are assigned in the CALIOP approach. Major findings and conclusions are summarized in section 4 and 5.

## 2 Methods

### 2.1 Study area

Due to the rapid progress of urbanization and industrialization, eastern China has become one of the regions with the highest pollution in the world, and a hotspot for exploring direct and indirect effects of aerosol particles. The aerosol concentrations are very high with a variable and complex composition. Both direct aerosol emissions and secondary aerosol formation contribute to high concentrations of black carbon, other carbonaceous aerosol types, sulfate, nitrate and dust aerosol, etc. High concentrations of aerosol particles over eastern China have strong implications on aerosol-cloud-climate interactions (Kourtidis et al., 2015), which are further complicated by the Asian monsoon system (Li et al., 2016). Eastern China is strongly influenced by the summer monsoon with precipitation belts moving from southern China in early April to northern China in July, and back to southern China in August. The monsoon influences aerosol transport and wet deposition (Liu et al., 2011; Luo et al., 2014), while in turn aerosol particles affect the distribution of precipitation and monsoon intensity (Li et al., 2016). Considering the occurrence of different aerosol types, their emission levels and meteorological and climate conditions, three regions were selected to examine the temporal and spatial variation of aerosol properties and their vertical distributions, i.e. the Beijing-Tianjin-Hebei (BTH) area, the Yangtze River Delta (YRD) and the Pearl River Delta (PRD) (see Fig. 1).

The Beijing-Tianjin-Hebei (BTH) area (35.5°N-40.5°N, 113.5°E-120.5°E) has a temperate monsoon climate. AOD is often high due to intensive human activities and can be augmented by the transport of desert dust in the spring. The Yangtze River Delta (YRD) (28°N-33°N, 117°E-122°E) is an area with a subtropical monsoon climate, which is regarded as a major source region of black carbon and sulfate (Wang et al., 2014; Andersson et al., 2015; Cheng et al., 2017). The Pearl River Delta (PRD) (21.5°N-24.5°N,

111.5°E-115.5°E) is an area with a tropical monsoon climate, which is influenced by both high anthropogenic aerosol emissions and a small fraction of marine aerosol (Streets et al., 2003, 2008; Lei et al., 2011; Xu et al., 2015; Heese et al., 2017).

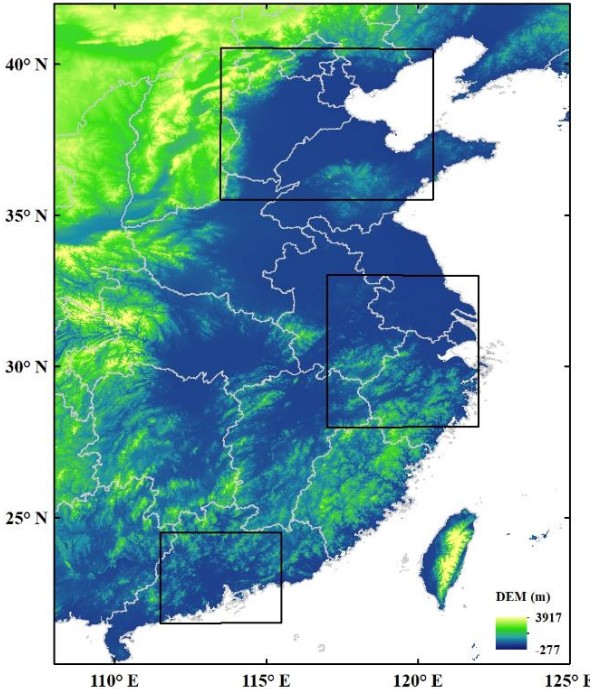

**Figure 1. Elevation map of Eastern China showing the study areas, i.e. Beijing-Tianjin-Hebei (BTH, 35.5°N-40.5°N and 113.5°E-120.5°E), the Yangtze River Delta (YRD, 28°N-33°N and 117°E-122°E) and the Pearl River Delta (PRD, 21.5°N-24.5°N and 111.5°E-115.5°E). These areas are indicated by the black rectangles. The elevation data is downloaded from the website https://search.earthdata.nasa.gov/ (last access: 20 May 2021).**

**2.2 Data sources**

**2.2.1 MODIS/Aqua**

The MODIS sensor was launched by NASA (National Aeronautics and Space Administration, USA) aboard the Aqua satellite in 2002. It observes the Earth using 36 spectral bands from the UV to the thermal infrared and has a swath of 2330 km cross track, providing global coverage in 1-2 days. The Aqua equator

crossing time is approximately 13:30 local time (ascending mode). For cloud-free pixels, three MODIS channels (0.47, 0.66 and 2.12 µm) are used to retrieve the AOD over land (Remer et al., 2005; Levy et al., 2013). Different algorithms are used to retrieve AOD from MODIS data, depending on the surface properties. Two dark target (DT) algorithms are used, one over vegetated/dark-soiled land, and another one over ocean, as described by Levy et al. (2013). Over bright surfaces, the deep blue (DB) algorithm is used

(Hsu et al., 2004; 2013) which was enhanced to return AOD over all land types (Sayer et al., 2013; 2014). The AOD at 550 µm, obtained by interpolating between the AOD at 0.47 and 0.66 µm using the Ångström exponent, is one of the most widely used products in aerosol studies; it is publicly available at https://search.earthdata.nasa.gov/ (last access: 20 May 2021). Updates are regularly provided by the MODIS team at NASA and the most recent version is Collection 6.1 (C6.1) which was issued by the end of

2017. C6.1 is an improved version of C6. MODIS C6 was described in detail by Levy et al. (2013) and the

C6 AOD products over China were validated by, e.g., Tao et al. (2015), Shi et al. (2017) and de Leeuw et al. (2018). C6.1 merged DTDB AOD products over China were initially validated by Sogacheva et al. (2018a). Over China, the differences between the C6 and C6.1 AOD are small, except over certain areas like the Tibetan Plateau, Sichuan Province and the NW of China (Sogacheva et al. 2018a). Over the area considered in the current study the C6 and C6.1 AOD are similar (see Fig. 6 in Sogacheva et al. 2018a). A comprehensive validation of the C6 and C6.1 DT AOD is presented in Che et al. (2019) and Bilal et al. (2019). More detailed information on the aerosol retrieval algorithms is available at http://modis-atmos.gsfc.nasa.gov (last access: 20 May 2021). In this study, we use the MODIS C6.1 level-2 merged product (C6.1 MYD04) (Levy et al., 2013), i.e. the merged DTDB AOD at 550 nm (from here on referred to as AOD) with a spatial resolution of $10\times10$ km$^2$ and better coverage than the individual DT or DB products. In this study, the daily MYD04 AOD data for the period January 2007-December 2020 were averaged to annual, seasonal and monthly values and used to analyse the temporal variations of the AOD on different time scales and the spatial distribution of the AOD in the three study regions.

### 2.2.2 CALIOP/CALIPSO

CALIOP, launched onboard CALIPSO on April 28, 2006 is optimised for aerosol and cloud measurements (Winker et al., 2003). CALIOP is a space-borne near-nadir dual-wavelength lidar (532 nm and 1064 nm) that provides high-resolution vertical profiles of aerosol and clouds. The 532 nm channel is polarisation sensitive. Its footprint is very narrow, with a laser beam diameter of 70 m on the ground. The vertical resolution of the CALIOP product varies with altitude (h): 30 m for h = 0 - 8.2 km, 60 m for h = 8.2 - 20.2 km, and 180 m for h = 20.2 - 30.1 km; the horizontal resolution is 333 m for altitudes from the surface up to 8.2 km, 1 km for altitudes ranging from 8.2 km to 20.2 km, and 1.667 km for altitudes from 20.2 km to 30.1 km (Liu et al., 2009). In the current study the tropospheric column AOD is derived from the Version 4 CALIOP Level 2 5 km aerosol layer products to examine patterns in the spatial distributions of aerosol types during different atmospheric pollution regimes in the three study areas (BTH, YRD and PRD). The Version 4.10 CALIOP level 2 vertical feature mask (VFM) product provides the horizontal and vertical distributions of aerosol layers as well as aerosol types (Kim et al., 2018). The CALIOP sub-type detection scheme uses the input parameters - altitude, location, surface type, corrected depolarization ratio, and integrated attenuated backscatter measurements - to identify the aerosol types. Compared with Version 3.0, the Version 4.10 (V4) CALIOP level 2 contains substantial updates to aerosol subtyping algorithms and the following aerosol types are defined: clean marine (sea salt), clean continental (clean background), polluted continental/smoke (urban/industrial pollution), elevated smoke (biomass burning aerosol), dust (desert), polluted dust (dust mixed with anthropogenic aerosol such as biomass burning smoke or urban pollution) and dusty marine (Kim et al., 2018). A limitation of identifying smoke layers according to altitude is that pollution lofted by convective processes or other vertical transport mechanisms can be misclassified as elevated smoke (Kim et al., 2018). This limitation needs to be kept in mind for the interpretation of the observations and where appropriate, will be mentioned. It is further noted that the CALIOP typing is done on integrated layers that are detected by a separate algorithm, which is not designed to detect differences in aerosol type. Smaller thresholds on depolarization and an attenuation-related depolarization bias can also affect the type classification. What's more, layer heights of contiguous aerosol layers of different

types do not accurately reflect the boundaries between different aerosol types (Burton et al., 2013). Daytime signals can be affected by background sunlight and reduce the SNR (signal to noise ratio), resulting in a larger fraction of undetected aerosol layers during daytime than during nighttime, and underestimation of the CALIOP extinction coefficients and AOD which is larger during daytime than during nighttime (Kim et al., 2017). The larger fraction of undetected aerosol layers during daytime may

also lead to underestimation of the frequency of occurrence (FO) of daytime aerosol types, especially in the upper level (Huang et al., 2013). An overview of the evaluation of the CALIOP AOD versus other measurements shows a low bias of the CALIOP AOD of the order of about 30%. Kim et al. (2018) shows that the CALIOP V4 AOD is still biased low over ocean but less than V3. Over land the V4 vs V3 improvement was not quantified because of the larger uncertainties in the MODIS AOD data which are

used as reference. To avoid day/night differences in the CALIOP data, in this study only nighttime measurements at 532 nm were used to investigate the vertical distribution of aerosol types and extinction coefficients. The vertical distribution of the frequencies of occurrence of CALIOP-derived aerosol types during nighttime are compared with those derived during daytime in Sect. 3.5.

### 2.2.3 ERA-Interim

Meteorological parameters are used to examine the role of meteorological conditions on the vertical distribution of aerosol types. Meteorological parameters are available for the whole world, with different spatial resolutions, every six hours, from the daily ERA Interim Reanalysis (http://apps.ecmwf.int/datasets/data/interim-full-daily/; last access: 20 May 2021). Daily temperatures at the 1000 hPa and 700 hPa levels and pressure vertical velocity (PVV) at the 750 hPa level on 0.125°×0.125°

grids are used with the closest collocation with the CALIOP (nighttime) overpass time (18:00 UTC) over the study area.

### 2.2.4 Air mass trajectories

Backward trajectories of the air masses arriving at the center of the three study areas BTH (38°N, 117°E), YRD (30.5°N, 119.5°E) and PRD (23°N, 113.5°E) were determined using HYSPLIT

(https://www.ready.noaa.gov/HYSPLIT_traj.php; last access: 20 May 2021) and GDAS meteorological data (ftp://arlftp.arlhq.noaa.gov/pub/archives/gdas1/; last access: 20 May 2021). The air mass trajectories were determined for the arrival points at heights of 500 m, 1000 m and 3000 m, i.e. the centers of the height ranges with high frequency of occurrence of the different aerosol types as determined from CALIOP data (Sect. 3.3). The air mass back trajectories were determined over 48 hours, at steps of 6 hours.

 **2.3 Data processing**

In the MODIS Level 2 products, aerosol properties are only retrieved for strictly cloud-free pixels, as determined using a cloud-detection scheme. Through a sensitive cloud detection scheme, the MODIS aerosol algorithm could minimize cloud contamination (Martins et al., 2002). However, because cloud detection schemes are not perfect, some residual clouds may occur resulting in high AOD (Kaufman et al., 2005b). To avoid such problems and use upper AOD limits similar to those of CALIPSO, cases with MODIS AOD greater than 3.0 were discarded in the analysis.

CALIOP version 4 level 2 aerosol products from January 2007 to December 2020 were employed in this study. All CALIOP data include both cases: the aerosol layers in cloudy profiles and in fully cloud-free profiles. CALIOP often cannot detect the full profile of aerosol due to the low instrument sensitivity near the surface, i.e. CALIOP may lose detection capability when the attenuated aerosol backscatter signal is smaller than $2\sim4\times10^{-4}$ $km^{-1}$ $sr^{-1}$ km (Winker et al., 2009; Huang et al., 2013). In particular, the aerosol profile near the surface (below 1.5 km) has high uncertainties which may increase the error in the CALIPSO AOD (Guo et al., 2016a). The uncertainties can be constrained through data screening to some degree. The CALIOP AOD was calculated using only data with quality control flags within the following limits: (1) $0 <= AOD_{532nm} <= 3.0$; (2) $-100 <= CAD\_Score <= -20$; (3) $Ext\_QC = 0, 1$; and (4) $0 < AOD_{532nm,unc}/AOD_{532nm} <= 100\%$, where $AOD_{532nm}$ is the aerosol optical depth at 532 nm wavelength, CAD_Score is the cloud-aerosol discrimination score, Ext_QC is the extinction quality control flag and $AOD_{532nm,unc}$ is the uncertainty of the AOD at a wavelength of 532nm. The aerosol extinction vertical profiles used in this study were selected following similar quality control procedures: (1) $0 <= AOD_{532nm} <= 3.0$; (2) $-100 <= CAD\_Score <= -20$; (3) $Ext\_QC = 0, 1$; and (4) $0 < AOD_{532nm,unc}/AOD_{532nm} <= 100\%$, and (5) extinction coefficients with uncertainty of $99.99$ $km^{-1}$ in the profile are rejected. The CALIOP Feature_Classification_Flags were used to infer aerosol type occurrence at different altitudes in the troposphere. Prior to calculating the aerosol type variation, the aerosol layers with CAD scores between -100 and -20 were selected to ensure that only data of good quality were used. Meteorological conditions can affect the vertical aerosol variation such as transport of the aerosol particles from the lower atmosphere to elevated layers by heavy wind and deep convection (Yumimoto et al., 2009), long-range transport (Guo et al., 2016a) or disconnected layers transported at different heights (Petäjä et al., 2016). Such conditions are explored using meteorological quantities which influence the aerosol properties; here, lower tropospheric stability (LTS) and pressure vertical velocity (PVV) are considered.

**3 Results and discussions**

**3.1 Yearly, seasonally and monthly variation of MODIS AOD**

Time series of annually averaged MODIS AOD over the BTH, YRD and PRD for the years from 2007 to 2020 are presented in Fig. 2(a). Over each of the three regions, the annually averaged AOD varies in a similar way, with the AOD over the PRD about 0.1 lower than over the BTH and the YRD. The AOD

over the YRD was somewhat higher than over the BTH before 2010, whereas after 2011 the AOD over the BTH was highest. With interannual variations during the whole study period, the values in each region did not change much in the beginning of the study period. However, the AOD peaked in 2011 over the YRD, in 2012 over the BTH and PRD. After these years the AOD decreased until the end of the study period, but slower during the last 3-4 years. The AOD decrease indicates that policy measures to control anthropogenic emissions of particulate matter and precursor gases in China are effective (Jin et al., 2016: van der A et al., 2017; Sogacheva et al., 2018b; He et al., 2018; Xie et al., 2019). The annual mean AOD averaged over the whole study period is smallest in the PRD with a value of 0.41±0.09 (annual mean±standard deviation); over the BTH and the YRD the annual mean AODs averaged over the study period have similar values of 0.56±0.07 and 0.55±0.09, respectively (See Supplement Table S1).

The seasonal variation of the MODIS AOD over the three regions is illustrated in Fig. 2(b), which shows the mean AOD in each season, averaged over the years 2007-2020, for each region. Here spring is defined as MAM, summer as JJA, autumn as SON and winter as DJF. The seasonal variation of the AOD is different in each region, and closer inspection shows that the main difference actually occurs in the summer when the 14-year averaged seasonal AOD is lowest in the PRD (0.32) and highest in the BTH (0.71). In spring and autumn, the 14-year averaged AOD is similar in all three regions: around 0.60 in spring and about 0.41 in the autumn. Detailed statistics of the seasonally averaged MODIS AOD are provided in Table S2 of the Supplement. In the winter the AOD is similar to that in the autumn, with somewhat higher value in the BTH and a little lower in the other two regions. This seasonal variation is similar to that observed using ATSR and MODIS-Terra (C6 DTDB) AOD data averaged over 2000-2011 (de Leeuw et al, 2018). During the summer, the direct emission of aerosol particles and precursor gases (contributing to secondary formation of aerosol particles) from straw burning contribute to the high AOD over the BTH. The high relative humidity during the summer monsoon in the BTH results in the growth of aerosol particles and thus a shift of the particle size distribution to larger particles and an increase of the extinction and the AOD. In the spring, the high AOD over the YRD and BTH may be due to the contribution of long-range transported dust, while over the YRD also hygroscopic growth during elevated RH early in the monsoon season may contribute. The high AOD in the PRD in the spring may be related to long-range transport of pollutants from biomass burning in southeast Asia, which then mixes with moist air particles at the top of the boundary layer (Deng et al., 2008; Heese et al., 2017; Zhang et al., 2018). In the autumn the whole eastern region is dominated by westerly winds, which contributes to the diffusion of aerosol particles. Meanwhile, the impact of dust storms is relatively small in the autumn, which is also one of the reasons why the AOD is relatively low in this season. Conversely, during the winter, northerly winds prevail, bringing dry and clean air. In this situation the aerosol tends to be transported to the south and thus the aerosol concentrations over the BTH are reduced (Qi et al., 2013; Si et al., 2018). Hou et al. (2020) discussed four different synoptic situations giving rise to different transport schemes in east China resulting in either the accumulation or dissipation of aerosol in the BTH and YRD regions. The effects of long-range transport, such as that of desert dust in the BTH, the biomass burning over the PRD in the spring and the westerly winds in the autumn and the northerly winds in the winter over the three areas are confirmed by air mass back

trajectories presented and discussed in Sect. 3.6.

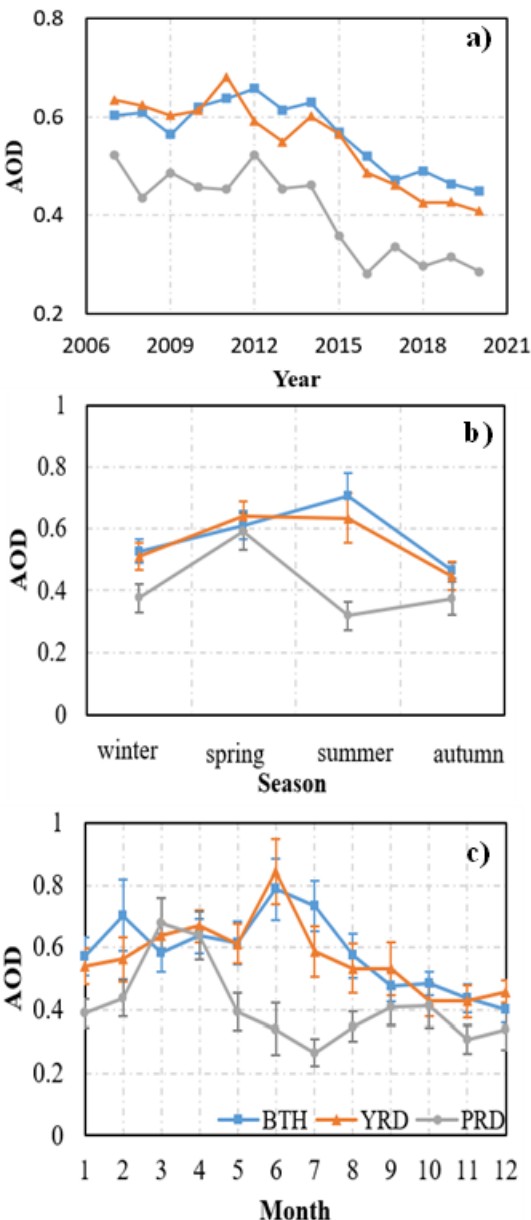

**Figure 2. Annually (a), seasonally (b) and monthly (c) averaged MODIS AOD over the three study regions. The data for the three regions are color-coded.**

The monthly mean MODIS AOD over the three regions, averaged over the 14-years 2007-2020, is presented in Fig. 2(c). Figure 2(c) shows that the largest differences between the regions occur from May to August. The summer AOD peaks in the BTH and in the YRD clearly occur in June (AOD of 0.79 and 0.84, respectively), with a fast decline thereafter. In both regions, the AOD is higher in the period before the summer (0.6-0.7) and declines from September (~ 0.5) to December (~ 0.4). In contrast,

in the PRD the AOD peaks twice, in March (0.68) and in October (0.41), with much lower values in the summer and a clear minimum in July (0.27). The differences between the AOD variations in the three regions are due to processes discussed above for the seasonal variation, while in addition the effect of the East Asian summer monsoon moving from the south of China in April to the North in July and then back to the south affects the month-to-month variations (Luo et al., 2014). The monsoon is accompanied

by heavy rain resulting in the effective removal of aerosol particles by wet deposition and thus the monthly AOD variations over the PRD with one peak before the pre-summer rain in March, the minimum during the summer rain period in July and the second AOD peak after the rainfall in October (Fig. 2(c)). With the seasonal progression of the monsoon to the north and weakening rainfall the monsoon arrives later in the year over the YRD and the BTH where the AOD peaks occur in June. The

monthly mean data for each year and in each region, statistics and the number of overpasses included in the monthly averaged MODIS AOD are provided in Tables S3-S8 of the Supplement.

In summary, the data in Fig. 2 show that AOD differences between the PRD and the other two study areas are largest during the summer months, by a factor of about 2. This difference is the main reason for the lower annual mean AOD over the PRD. During some months (March-April) the AOD is similar in

all three study areas.

### 3.2 Spatial variation of aerosol properties and aerosol types

Maps showing the spatial variation of the annual mean AOD over the three study areas, derived from MODIS and CALIOP data and averaged over the whole study period (2007-2020), are presented in Fig. S1. Statistical information on these data is summarized in Table S9. Figure S1 shows that the spatial

patterns of the MODIS and CALIOP AODs are similar. However, the CALIOP AOD is clearly smaller than that from MODIS, as quantitatively illustrated by the data in Table S9. Underestimation of the AOD by CALIOP has been reported and explained in the literature (cf. Kim et al., 2017, for an overview). It is noted that the comparison in Fig. S1 and Table S9 was made for all available samples and no selection was made based on collocation. Comparison of the maps in Fig. S1 clearly shows the

much smaller number of samples in the CALIOP data, due to the much smaller coverage of CALIOP as a result of the smaller swath width and thus substantially smaller number of CALIOP overpasses (Table S9). Hence, the differences between the MODIS and CALIOP AOD are likely augmented by the highly non-uniform data sample and to the fundamentally different algorithms and operation of the sensors. This was also reported by de Leeuw et al. (2018). In view of these differences, the spatial

variation and time series analysis is made using MODIS data, whereas the vertical information is provided from CALIOP observations.

### 3.2.1 Spatial variation of the MODIS AOD over the three study areas

The spatial distributions of the seasonal mean MODIS AOD over the three urban clusters, averaged over the years from 2007 to 2020, and plotted with a resolution of 0.1°x 0.1°, are shown in Fig. 3. In

Fig. 3, some small areas occur where no data are available; these areas are left white. As mentioned in Sect. 2.3, MODIS data with AOD>3.0 were discarded. It is noted that aerosol retrieval over areas with very high AOD may not be successful due to problems with discrimination between high AOD and clouds. The AOD>3.0 threshold avoids confusing cloudy pixels as high AOD cases. The spatial patterns over each of the three regions are similar in all seasons, but the AOD values vary from season

to season. The AOD over the BTH is low over the mountains in Shanxi province in the Northwest and high in the Southeast over Hebei and Shandong. The mountains separate the North China Plain (NCP)

in the east, with a very high degree of industrialization and a very high population density resulting in very high pollution, from the cleaner areas in the west. The mountains prevent the transport of pollution which accumulates along the ridge in meteorological conditions when the wind is from

south-easterly directions (Sundström et al., 2012), as observed during all seasons. The heavy industries and power plants in the NCP are responsible for the high AOD. Meanwhile, the AOD in the summer may also be enhanced by emissions of aerosol particles and precursor gases from straw-burning (Kang et al., 2016a; Kumar et al., 2015; Si et al., 2018). Over the YRD, the AOD is lower in the Zhejiang and southern Anhui provinces as compared to other areas, during all seasons. The AOD is highest in the

eastern part of the YRD, especially Shanghai and Jiangsu. There is a line with enhanced AOD going from Shanghai to the southwest of Zhejiang, i.e. over the Jin-Qu basin with high population density and much industrial activity. The AOD is lower over the mountains on both sides of the basin. The AOD spatial distribution over the PRD shows a ring-shaped pattern, with the highest values in the center and decreasing toward the outside of the ring. The highest AOD areas cover the busy industrial centers with

much economic activity and high traffic density, leading to elevated anthropogenic pollutants from coal, biomass burning and industrial emissions (Chen et al., 2014; Mai et al., 2018; Zhang et al., 2018).

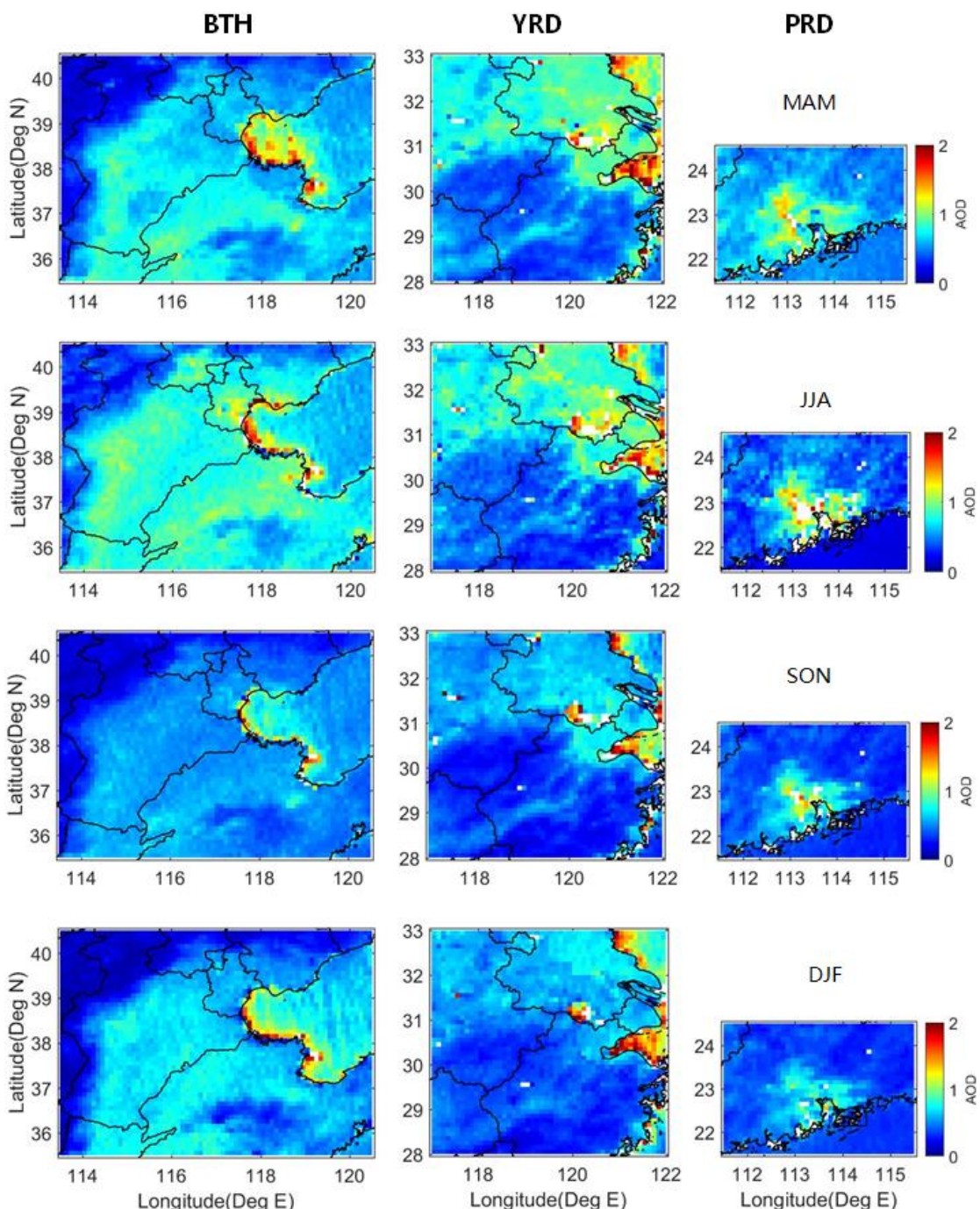

**Figure 3. Spatial distributions of seasonally mean MODIS AOD over the BTH (left column), the YRD (middle column), and the PRD (right column), for spring (MAM), summer (JJA), autumn (SON), and winter (DJF) (top to bottom rows), averaged over the study period from 2007 to 2020.**

### 3.2.2 General distribution of aerosol types over major urban clusters

Aerosol types were obtained from the CALIOP VFM files (nighttime) over the three regions. The relative frequency of occurrence (rFO) of each aerosol type in the atmospheric column over each of the three study areas was calculated by dividing the number of occurrences of each aerosol type in the whole vertical column by the total number of CALIOP aerosol observations. The results, averaged over the years 2007-2020, are presented in Fig. 4. Over the BTH, polluted dust is the most dominant aerosol

type, with an rFO of 45%. The rFO of dust aerosol is 28%. These numbers imply that the deserts in the northwest of China have a very large contribution to the total column-integrated aerosol over the BTH (Guo et al., 2016a, b; de Leeuw et al., 2018). Polluted continental and elevated smoke aerosol both have an rFO of 7%. The rFOs of clean marine and clean continental aerosol over the BTH are very small, with about 2% each.

Similar to the BTH, also over the YRD polluted dust (35%) and dust (22%) have the largest rFOs in the atmospheric column. Polluted continental aerosol also occurs frequently (20%). Elevated smoke aerosol accounts for 15%. Clean marine and clean continental aerosol contribute only little over the YRD, with rFOs of about 2% and 4%, respectively. Dusty marine aerosol has the lowest rFO over the YRD, with 2%.

The aerosol composition over the PRD is substantially different from that over the BTH and YRD, with an rFO of elevated smoke aerosol of 30%. The rFOs of polluted dust and polluted continental aerosol are 17% and 26%, respectively. In contrast to the other two regions, clean marine aerosol has a substantial FO (13%) over the PRD and for dust it is only 3%. The rFO of clean continental aerosol is higher (6%) over the PRD than the other two regions. Although local anthropogenic pollution exerts a major influence on aerosol over the PRD, the northwest winter monsoon may transport continental aerosol (Heese et al., 2017), and the southeast summer monsoon may transport marine aerosol to this region (Wu et al., 2013; Heese et al., 2017).

These data show the large differences between the PRD and the other two regions. The rFOs of clean marine, polluted continental, clean continental and elevated smoke aerosol are lowest over the BTH and highest over the PRD. In contrast, polluted dust and dust have the largest rFOs over the BTH whereas their rFOs over the PRD are small. Transport pathways, i.e. air mass back trajectories, are presented and discussed in Sect. 3.6.

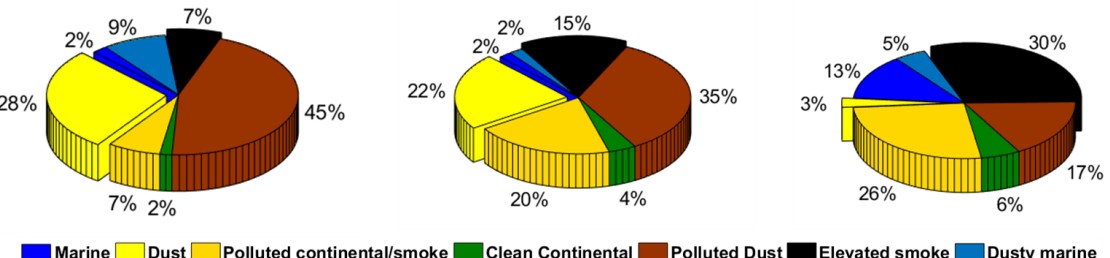

**Figure 4. Relative frequencies of occurrence of different CALIOP aerosol types over the BTH (left), YRD (middle) and PRD (right), averaged over the time period 2007-2020.**

**3.3 Vertical distribution of aerosol types over major urban clusters**

**3.3.1 Vertical profiles of aerosol extinction coefficients during different AOD conditions**

The aerosol extinction coefficient, i.e. the sum of the scattering and absorption by aerosol particles, varies with altitude above the surface due to changes in aerosol properties (see below). Extinction profiles, derived from the vertical variation of the lidar signal (at a wavelength of 532 nm) provide a measure for the vertical variation of aerosol concentrations, weighed by the optical properties of the

aerosol particles. The vertical distribution of the aerosol properties depends on meteorological conditions such as vertical mixing, boundary layer height and the relative humidity profile, as well as the origin of the aerosol (local or long-range transported at elevated levels). In this study, profiles clustered for certain conditions are averaged over the whole 14-year study period resulting in the loss

of detail (such as varying boundary layer heights) and rather smooth profiles. Figure 5 shows nighttime aerosol extinction coefficient profiles averaged over each of the three regions during moderately polluted, polluted and heavily polluted conditions. This distinction was made based on the CALIOP AOD (obtained by integration of the extinction coefficient profiles over the tropospheric column) which was used to divide the profiles into three equally sized subsets. A histogram of

CALIOP AOD values showing the different categories and corresponding number of cases for each region are reported in Fig. S2 and Table S10. The annual mean extinction coefficient profiles were calculated following procedures discussed in Amiridis et al. (2013) and Tackett et al., (2018). The mean of the quality-assured extinction coefficient profiles was first calculated at the overpass level - based on L2 profiles per overpass. Then seasonal and annual profiles were calculated using the mean

profiles for all overpasses. The extinction coefficients decrease monotonically from close to the surface up to about 2 km. The maximum occurring close to the surface is an artefact ascribed to the CALIOP retrieval algorithm which sets the aerosol base at 90 m above the surface to limit contamination due to surface effects on the lidar signal (Koffi et al., 2012). Above 2 km some structure is visible, in particular in heavily polluted conditions over the YRD and PRD. The extinction

coefficient profiles are distinctly different, not only for the different AOD conditions but also over the three regions. As expected, the largest extinction coefficients occur in heavily polluted conditions and the lowest extinction coefficients in moderately polluted conditions, but also the profile shapes are different in each of the three regions. Overall, the extinction coefficients are lowest over the PRD and highest over the YRD.

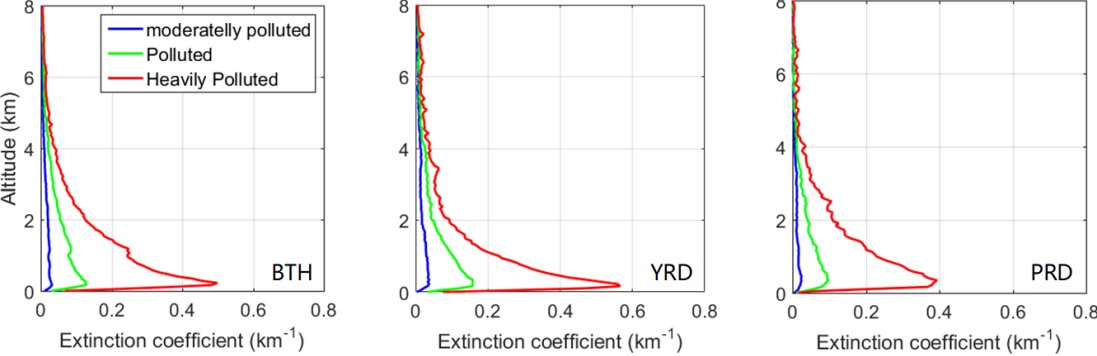


**Figure 5. Aerosol extinction coefficient profiles, averaged over the years 2007-2020, over the BTH (left), YRD (middle) and PRD (right), grouped in different CALIOP AOD ranges for moderately polluted, polluted and heavily polluted conditions (see caption).**

### 3.3.2 Vertical distributions of aerosol types in different seasons

Typically, aerosol type and optical properties vary with altitude. The frequency of occurrence (FO) of the different aerosol types can be calculated through two approaches: one approach is to calculate the

frequency of occurrence of each aerosol type by dividing by the number of CALIPSO measurements (including both clear air and aerosol) in the whole vertical layer; the other approach is to calculate the frequency of occurrence of each aerosol type by dividing the number of CALIPSO measurements (including both clear air and aerosol) within each vertical range. Here, the former definition is designated as All_FO (in %), the latter definition is designated as Layer_FO. It is noted that these profiles show the frequency of occurrence of each aerosol type, normalized to the sum of all aerosol types over the whole profile (All_FO) or each vertical layer (Layer_FO). Hence the FO only indicates a relative number, i.e. ratio of the number of times a certain aerosol type has been assigned by the VFM algorithm to the total number of times that any aerosol type was assigned. The vertical distribution of the All_FO of the different aerosol types during nighttime over the three regions during the spring, summer, autumn and winter, averaged over the years 2007-2020 are presented in Fig. 6. For comparison, similar aerosol type profiles determined using the Layer_FO approach are presented in Fig. S3. Annual mean vertical distributions of the All_FO and Layer_FO of different CALIOP aerosol types are provided in Figs. S4 and S5. The comparison of the aerosol type profiles derived by using the two approaches, shows the noisy character of the profiles resulting from the Layer_FO approach. For some aerosol types the profiles are in good agreement in the lower 2-3 km, for other they are not. At higher altitudes the profiles are often very different, with high FO values from the Layer_FO approach, which makes it hard to compare with values at lower altitude and provides unrealistic vertical distributions. Therefore, in the following we will focus on the vertical distribution of All_FO and, unless specified otherwise, referred to as FO. Profiles determined using the Layer_FO approach are provided in the Supplement, as they provide information on the contributions of different aerosol types as function of height, but are not discussed.

In the FO profiles in Fig.6, three different aerosol layers can be clearly distinguished with one or more dominating aerosol type and smaller FOs from other types. As indicated in Section 3.3.1, in the multi-year averages the concept of planetary boundary layer structure with mixed and residual layers with the free troposphere above (Stull, 1988) cannot be clearly distinguished. Therefore we denote the three aerosol height ranges A, B and C. Range A extends from the surface to about 2 km and does not have a distinct maximum. Range B is interspersed with Range A and extends from the surface, where the FO is very small, to about 3 km, with a distinct maximum at about 1 km. Range C extends from about 1.5 km to 4-5 km with a distinct FO peak around 3 km. In spring (MAM), range C may extend to 6 km. In addition, some aerosol types may occur over the whole column without a distinct layering (e.g. dust aerosol over the BTH and the YRD in the spring (MAM)) which is denoted as range D. It is noted that the definitions used in the CALIOP classification approach (Kim et al., 2018) for elevated smoke, with tops higher than 2.5 km above ground level, and for dusty marine, with tops lower than 2.5 km above ground level (i.e., a simple approximation of a region above the PBL) in CALIOP V4 (Kim et al., 2018) artificially introduces the boundaries of these height ranges. As an example, this stratification is illustrated in Fig. 6 for the profiles over the PRD in the autumn (SON) where range A contains clean marine and dusty marine aerosol (indicated by green block), range B contains polluted continental aerosol (indicated by red triangle) and range C contains elevated smoke and clean continental aerosol

(indicated by yellow circle). Polluted dust aerosol occurs mainly in range B with a small contribution in range C.

The occurrence of aerosol in range D indicates long-range transport of aerosol generated at or injected to higher altitudes and diffusing over the atmospheric boundary layer due to gravitation and other mixing processes. Such vertical mixing obviously affects all aerosol types and the occurrence of different types of aerosol in all three ranges A-C which therefore contain different aerosol types with one or more dominating types. Upward mixing is however limited due to the occurrence of inversion layers which provide an effective lid on the layer below, while on the other hand convective mixing (e.g. in cloud systems) and the formation of disconnected layers may result in the occurrence of aerosol aloft. There are distinct differences in the vertical distributions between the three regions as well as between the seasons in each region. Note that the explanation of the vertical distribution of the aerosol types focuses on physical processes, whereas, as mentioned above, the CALIOP aerosol type classification results from the consideration of statistics on the occurrence of certain aerosol types (Kim et al., 2018). The observed distributions of aerosol types are thus to a certain extend biased by the CALIOP classification approach, yet differences are observed. Below we also discuss long range transport, air mass trajectories showing transport pathways are presented in Sect. 3.6.

The aerosol type over the BTH is dominated by polluted dust with a strong peak in range B, which reflects the contribution of pollution, and a substantial FO at higher altitudes (range D), in particular in spring and summer when a peak is visible at an altitude of around 3 km (range C). Ranges C and D reflect the long-range transported dust component and the mixing with pollution due to upward transport. Dust is the second most important aerosol type over the BTH in all seasons except summer. In spring, dust occurs in all ranges A-D and dominates at higher altitudes, i.e. from 1.8 km to as high as 8 km. In the winter, dust dominates above ~ 4 km. Elevated smoke is important in the summer with a strong peak in range C (above 2 km). During the summer, the direct emission of aerosol paticles and precursor gases (contributing to secondary formation of aerosol particles) from straw burning contributes to the high AOD over the BTH. The larger boundary layer height (BLH) in the summer allows for mixing over a deeper layer, which may promote the larger vertical extent of elevated smoke (see however Sect. 2.2.2). Dusty marine aerosol is observed over the BTH in all seasons, mainly in range A and extending from the surface up to 3-4 km, together with clean marine aerosol which has much smaller FOs and is negligible in spring. A new dusty marine aerosol type is introduced in CALIOP V4, to identify mixtures of dust and marine aerosol. As the BTH is located to the west of the Bohai Sea, marine aerosol occurs most frequently around the coast, and dusty marine aerosol also occurs most frequently when dust settles into the marine boundary layer (MBL) as it approaches the BTH area. Polluted continental aerosol is observed in range B, up to 2-3 km, and is the third most important aerosol type in the summer season.

Over the YRD, the dominating aerosol type varies with the seasons. The most abundant aerosol types in range B are polluted dust, which dominates in the spring and winter, and polluted continental, which dominates in the summer and autumn. Polluted continental aerosol is confined to range B, together with polluted dust which in the winter and spring however extends higher up into range C, to 7 and 8 km respectively (range D), reflecting the influence of long-range transport. Dust is mainly observed in

ranges C and D and dominates in the winter and spring with the peak FO at about 5 km. This indicates that dust aerosol is often transported at high altitudes from north-west China across the mountains during westerly winds (Luo et al., 2014; Guo et al., 2016a; Proestakis et al., 2018). During the summer and autumn the dust FOs are not only much smaller but dust also occurs at lower altitudes.

Elevated smoke aerosol occurs below 4 km (range C) where it dominates in the summer and autumn with a maximum FO at about 3 km. Other aerosol types are observed with much smaller FOs over the YRD, with some clean marine aerosol in range A and clean continental aerosol at somewhat higher elevations.

Over the PRD, elevated smoke (in the winter and spring) and polluted continental (in the summer and

560 autumn) are the dominant aerosol types, while also clean marine aerosol is present in each season in range A, from the surface up to about 2.5 km. Polluted continental aerosol is observed in range B and elevated smoke aerosol peaks at about 3 km (range C), except in the spring when elevated smoke aerosol dominates with relatively constant FOs in an extended layer between 2.5 and 4 km and decreasing toward the surface. Polluted dust occurs in range B and, in the spring, the FO maximum

occurs at about 4 km. Some clean continental aerosol is observed in range C in all seasons. Dust is not prominent and mainly occurs with small FOs in range C.

In general, the peak FO of polluted continental aerosol at an altitude of 1 km shows larger values over the PRD than that over the BTH and YRD. Moreover, polluted continental aerosol in the three target regions is always observed close to the ground (range B) in all seasons (Liu et al., 2020). The peak FOs

of polluted dust aerosol at lower altitude (at an altitude of about 1 km in range B) in the three regions were highest in the winter. This may be explained by enhanced emissions due to the use of fossil fuel for domestic heating in the cold winter season and the mixing of the direct-emitted and secondary aerosol particles with long-range transported and locally generated dust particles. This effect may be enhanced during meteorological conditions conducive of the formation of haze (low windspeed, low

inversion height) which often occur during winter time (Zhang et al., 2008; Tian et al., 2017). The FO of clean marine aerosol (range A) is much higher over the PRD than in other regions in all four seasons, due to the vicinity of the ocean and the southeast summer monsoon which transports oceanic winds over the region.

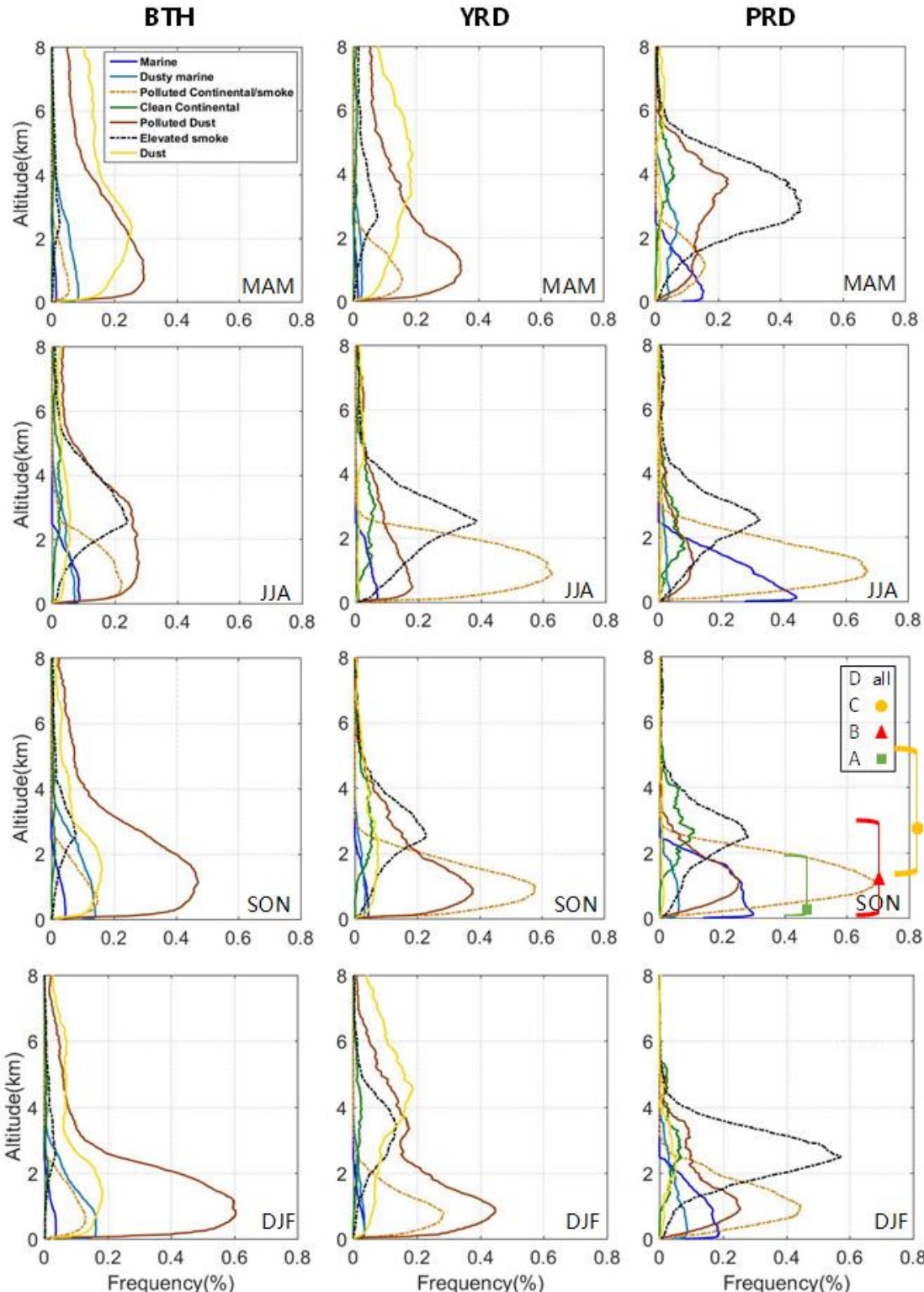

Figure 6. Vertical distribution of the nighttime FO (i.e. All_FO as explained in the text) of different CALIOP aerosol types (see legend) by season, averaged over the years 2007-2020, over the BTH (left), YRD (middle) and PRD (right). The designation of different layers is illustrated for the PRD, autumn, see text.

### 3.3.3 Vertical profiles of aerosol types during different CALIOP AOD conditions

Vertical profiles of night-time aerosol types grouped in different CALIOP AOD ranges with equally sized subsets (as was done for extinction coefficient profiles in Fig. 5) are presented in Figure 7. Over the BTH, dominating aerosol types are polluted dust and dust, but the vertical distributions of the FO of these types do not change much with increasing AOD. For polluted dust, in range B, the depth of the range increases somewhat with increasing AOD, whereas for dust the maximum FO increases with increasing AOD. However, the FOs of polluted continental aerosol, in range A, and elevated smoke aerosol, in range C, increase substantially, whereas clean and dusty marine aerosol, both in range A, are detected much less frequently in heavily polluted conditions than in the other two conditions.

Over the YRD, the dominating aerosol types in polluted and heavily polluted conditions are polluted continental and polluted dust, both in range B with similar FO, but with polluted dust extending well above 2 km in ranges C and D. The contribution of elevated smoke aerosol in range C increases somewhat with increasing pollution level. Dust aerosol also contributes substantially with little variation of the FO in the three conditions and also in the vertical. Clean marine aerosol, in range A, has a substantial FO in moderately polluted conditions. The same applies to clean continental aerosol which however occurs mainly in range C. In polluted and heavily polluted conditions the contributions of both these aerosol types are negligible.

Over the PRD, the aerosol in moderately polluted conditions is dominated by clean marine aerosol in range A reflecting the influence of the ocean south of the PRD. Smaller contributions from elevated smoke, polluted dust and clean continental aerosol (in order of decreasing FO) occur in range C. Some polluted dust is also observed in range B, with somewhat more polluted continental. With increasing pollution, the depth of range A increases but the peak FO of dusty marine aerosol decreases to very small in heavily polluted conditions. The increase of pollution seems to be particularly caused by the strong increase of polluted continental aerosol in range B and, to a lesser extent, elevated smoke aerosol in range C. Also the FO of polluted dust in range B increases while it decreases higher up in range C. Polluted continental, elevated smoke and polluted dust strongly dominate the aerosol in heavily polluted conditions.

The data presented above show some clear differences over the three study regions. Over the BTH the influence of dust clearly dominates the degree of pollution, with both dust and polluted dust near the surface in range B, extending to the top of the study area at 8 km. The effect of dust is smaller but clearly present in the YRD whereas in the PRD it is rather small. The effect of elevated smoke in the free troposphere (range C) is clear in all three regions, increasing with degree of pollution. Elevated smoke effects are strongest over the PRD and smallest over the BTH. Similar considerations apply to polluted continental aerosol: increasing with increasing pollution with the largest effect over the PRD and the smallest effect over the BTH. Marine aerosol (clean or dusty) is also present in all three study regions, but mainly in moderately polluted conditions and decreasing with distance to the ocean, i.e. large over the PRD and small over the BTH.

Generally, the peak FO of elevated smoke aerosol over the three regions is largest in heavily polluted conditions. This can be attributed to the stronger efficiency of smoke aerosol particles for the absorption of sunlight, which increases the AOD (Small et al., 2011; Liu et al., 2017). What's more, the

FO of elevated smoke aerosol in low AOD conditions is distributed higher in the atmosphere over the three regions. A remarkable feature is that the vertical profiles of the dust aerosol FO show little variation in different AOD conditions over the three regions, except for an obviously decreasing height of the peak FO of dust aerosol over the YRD. The vertical distributions of the frequency of occurrences for different aerosol types, derived using the Layer_FO approach, sub-divided into different CALIOP AOD ranges with equally sized subsets, are presented in Fig. S6.

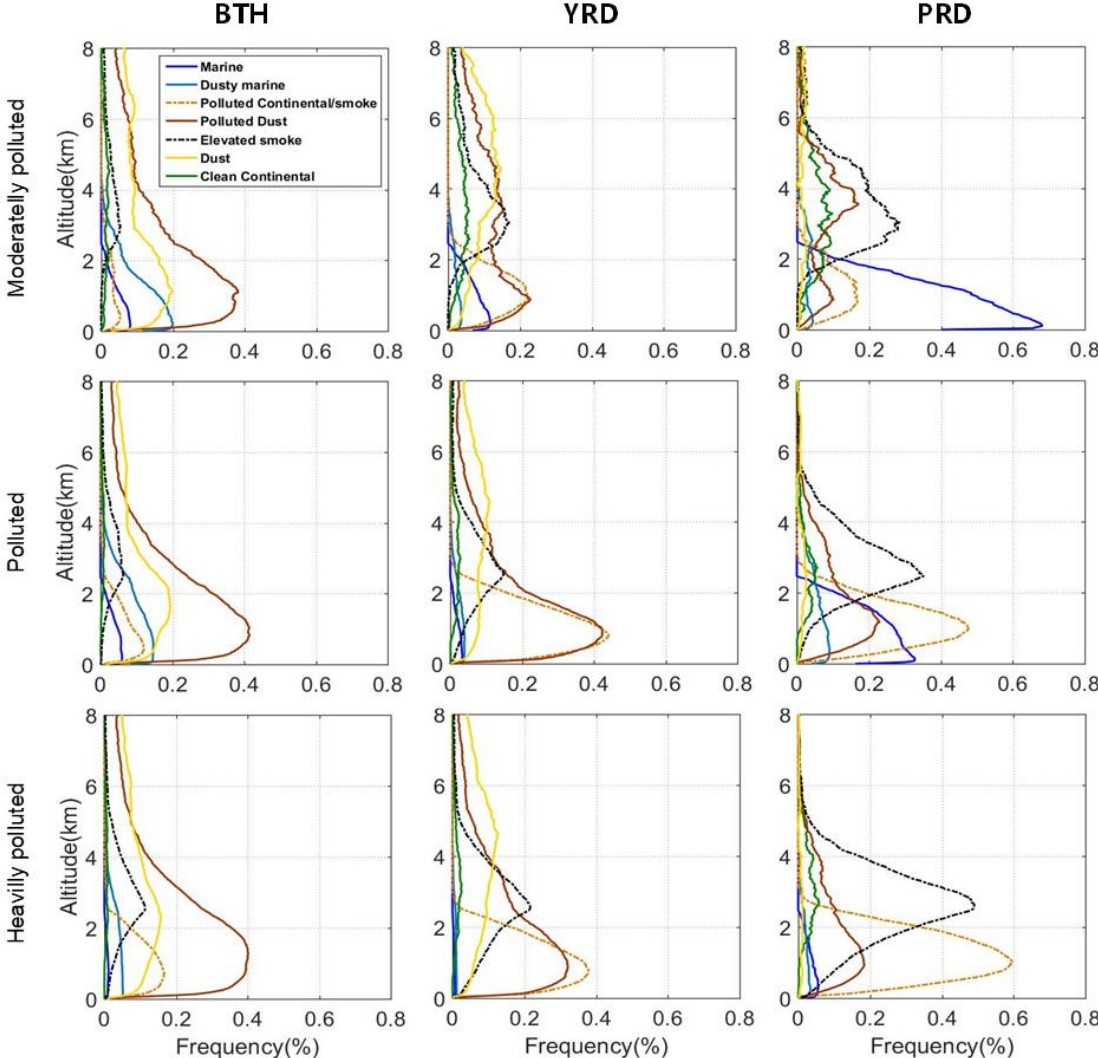

**Figure 7. Vertical distribution of the nighttime FOs of different CALIOP aerosol types (see legend), averaged over the years 2007-2020, grouped in different CALIOP AOD ranges for moderately polluted (top), polluted (middle) and heavily polluted (bottom) conditions (see caption), over the BTH (left), YRD (middle) and PRD (right).**

## 3.4 Vertical distributions of aerosol types during different meteorological conditions

Many studies show the strong effects of local meteorological conditions on the occurrence of air pollution and the formation of haze (Sun et al., 2015; Shen et al., 2020; Lakshmi et al., 2020). However, most previous studies focused on the influence of meteorological conditions on AOD and aerosol concentration (He et al., 2008; Qiu et al., 2011; Tian et al., 2017). In this study, the focus is on aerosol types and their vertical distributions. The observations over the YRD are selected as an example to

illustrate effects of local meteorological conditions (BL thermodynamics) on the aerosol vertical distribution during nighttime.

### 3.4.1 Influence of BL dynamics on the vertical distributions of aerosol types over the YRD

The occurrence of aerosol depends on local sources, i.e. direct production and secondary formation from precursor gases, and processes affecting their transformation and dispersion, as well as long-range

transport from remote sources. Atmospheric circulation and the resulting weather conditions affect the formation and transformation of aerosol particles (Zhang et al., 2008; Cao et al., 2013). The pressure vertical velocity (PVV) is a measure of dynamic convection strength, i.e. vertical mixing. As shown above, aerosol particles mainly occur at heights from near the surface to 4 km, so the PVV at 750 hPa (about 2.4 km) could be used to characterize the atmospheric dynamic conditions. A negative PVV is

indicative of ascending air masses and a positive PVV indicates descending air masses (Jones et al., 2009). The mean vertical distributions of the FO of aerosol types over the YRD, averaged over the years 2007-2020 and stratified by negative and positive PVV are presented in Fig. 8 and Fig. S7 (for Layer_FO). The peak FOs of polluted dust in layer B are larger when PVV>0, indicating that descending motion of air masses is conducive of the deposition and accumulation of aerosol in the

lower atmospheric layers (Tian et al., 2017). Also the occurrence of dust in layer D becomes larger when PVV>0. However, the peak FO of elevated smoke aerosol in range C is smaller in that situation. The FOs for clean marine and clean continental aerosol in ranges A and B are so small that a possible effect of boundary layer dynamics would be too small to observe changes under different PVV conditions.

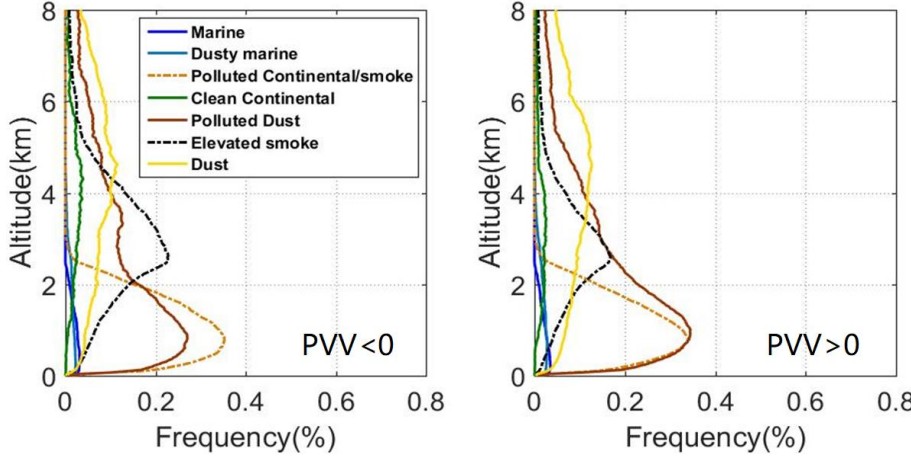

**Figure 8. Vertical distributions of the nighttime FOs of different aerosol types (see legend) over the YRD, averaged over the years 2007-2020. The profiles are stratified by pressure vertical velocity (PVV), as a measure for the strength of vertical mixing (see text), at 750 hPa: i.e. for PVV<0 (left) and for PVV>0 (right).**

### 3.4.2 Influence of BL thermodynamics on the vertical distributions of aerosol types over the YRD

The thermal stability of the atmosphere is closely related to the diffusion and accumulation of aerosol (Kipling et al., 2016). The stability of the lower atmosphere is one of the common atmospheric thermal

conditions, which is used to describe the increase or decrease of the vertical motion of the atmosphere. The lower tropospheric stability (LTS) is calculated from the difference of the potential temperature in the free atmosphere (700 hPa) and near the surface (1000 hPa), indicating a measure of the atmospheric thermodynamic state (Klein and Hartmann, 1993). The larger the LTS, the more stable the atmosphere and the tendency to suppress vertical motion; and vice versa, the smaller the LTS, the more unstable the atmosphere and the tendency to facilitate vertical motion. In this study, all aerosol samples were divided into three equally sized subsets from the lowest to the highest LTS. Mean vertical distributions of the FO of the aerosol subtypes, averaged over the years 2007-2020, for each subset are presented in Fig. 9 and Fig. S8 (for Layer_FO). The peak FOs of polluted continental and polluted dust aerosol in layer B are larger and occur at somewhat lower altitude when the atmosphere becomes more stable. Figure 10 also shows that the peak FO of elevated smoke aerosol occurs around 3 km (i.e. in the free troposphere above the atmospheric boundary layer). Due to the heat released by fossil fuel combustion and biomass burning, the temperature near the ground will rise, and the updraft results in the transport of smoke aerosol into the higher atmosphere, i.e. range C. The occurrence of elevated smoke in layer C is consistent with the definition used in the CALIOP classification approach for elevated smoke, i.e. the layer with tops higher than 2.5 km above ground level (Kim et al., 2018). The data in Fig. 9 show that the FO of elevated smoke aerosol in range C decreases when the atmosphere becomes more stable. In contrast, the peak FOs of polluted dust and dust aerosol above 2 km gradually increases with the increase of LTS.

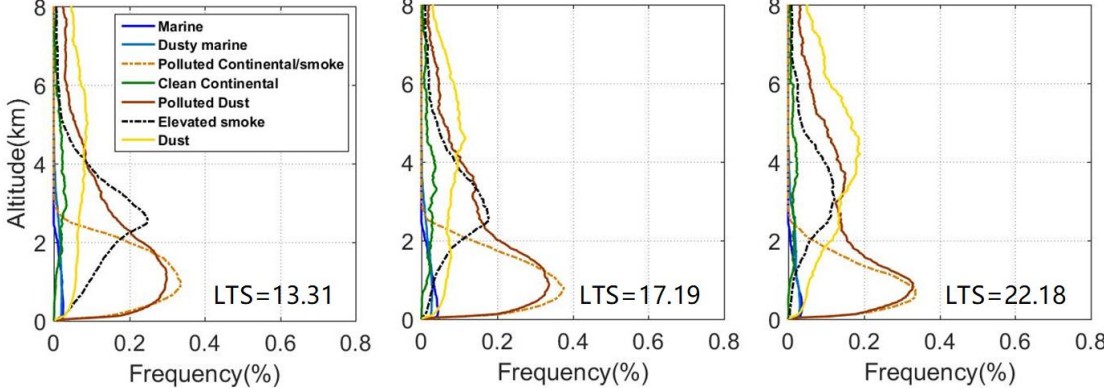

**Figure 9. Vertical distributions of the nighttime FOs of different aerosol types (see legend) for data stratified by unstable atmosphere (left), neutral stable atmosphere (middle) and stable atmosphere (right) over the YRD, averaged over the years 2007-2020.**

**3.5 Day/night variation of the vertical distributions of aerosol types**

The two CALIPSO overpasses, at 1:30 am and pm local time, provide information on the day/night differences of the vertical distribution of the FO of the aerosol types derived from the CALIOP observations. These differences will be discussed based on difference plots (night - day) of FO vertical distributions. However, as discussed in Sect. 2.2.2, the occurrence of undetected aerosol layers results in underestimation of the CALIOP extinction coefficients and AOD and the fraction of undetected aerosol layers is larger during daytime than during nighttime. Furthermore, aerosol in the boundary layer is relatively well detected as compared with aerosol near the top of the boundary layer

(Kim et al., 2017) which may lead to distortion of the vertical profile. Hence day/night differences
may occur between the vertical distributions of the aerosol type FO due to the CALIOP processing
which affect the interpretation of day/night differences in the profiles.

Day/night differences in the vertical structure of aerosol properties are expected due to natural
processes such as direct production or formation of secondary aerosol, transformation of aerosol
particles in the atmospheric boundary layer, vertical mixing and transport from remote locations, wet
and dry deposition. Below the observed day/night differences are briefly discussed based on
consideration of such processes. Separation of these effects from those due to the CALIOP processing,
and determination of their relative importance, are beyond the scope of the current study. This would
require a study on the effects of biases in the retrievals, detection thresholds, noise and the influence
of quality control flags.

Figures 10 and S9 (for Layer_FO) show the vertical distributions of the FO of the aerosol types
during the daytime overpasses over each of the three regions, for each season averaged over the years
2007-2020. Comparison of the daytime vertical distributions in Fig. 10 with the nighttime vertical
distributions in Fig. 6 shows substantial differences which depend on altitude, aerosol type, season
and region. To clearly illustrate these differences, difference plots (night – day) are presented in Fig.
11 and Fig. S10 (for Layer_FO). Fig. 11 shows, for instance, that in the summer, in all three regions,
the maximum FO of the polluted dust layers is larger during the day than during the night (negative
night-day difference). The difference is much larger over the PRD than over the YRD which in turn is
larger than over the BTH. In contrast, for elevated smoke the maximum FO during the day over the
PRD and YRD is substantially smaller than during the night (positive night-day difference), while
over the BTH the day/night difference is rather small. For clean marine aerosol the FO is larger
during the night in the PRD and negligible in the other two regions.

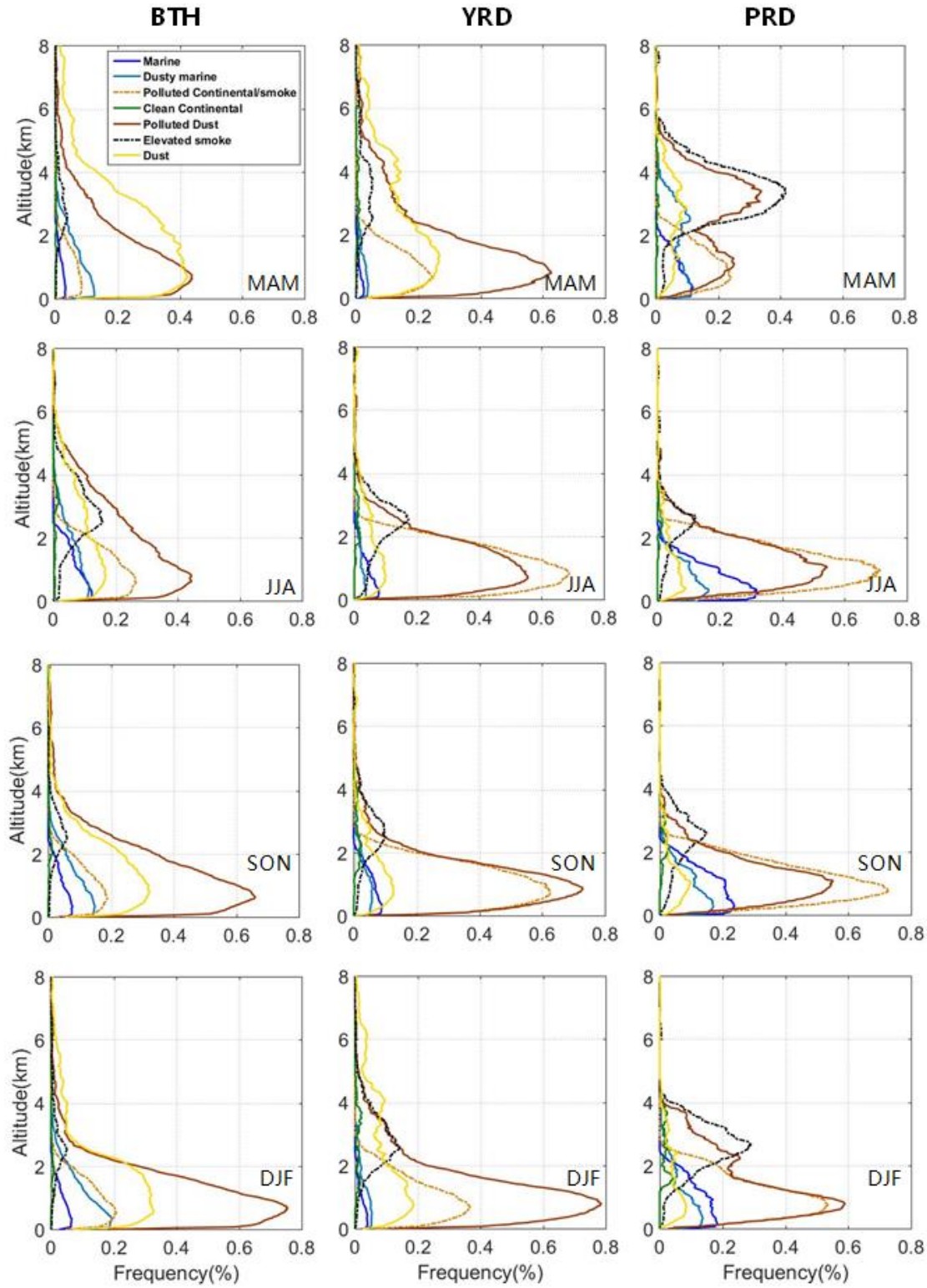

**Figure 10. Vertical distribution of the FO of different CALIOP aerosol types (see legend) during daytime, for each season over the BTH (left), YRD (middle) and PRD (right), averaged over the years 2007-2020.**

For polluted continental aerosol, however, the difference profiles in Fig. 11 clearly show that the daytime FO is highest in a layer adjacent to the surface (night minus day negative) which is clearly separated from the layer above where the FO is higher during the night. The higher daytime FO indicates the accumulation of this aerosol type in a turbulent mixed layer which expands under the

influence of solar heating. Furthermore, anthropogenic activities result in stronger emissions from, e.g., domestic activities and traffic, of particulate matter, aerosol precursor gases and secondary formation by photochemical processes, during the day than during the night. This results in a diurnal cycle with substantial day/night differences of AOD and particulate matter (Lennartson et al., 2018), with higher aerosol concentrations during daytime. After sunset, radiative cooling of the surface results in the formation of a stable nocturnal boundary attached to the surface which lifts the mixed layer into a disconnected residual layer (Stull, 1988), including the polluted continental aerosol which is thus observed at higher elevation. This separation is most clearly observed over the PRD during the summer and autumn, although the distribution over two different layers is also suggested by the profiles in winter and spring. The profiles over the YRD in the summer and autumn behave similarly, although weaker than over the PRD. In all these cases, polluted continental is the dominant aerosol type, or one of the most dominant aerosol types, in the lower 2 km. Polluted continental aerosol is emitted, or formed from precursor gases, near the surface and its transport to higher elevations is prohibited by the temperature inversion at the top of the mixed layer. The formation from precursor gases often involves a photo-chemical reaction, i.e. requires the availability of solar radiation and thus occurs during daytime.

In contrast, marine aerosol is directly emitted from the ocean in high wind conditions when waves break (de Leeuw et al., 2011). Marine aerosol is confined to the mixed layer (range A) and significant FOs of clean marine and dusty marine aerosol are mainly observed over the PRD, in all four seasons. Fig. 11 shows that the FO of dusty marine aerosol is higher during the day than during the night, whereas in contrast, the FO of clean marine aerosol is higher during the night. Over the ocean the air-sea temperature difference is much smaller than over land and thus nocturnal boundary layers are not formed over the ocean. Marine aerosol is transported to the study area, which is over land, in on-shore wind and hence marine aerosol is well distributed over the lower boundary layer.

Dust is long-range transported from the deserts in the north and west of China where it is emitted to high elevations before is passes over the mountains to east China (Proestakis et al., 2018). The day/nighttime difference profiles show a separation at altitudes of 2-4 km, above which the FO of dust is larger during the night while at lower altitudes the dust FO is larger during the day. The altitude depends on the season as is most clearly illustrated from the profiles over the BTH where the dust occurs most frequently: the separation is at about 4 km in the spring and summer, at about 3 km in the autumn and at about 2.5 km in the winter. Clearly, the dust aerosol is transported from above to the lower layers where it mixes with pollution to form polluted dust. The concentrations of clean continental aerosol are too small to discuss in terms of day/night differences, except over the PRD in the summer and autumn where it is observed in range C and behaves similar to elevated smoke aerosol, i.e. the FO is higher during the night than during the day.

Figure S11 shows the difference between the frequency of occurrence of aerosol layer top and base during nighttime and daytime observations (night minus day), for each season averaged over the years 2007 - 2020 over the three study regions. The higher frequency of occurrence of both aerosol layer top and base at higher altitudes (> 2 km) during the night may be caused by two effects. As discussed above, background sunlight reduces the SNR in the lidar signal which increases the fraction of

undetected aerosol layers with respect to the fraction of undetected layers at night (Liu et al., 2009; Kim et al., 2017). On the other hand, the CALIPSO overpasses are in the early afternoon and after midnight but the diurnal occurrence of deep convection and precipitation reach their maximum in the late afternoon or early evening (Huang et al., 2013). Hence, compared with daytime, during the night a higher frequency of occurrence of aerosol types at high altitudes is detected by CALIOP due to the deep convective activity.

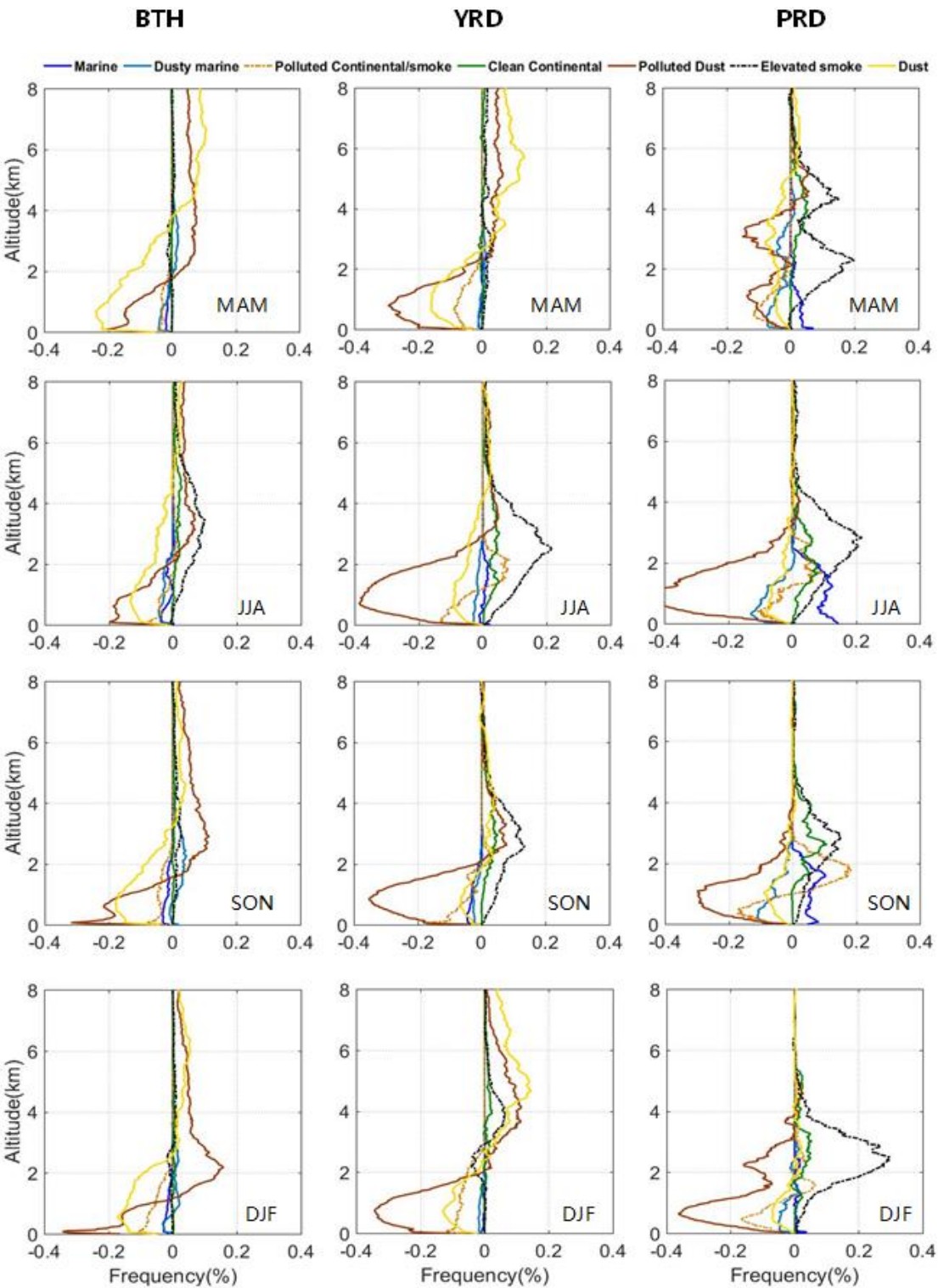

**Figure 11. Differences between nighttime and daytime vertical distributions of the FO of CALIOP-derived aerosol types (see legend) (nighttime minus daytime) by season, averaged over the years 2007-2020, over the BTH (left), YRD (middle) and PRD (right).**

**3.6 Air mass back trajectories and origin of aerosol over the three study regions**

In the above, the spatial distributions of the AOD and the vertical distributions of the aerosol types over the BTH, the YRD and the PRD were explained in terms of aerosol origin, local versus remote production and long-range transport, while also noting the CALIOP aerosol type classification method (Kim et al., 2018). To further illustrate the effect of transport and the differences between the

three study regions, 48-h backward air mass trajectories for each region and arriving at 500 m, 1000 m and 3000 m were computed as described in Sec. 2.2.4, for every day in the period 2007-2020. These air mass trajectories were clustered by season and the results for each study region are presented in Figures S12-S14. The trajectories clearly show the differences between the three regions, and for each region between seasons, and also the arrival height. Air masses arriving in the BTH

regions show the long-range transport from northerly and north-westerly directions, i.e. explaining the dust transport from the deserts such as Gobi and Taklamakan during all seasons except summer. During summer the origin of the 48-h air mass trajectories is relatively close to the BTH, especially due to the reduced transport form northwesterly directions. During summer, transport from south-southwesterly and south-easterly directions contributes more than during other seasons. During

all seasons there is a rather strong local contribution, weakening with distance to the BTH over a distance of the order of 1000 km. Part of the air masses originate over the Bohai sea, explaining a marine component in the aerosol types. The distribution of the air mass trajectories arriving at 1000 m is similar to that of the air masses arriving at 500 m, although some air masses go back over a somewhat longer distance. This also applies to air masses arriving at 3000 m, but the distances are

substantially longer and there is a wider distribution over the directions, in particular during the autumn and winter when there are more contributions from westerly and southwesterly directions.

The 48-h back trajectories arriving at the YRD at 500 m are much shorter than for the BTH, with stronger contributions from easterly and northerly directions, including those over the East China Sea and the Yellow Sea. Air masses originating from northwesterly directions contribute during all

seasons except summer. During summer the 48-h air mass trajectories originate from the east and south of China and over the oceans. In all seasons, only a very small fraction of the air masses originates from the west of China with Xinjiang (Taklamakan desert) and thus the observed dust aerosol type originates from the Gobi Desert in the north of China. Transport to the YRD from the north is an important factor, together with locally generated aerosol and regional transport from

eastern and southern China and the East China Sea. This effect is much stronger for air masses arriving at 3000 m, with much longer trajectories and, like over the BTH, a wider distribution of directions. In particular, the contribution of components from westerly directions (SW-NW) is substantially larger.

For 48-h air mass trajectories arriving at the PRD at 500 m the distribution is quite different from

those arriving at the BTH and YRD. The 48-h trajectories of the air masses arriving at the PRD are mostly shorter than over the other study areas and in the spring and summer from southerly directions over the South China Sea. In the autumn and winter the trajectories from the south are much shorter and there is a larger contribution from northerly and easterly directions (Central and East China). Also, in these seasons, a small fraction of the trajectories originates from northerly directions suggesting a

possible contribution of dust aerosol originating from the Gobi Desert. In the winter season, a substantial fraction of the trajectories originates from South-East Asia, over longer distances than in other seasons. The distribution of the 48-h air mass trajectories arriving over the PRD at 1000 m is similar to that over air masses arriving at 500 m, also as regards the lengths of the trajectories. The distribution of the air mass trajectories arriving at 3000 m is similar to those arriving at 500 m and

1000 m during the summer, but during other seasons more trajectories originate from north-westerly directions, with distinct difference between spring, autumn and winter and go back further than at lower altitudes. The longer trajectories may result in different aerosol types, transported form other regions, than at lower levels, as also observed in the CALIOP data.

**4 Discussion**

Based on the observations described above, the regional and seasonal variations of the spatial and vertical distributions of aerosol properties are discussed in the following.

-    In the summer, the AOD is highest over the BTH and YRD (Fig. 2b), which is attributed to more abundant water vapor and higher temperatures in the summer resulting in strong convection causing deeper boundary layers. The moist air results in higher RH and thus in the swelling of

aerosol particles, i.e. a shift of the particle size distribution to larger sizes which in turn results in higher extinction and AOD. The higher temperature results in faster chemical reactions and thus formation of secondary aerosol. Dynamically, the altitude of the peak FO of elevated smoke increases when the ascending motion of air masses occurs or in unstable atmosphere (Fig. 8 and Fig. 9). In addition, biomass burning is the main source of elevated smoke aerosol in the summer.

This is in line with elevated smoke aerosol being the second dominant aerosol type above 2 km over the BTH in the summer (see however Sect. 2.2.2). Over the YRD, elevated smoke is the dominant aerosol type above 2 km during the summer (Fig. 6). In contrast, the AOD is lowest over the PRD, which may be due to wet removal by precipitation during the East Asian summer monsoon. Fig. 6 also shows that the FO of clean marine aerosol over the PRD is highest in the

summer. Moreover, polluted continental aerosol dominates the aerosol below 3 km over the YRD and PRD in the summer.

-    In the spring, the AOD is highest over the PRD (Fig. 2b), which may be related to long-range transport of pollutants from biomass burning in southeast Asia (as discussed in Sect. 3.6). This is consistent with the observation that elevated smoke is the dominant aerosol type above 2 km,

following the CALIOP classification, and the FO extends to higher altitudes than in other seasons (Fig. 6). Over the BTH and YRD, dust and polluted dust dominate from near the surface to the upper troposphere, leading to the higher AOD over these two areas. Over the YRD, the altitude of the peak FO of dust in MAM and DJF is substantially higher (5 km) than over the BTH (Fig. 6),

which may be due to the long-range transport of dust aerosol by westerly and northerly winds from dust generated from the Taklamakan and Gobi deserts in the west and north of China (see Sect. 3.6).

- In the autumn, the AOD over the three regions is relatively low (Fig. 2b). In this season, the whole eastern region is dominated by westerly winds, which transports relatively clean air to the study areas and contributes to the diffusion of aerosol particles. The impact of dust storms is relatively small in the autumn.

- In the winter, the prevailing northerly winds bring dry and clean air to the study areas and hence the aerosol concentrations, and thus AOD, are low (Qi et al., 2013; Si et al., 2018). Nevertheless, high AOD does occur frequently in the winter, which is reflected in the average AOD of 0.7 over the BTH in February. This is attributed to the occurrence of high AOD during weather conditions conducive of the formation of haze (low wind speed, low BLH, stable stratification). The peak FO of elevated smoke is smaller and that of dust, polluted dust and polluted continental aerosol is larger when the atmosphere is stable (Fig. 9).

- With regard to the altitude of the peak FOs of the aerosol types over the three regions, the order from low to high altitude is overall as follows: dust > polluted dust > clean continental/elevated smoke > polluted continental > clean marine/dusty marine.

**5 Summary and Conclusions**

The four dimensional (time scales from yearly to monthly, horizontal and vertical) variations of aerosol properties over three major urban clusters in eastern China have been investigated using observations from passive (MODIS/Aqua) and active (CALIOP/CALIPSO) instruments, both flying on the A-Train satellite constellation, together with ERA Interim Reanalysis meteorological data and GDAS meteorological data, for the years 2007-2020. Three areas, BTH, YRD and PRD have been selected because of the diverse natural and anthropogenic aerosol sources as well as different climatic characteristics, providing a unique natural laboratory for the investigation of aerosol properties.

On the inter-annual scale, the highest average AOD occurs in 2011 over the YRD, and the highest AOD over the BTH and PRD are lower and occur in 2012. After 2011 and 2012, respectively, the AOD shows a decreasing trend as was also observed in other studies (de Leeuw et al., 2018; Sogacheva et al, 2018). Between 2007 and 2020, the average AOD over the three representative regions decreased overall, although in the last 3-4 years this decrease has become slower. On a seasonal scale, the AOD over the BTH peaks in the summer, whereas over the YRD and the PRD the AOD peak occurs in the spring. On a monthly scale, the summer AOD peaks over the BTH (0.79) and the YRD (0.84) occur in June. In contrast, over the PRD two AOD peaks are observed, one in March (0.68) and a weaker one in October (0.41), whereas, much lower AOD values occur in the summer with a clear minimum in July. Regarding the AOD spatial distribution over the three regions, the AOD over the BTH is high in the Southeast over Hebei and Shandong and low over the mountains in Shanxi province in the Northwest. The AOD over the YRD is lower over the Zhejiang and southern Anhui provinces as compared to other areas during the whole year. Over the PRD, the AOD spatial distribution shows a

ring-shaped declining pattern, with the highest values in the inner ring and decreasing toward the outer ring.

Comparing the aerosol types in the three urban clusters, the rFOs of clean ocean, polluted continental, clean continental and smoke aerosol were the lowest over the BTH, while over the PRD they were highest. In contrast, the rFOs of polluted dust and dust aerosol over the BTH dominated the aerosol composition, while over the PRD the contributions of these types were the lowest. The altitude dependence of the frequencies of occurrence of aerosol types was also investigated. Over the BTH, the top two dominating aerosol types in the altitude range from 1.8 km up to 8 km are dust and polluted dust in all seasons, except for the summer. Elevated smoke (see comment in Sect. 2.2.2 on CALIOP classification of elevated smoke) is the second dominant aerosol type above 2 km and polluted continental is the second dominant aerosol type below 2 km in the summer. Over the YRD, dust is detected from near the surface to the upper troposphere and the altitude of the peak FO is about 5 km in the spring and winter. Elevated smoke dominates at altitude ranges from 2 km to 5 km in the summer and autumn, while polluted continental is dominant below 2 km. Over the PRD, elevated smoke is the dominant aerosol type above ~ 2 km in all seasons and clean marine is the second most frequently observed aerosol type below ~ 2 km in the summer and autumn. Moreover, polluted continental in the three study regions is always observed close to the ground in all seasons. In addition, the FOs of polluted continental and clean marine aerosol are larger over the PRD than over the BTH and YRD.

The change in the distribution of the frequency of occurrence of the aerosol types with increasing AOD shows that the peak FOs of clean continental aerosol and clean marine aerosol gradually decrease with increasing AOD in the three regions. In heavily polluted conditions, the peak FO of elevated smoke aerosol at an altitude of ~ 2.5 km is largest over the three regions. The FOs of elevated smoke and dust aerosol in low AOD conditions occur at higher altitude than during the other two AOD conditions. The extinction coefficient of the aerosol below 6 km is lowest over the PRD and highest over the YRD.

The variation of the aerosol vertical distribution was also analysed in terms of dynamic and thermodynamic boundary layer conditions. Dynamically, the downward motion of air parcels can increase the FOs of polluted dust at 1 km. With regard to thermal stability and vertical mixing, using LTS as a proxy, the peak FOs of dust and polluted dust increase below 2 km when the atmosphere becomes more stable. Conversely, the peak FOs of elevated smoke around 2.5 km gradually decrease with the increase of LTS or when air masses descend.

In this study, nighttime CALIOP observations were used to study the vertical distribution of aerosol types and extinction coefficients. During the night, meteorological conditions, emissions, atmospheric chemistry and aerosol processes are different from those during the day. The two CALIPSO overpasses, at 1:30 am and pm local time, were used to evaluate daytime/nighttime differences between the vertical distributions of the frequency of occurrence of CALIOP-derived aerosol types. These differences are influenced by the CALIOP detection and processing approach and further depend on the aerosol type, altitude, season and location and the analysis suggests effects of aerosol transport and boundary layer processes on the vertical distribution of different aerosol types.

Air mass trajectories show the differences in the origin of the aerosol observed over the three study areas. The distributions the 48-h air mass trajectories during the four seasons show substantial

differences between the directions from which air masses are transported to the three study areas, and thus the origin of the aerosol. These air mass distributions vary by season, in particular during the summer they are much different from the distributions in other seasons. The air mass trajectory

distributions also vary with height, not only as regards the length of the trajectory but also as regards their origin. Hence, the aerosol types may vary with height, as observed, due to different origins of the aerosol observed at different heights. It is noticed that the CALIOP aerosol type classification method influences the observations and introduces uncertainties, but do not lead to contradiction in the interpretation.

In summary, the aerosol properties, aerosol types and vertical profiles in different AOD and meteorological conditions over three representative regions over China were described, using synergetic use of aerosol products from active and passive sensors. Air mass trajectories were used to explain the transport pathways to the three study areas. The nature of aerosol effects on Earth's climate depends strongly on the aerosol vertical distribution. When absorbing aerosol is located above bright

clouds, warming effects are amplified. The atmospheric lifetime of aerosol in the free troposphere is much longer than the boundary layer. Aerosol in the free troposphere is transported further away from its sources than at lower altitudes, which further affects the geographic pattern of aerosol impacts. The vertical distribution of tropospheric aerosol is especially valuable for evaluation of global aerosol models because it is a signature of the combined effects of aerosol emissions, the strength of vertical

lifting and exchange, atmospheric transport patterns, and removal processes (Winker et al., 2013). The results from this study can be used to improve model assessment of the direct and indirect aerosol effects in eastern China (Wang et al., 2011; Wu et al., 2016). In addition, aerosol particles also play an adverse role on air quality and human health and bring about millions of premature deaths in the world (Chen et al., 2020). The integrated mass of dry particles (PM2.5) related to AOD is often used as an

indicator for evaluating air quality and human health (van Donkelaar et al., 2016). The aerosol vertical distributions add value in air quality forecasting and human health research due to its relationship with AOD.

*Data availability*

All data used in this study are publicly available. The satellite data from the MODIS instrument used in

this study were obtained from https:// ladsweb.modaps.eosdis.nasa.gov/search/ (last access: 20 May 2021). The satellite data from the CALIOP instrument used in this study were obtained from https://subset.larc.nasa.gov/calipso/ (last access: 20 May 2021). The ECMWF ERA-Interim data were collected from the ECMWF data server http://apps.ecmwf.int/datasets/data/interim-full-daily/levtype=pl/ (last access: 20 May 2021). The

GDAS meteorological data were collected from ftp://arlftp.arlhq.noaa.gov/pub/archives/gdas1/(last access: 20 May 2021).

*Author contributions*

YL and TL designed the research. YL led the analyses. YL and GL wrote the manuscript with major

input from JH and further input from all other authors. All authors contributed to interpreting the
results and to the finalization and revision of the manuscript.

*Competing interests*

The authors declare that they have no conflict of interest.

*Acknowledgements*

This work was supported by the National Natural Science Foundation of China (Grant No. 42001290),
the China Postdoctoral Science Foundation (Grant No. 2018M630733) and the CAS Strategic Priority
Research Program (Grant No. XDA19030402). Many thanks are expressed to NASA for making
available the MODIS and CALIOP data. We are grateful for the easy access to the daily ERA Interim
Reanalysis data provided by ECMWF. We also thank the reviewers of this paper for their valuable
comments which helped improve the manuscript.

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
