# Peer review of "Multi-dimensional satellite observations of aerosol properties and aerosol types over three major urban clusters in eastern China"

_Atmospheric Chemistry and Physics, 2020_

## Author Response (AR1)

**Reply to comments by Reviewer#1 on "Multi-dimensional satellite observations of aerosol properties and aerosol types over three major urban clusters in eastern China"**

June 09, 2021

We thank the reviewer for the thorough reading of the manuscript and the thoughtful comments which are helpful not only for this manuscript but also for our future research. Our replies to all comments are shown below in red.

**Comments**

**1. In the framework of the study CALIPSO nighttime observations are used. The authors justify their choice through the illumination conditions and the related lower daytime SNR. However, I would suggest to extend their justification through CALIPSO daytime and nighttime performance, as provided in the literature.**

**Answer**: We made the following change in the revised manuscript (see pg. 7 lines 226-237, pg.28 lines 696-710 and pg. 31-32 lines 766-776).

The text "Daytime signals can be affected by background sunlight and reduce the SNR (signal to noise ratio), resulting in a larger fraction of undetected aerosol layers during daytime than during nighttime, and underestimation of the CALIOP extinction coefficients and AOD which is larger during daytime than during nighttime (Kim et al., 2017). The larger fraction of undetected aerosol layers during daytime may also lead to underestimation of the frequency of occurrence (FO) of daytime aerosol types, especially in the upper level (Huang et al., 2013). An overview of the evaluation of the CALIOP AOD versus other measurements shows a low bias of the CALIOP AOD of the order of about 30%. Kim et al. (2018) shows that the CALIOP V4 AOD is still biased low over ocean but less than V3. Over land the V4 vs V3 improvement was not quantified because of the larger uncertainties in the MODIS AOD data which are used as reference. To avoid day/night differences in the CALIOP data, in this study only nighttime measurements at 532 nm were used to investigate the vertical distribution of aerosol types and extinction coefficients." was added into section 2.2.2 in the revised manuscript (see pg. 7 lines 226-237).

The text "However, as discussed in Sect. 2.2.2, the occurrence of undetected aerosol layers results in underestimation of the CALIOP extinction coefficients and AOD and the fraction of undetected aerosol layers is larger during daytime than during nighttime. Furthermore, aerosol in the boundary layer is relatively well detected as compared with aerosol near the top of the boundary layer (Kim et al., 2017) which may lead to distortion of the vertical profile. Hence day/night differences may occur between the vertical distributions of the aerosol type FO due to the CALIOP

processing which affect the interpretation of day/night differences in the profiles. Day/night differences in the vertical structure of aerosol properties are expected due to natural processes such as direct production or formation of secondary aerosol, transformation of aerosol particles in the atmospheric boundary layer, vertical mixing and transport from remote locations, wet and dry deposition. Below the observed day/night differences are briefly discussed based on consideration of such processes. Separation of these effects from those due to the CALIOP processing, and determination of their relative importance, are beyond the scope of the current study. This would require a study on the effects of biases in the retrievals, detection thresholds, noise and the influence of quality control flags." was added into section 3.5 in the revised manuscript (see pg.28 lines 696-710).

The text "Figure S11 shows the difference between the frequency of occurrence of aerosol layer top and base during nighttime and daytime observations (night minus day), for each season averaged over the years 2007 - 2020 over the three study regions. The higher frequency of occurrence of both aerosol layer top and base at higher altitudes (> 2 km) during the night may be caused by two effects. As discussed above, background sunlight reduces the SNR in the lidar signal which increases the fraction of undetected aerosol layers with respect to the fraction of undetected layers at night (Liu et al., 2009; Kim et al., 2017). On the other hand, the CALIPSO overpasses are in the early afternoon and after midnight but the diurnal occurrence of deep convection and precipitation reach their maximum in the late afternoon or early evening (Huang et al., 2013). Hence, compared with daytime, during the night a higher frequency of occurrence of aerosol types at high altitudes is detected by CALIOP due to the deep convective activity." was also added into section 3.5 in the revised manuscript (see pg. 31-32 lines 766-776).

**2. In the end of Section 2.1, a map encompassing East China, delineating the three BTH, YRD, and PRD study regions, would be practical for the reader.**
**Answer**: Thank you for this advice. We have added the following map to Section 2.1 showing East China and delineating the three BTH, YRD, and PRD study regions with coordinates given in the caption (see pg. 5 in the revised manuscript).

[Figure]

Figure 1. Elevation map of Eastern China showing the study areas, i.e. Beijing-Tianjin-Hebei (BTH, 35.5°N-40.5°N and 113.5°E-120.5°E), the Yangtze River Delta (YRD, 28°N-33°N and 117°E-122°E) and the Pearl River Delta (PRD, 21.5°N-24.5°N and 111.5°E-115.5°E). These areas are indicated by the black rectangles. The elevation data is downloaded from the website https://search.earthdata.nasa.gov/ (last access: 20 May 2021).

**3. I would suggest to the authors to break down Section 2.2 - "Data sources" into three distinct sub-sections: "MODIS/Aqua", "CALIOP/CALIPSO" and "ERA-Interim".**
**Answer**: Thank you, we have followed this advice (see pg.5-7 in the revised manuscript).

**4. Whenever a web-link is provided, please follow the formalism on adding in a parenthesis the "last visit:" information (e.g. line 137).**
**Answer**: We made these changes for every occurrence in the revised manuscript.

**5. Lines 141-143 "Over China, the differences between the C6 and C6.1 AOD are small, except over certain areas like the Tibetan Plateau, Sichuan Province and the NW of China". Please include related references.**
**Answer**: We have added the relevant references in the revised manuscript (see pg. 6 line 188-189 in the revised manuscript).

**6. Why have the authors selected MODIS AOD 1.5 and CALIPSO AOD 3? Please include references. Moreover, regarding comparison methodology, I would argue the use of similar CALIPSO and MODIS upper AOD limits.**
**Answer**: In the revised version we have used the latest MODIS C6.1, merged DTDB product and the CALIOP V4.10 product, as described in section 2.2 in the revised manuscript (see pg. 5-7 in the revised manuscript). Also we changed the MODIS

AOD limit to 3.0, so CALIPSO and MODIS upper AOD limits are similar: "cases with MODIS AOD greater than 1.5" has been changed to "cases with MODIS AOD greater than 3.0" in the revised manuscript (see pg. 8 line 260-261 in the revised manuscript). This issue is shown throughout the revised manuscript (all the figures were changed/modified in this respect).

**7. Please include information of the pre-processing of MODIS/Aqua. Which Quality Assurance procedures and flags are used? Based on Figure 2, the authors have used a re-griding procedure in MODIS L2 AOD, which is not mentioned. Is any final smoothing applied to the data? Moreover, Figure 2 has same "blank" areas, without AOD values. Please discuss these aspects/address in the manuscript.**

**Answer**: About the pre-processing of MODIS/Aqua, cases with MODIS AOD greater than 3.0 are discarded in the analysis (see pg. 8 line 260-261 and our response to the previous comment nr 6). Based on Figure 3 in the revised manuscript, the spatial variation of the MODIS AOD over the three urban clusters, averaged over the seasons in the years from 2007 to 2020 are based on in 0.1°x 0.1° (see pg. 12 line 369-370 in the revised manuscript). The white pixels in figure 3 in the revised manuscript mean the data deficient in the location. The blank areas were caused in part by the selection of AOD < 1.5 (in the previous version). Setting the threshold to 3.0 does not completely remove the blank spots since pixels with no data still remain because in very polluted conditions, the retrieval of high AOD values often encounters difficulties due to discrimination between very high AOD and clouded situations.
We have re-organized the text "The spatial distributions of the seasonal mean MODIS AOD over the three urban clusters, averaged over the years from 2007 to 2020, and plotted with a resolution of 0.1°x 0.1°, are shown in Fig. 3. In Fig. 3, some small areas occur where no data are available; these areas are left white. As mentioned in Sect. 2.3, MODIS data with AOD>3.0 were discarded. It is noted that aerosol retrieval over areas with very high AOD may not be successful due to problems with discrimination between high AOD and clouds. The AOD>3.0 threshold avoids confusing cloudy pixels as high AOD cases." (see pg. 12 line 369-374 in the revised manuscript).

**8. "CALIOP is the first space-borne near-nadir dual-wavelength lidar (532 nm and 1064 nm)". This is not correct (e.g. ICESat). Please revise.**
**Answer**: Thank you, that is correct. We have changed the sentence to "CALIOP is a space-borne near-nadir dual-wavelength lidar (532 nm and 1064 nm) that provides high-resolution vertical profiles of aerosol and clouds." (see pg. 6 line 201-202 in the revised manuscript).

**9. Lines 155-161: The authors discuss the different levels of processing in CALIPSO algorithms. Please mention which refer to L1B and which to L2, and also that the study is based on L2. Moreover, add the word "Version" beside the "4.1".**
**Answer**: Thank you we have added this information in Sect. 2.2.2 (see pg. 6 in the

revised manuscript) and elsewhere.

**10. The authors mention that AOD is the vertical integration of extinction. However, it is not discussed in the manuscript how the mean AOD is calculated. The official and more robust way, as discussed in Amiridis et al. (2013) and Tackett et al. (2018), is calculating the mean quality-assured extinction coefficient profile at the overpass level - based on L2 profiles per overpass, and accordingly using all overpass-mean profiles to calculate the seasonal or annual profile. Not following this approach results in weighting effects, thus in not representative results. Please provide in the manuscript the methodology followed in the processing of CALIPSO profiles at mean-extinction coefficient and AOD at 532nm, and if the methodology is different, make necessary corrections.**

**Answer**: Thank you for this valuable suggestion. We have followed this suggestion and re-calculated the mean extinction profiles accordingly and introduced them in Figure 5 in the revised manuscript (see pg.18). The text has been revised as you explained above and added it in the revised manuscript (see pg.16 line 415-423): "This distinction was made based on the CALIOP AOD (obtained by integration of the extinction coefficient profiles over the tropospheric column) which was used to divide the profiles into three equally sized subsets. A histogram of CALIOP AOD values showing the different categories and corresponding number of cases for each region are reported in Fig. S2 and Table S10. The annual mean extinction coefficient profiles were calculated following procedures discussed in Amiridis et al. (2013) and Tackett et al., (2018). The mean of the quality-assured extinction coefficient profiles was first calculated at the overpass level - based on L2 profiles per overpass. Then seasonal and annual profiles were calculated using the mean profiles for all overpasses." (see pg.16 line 443-451 in the revised manuscript).

**11. Lines 168-169: "This is further illustrated …". Add in the end the Section/Figure that support this sentence.**

**Answer**: This is illustrated in Sect. 3.5 and Fig. 11 in the revised manuscript. We have added a reference to Sect. 3.5 at the end of Sect. 2.2.2 (see pg.7 line 235-238 in the revised manuscript).

The text "This is further illustrated with the comparison of vertical distributions of the FO of the aerosol types during day- and night-time overpasses. To avoid such problems, nighttime measurements at 532 nm were used in this study to investigate the vertical distribution of aerosol types and extinction coefficients. The vertical distribution of the frequencies of occurrence of CALIOP-derived aerosol types during nighttime are compared with those derived during daytime." has been changed to "To avoid day/night differences in the CALIOP data, in this study only nighttime measurements at 532 nm were used to investigate the vertical distribution of aerosol types and extinction coefficients. The vertical distribution of the frequencies of occurrence of CALIOP-derived aerosol types during nighttime are compared with those derived during daytime in Sect. 3.5.". (see pg.7 line 235-238in the revised

manuscript).

**12. Lines 174-180: Profiles of RH are provided in CALIPSO L2 Aerosol and Cloud Profiles, based on MERRA-2 model, which is used in the algorithms of CALIPSO in order to produce the different optical products. Possible use of ERA-Interim may results in some point of model-intercomparison. Have the authors considered the use of RH from CALIPSO datasets?**

**Answer**: We did use RH in our initial analysis, in Sect. 3.4.1, but in the revised version we decided to remove that section as well as references to analysis in terms of RH.

**13. Line 182: "cloud-free pixels". What is a pixel? Please define clarify in the manuscript. Is it the L2 Profile, the grided L3 profile, or the region (e.g. BTH region) to be cloud-free?**

**Answer**: We made the following change in the revised manuscript (see pg.8 line 256 in the revised manuscript). The sentence "Aerosol properties are only retrieved for strictly cloud-free pixels" has been changed to "Aerosol properties are only retrieved for strictly cloud-free pixels in the MODIS Level 2 products".

**14. It is not clear why the authors have used the limited period between 2007 and 2015. Since this period has already been discussed in Proestakis et al. (2017), it would be interesting and of added-value to include more years in the analysis. Expanding the timeseries would be of added value for timeseries analysis, especially due to the special orbital characteristic and overpass frequency of CALIPSO. Moreover, extending the observational period would address the question whether emissions have increased after 2015 or whether they are still declining due to regulations applied.**

**Answer**: This is correct, we have included data up to and including 2020 and re-analyzed all data for the whole acquisition period between 2007 and 2020. This issue is shown throughout the revised manuscript (all the figures were changed/modified in this respect). In deed the AOD continued to decrease after 2015 although in the last 3-4 years the decrease appears to be less than in the earlier years.

**15. In Section 3.1, please provide in the manuscript the trends, the related statistical significance, and discuss the outcomes.**

**Answer**: The trends of the AOD over the three target regions are now described in the revised manuscript. The related statistical significance also has been added by "The annual mean AOD averaged over the whole study period is smallest in the PRD with a value of 0.41±0.09 (annual mean ± standard deviation); over the BTH and the YRD the annual mean AODs averaged over the study period have similar values of 0.56±0.07 and 0.55±0.09, respectively (See Supplement Table S1).", "Detailed statistics of the seasonally averaged MODIS AOD are provided in Table S2 of the Supplement." and "The monthly mean data for each year and in each region, statistics and the number of overpasses included in the monthly averaged MODIS AOD are

provided in Tables S3-S8 of the Supplement." in the revised manuscript (see pg.9 line 292-296, line 303-304, and pg.12 line 345-347 in the revised manuscript).

**16. In Figures 1b and 1c please include vertical error-bars/uncertainty-bars. Moreover, what is missing is the information on the number of overpasses/profiles used for the calculation. Discuss in the manuscript the outcomes in combination with the number of overpasses/profiles, with respect to the representativeness of the results.**

**Answer**: Vertical error-bars have been included in Figure 2b and 2c in the revised manuscript (see pg.11). And the number of overpass/profiles used for the calculation are now reported in Table S6-S8 in the supplement. The data show that the number of overpasses in the BTH is generally highest and in the PRD is lowest. However, the number of overpasses in each month are almost the same in each region. The text "The monthly mean data for each year and in each region, statistics and the number of overpasses included in the monthly averaged MODIS AOD are provided in Tables S3-S8 of the Supplement." in the revised manuscript (see pg.12 line 345-347 in the revised manuscript).

Table S6. The total number of MODIS overpasses over the BTH during each month in the period from 2007-2020.

| BTH | | | | | | | | | | | | |
|---|---|---|---|---|---|---|---|---|---|---|---|---|
| Year Month | Jan | Feb | Mar | Apr | May | Jun | Jul | Aug | Sep | Oct | Nov | Dec |
| 2007 | 55 | 55 | 57 | 54 | 54 | 56 | 59 | 57 | 55 | 55 | 53 | 64 |
| 2008 | 56 | 57 | 54 | 55 | 55 | 57 | 55 | 55 | 54 | 59 | 57 | 66 |
| 2009 | 56 | 55 | 56 | 55 | 57 | 55 | 52 | 57 | 58 | 56 | 54 | 65 |
| 2010 | 53 | 54 | 58 | 55 | 54 | 52 | 58 | 56 | 55 | 56 | 56 | 65 |
| 2011 | 56 | 56 | 54 | 58 | 56 | 55 | 55 | 56 | 57 | 57 | 56 | 66 |
| 2012 | 55 | 57 | 55 | 57 | 56 | 55 | 54 | 57 | 57 | 55 | 55 | 67 |
| 2013 | 57 | 56 | 55 | 56 | 57 | 55 | 55 | 56 | 56 | 57 | 53 | 64 |
| 2014 | 59 | 58 | 55 | 55 | 55 | 57 | 55 | 56 | 54 | 57 | 55 | 65 |
| 2015 | 54 | 54 | 56 | 55 | 55 | 54 | 58 | 54 | 56 | 55 | 56 | 67 |
| 2016 | 56 | 53 | 56 | 54 | 59 | 56 | 55 | 55 | 55 | 57 | 57 | 65 |
| 2017 | 54 | 57 | 55 | 58 | 56 | 56 | 53 | 55 | 54 | 55 | 56 | 65 |
| 2018 | 57 | 55 | 56 | 57 | 56 | 54 | 56 | 56 | 55 | 57 | 57 | 64 |
| 2019 | 57 | 55 | 58 | 55 | 56 | 55 | 57 | 57 | 57 | 56 | 57 | 65 |
| 2020 | 55 | 56 | 55 | 56 | 55 | 57 | 57 | 36 | 46 | 56 | 55 | 69 |

Table S7. The total number of MODIS overpasses over the YRD during each month in the period from 2007-2020.

| Year / Month | Jan | Feb | Mar | Apr | May | Jun | Jul | Aug | Sep | Oct | Nov | Dec |
|---|---|---|---|---|---|---|---|---|---|---|---|---|
| 2007 | 48 | 46 | 52 | 48 | 46 | 48 | 47 | 47 | 47 | 47 | 47 | 51 |
| 2008 | 47 | 47 | 48 | 47 | 49 | 50 | 47 | 49 | 51 | 49 | 47 | 56 |
| 2009 | 48 | 48 | 47 | 49 | 48 | 49 | 45 | 48 | 50 | 47 | 47 | 58 |
| 2010 | 46 | 47 | 48 | 51 | 49 | 48 | 49 | 47 | 50 | 51 | 49 | 58 |
| 2011 | 52 | 49 | 46 | 48 | 50 | 47 | 45 | 45 | 50 | 50 | 47 | 53 |
| 2012 | 48 | 49 | 46 | 52 | 51 | 46 | 47 | 48 | 49 | 47 | 47 | 59 |
| 2013 | 48 | 46 | 50 | 51 | 52 | 51 | 49 | 48 | 48 | 48 | 48 | 55 |
| 2014 | 52 | 49 | 48 | 49 | 49 | 50 | 49 | 47 | 48 | 48 | 51 | 56 |
| 2015 | 49 | 50 | 48 | 50 | 49 | 47 | 50 | 49 | 46 | 48 | 49 | 58 |
| 2016 | 46 | 45 | 50 | 50 | 55 | 50 | 48 | 45 | 45 | 49 | 46 | 53 |
| 2017 | 49 | 48 | 50 | 48 | 49 | 49 | 47 | 45 | 49 | 48 | 48 | 57 |
| 2018 | 49 | 49 | 46 | 49 | 51 | 50 | 50 | 49 | 49 | 48 | 48 | 55 |
| 2019 | 47 | 49 | 50 | 51 | 45 | 49 | 47 | 49 | 49 | 50 | 49 | 56 |
| 2020 | 49 | 47 | 46 | 48 | 48 | 49 | 48 | 33 | 40 | 47 | 48 | 61 |

Table S8. The total number of MODIS overpasses over the PRD during each month in the period from 2007-2020.

| Year / Month | Jan | Feb | Mar | Apr | May | Jun | Jul | Aug | Sep | Oct | Nov | Dec |
|---|---|---|---|---|---|---|---|---|---|---|---|---|
| 2007 | 38 | 41 | 41 | 39 | 40 | 38 | 38 | 39 | 40 | 38 | 40 | 45 |
| 2008 | 39 | 39 | 40 | 38 | 38 | 36 | 39 | 38 | 40 | 36 | 37 | 49 |
| 2009 | 39 | 39 | 38 | 38 | 38 | 38 | 39 | 38 | 34 | 40 | 39 | 43 |
| 2010 | 39 | 39 | 35 | 36 | 36 | 40 | 37 | 37 | 39 | 40 | 40 | 46 |
| 2011 | 40 | 38 | 42 | 38 | 38 | 37 | 40 | 38 | 36 | 37 | 38 | 47 |
| 2012 | 38 | 40 | 37 | 37 | 37 | 40 | 40 | 37 | 38 | 38 | 39 | 48 |
| 2013 | 39 | 38 | 39 | 37 | 37 | 38 | 40 | 39 | 38 | 39 | 40 | 47 |
| 2014 | 36 | 38 | 39 | 39 | 41 | 38 | 38 | 40 | 38 | 39 | 41 | 46 |
| 2015 | 37 | 40 | 40 | 38 | 38 | 40 | 36 | 39 | 36 | 37 | 38 | 41 |
| 2016 | 37 | 39 | 35 | 40 | 39 | 38 | 38 | 40 | 38 | 39 | 39 | 48 |
| 2017 | 37 | 38 | 38 | 38 | 37 | 36 | 38 | 39 | 38 | 38 | 36 | 44 |
| 2018 | 39 | 37 | 39 | 38 | 35 | 39 | 39 | 40 | 40 | 38 | 37 | 43 |
| 2019 | 39 | 38 | 39 | 38 | 40 | 38 | 37 | 36 | 38 | 36 | 37 | 44 |
| 2020 | 37 | 37 | 40 | 40 | 40 | 38 | 38 | 26 | 30 | 38 | 38 | 44 |

**17. In the figures, please add the sensor name. For instance in Figure 1, modify the caption to "CALIPSO annually (a), …".**

Answer: Thank you for this comment. Actually in Fig. 1 (Fig. 2 in the revised manuscript) we used MODIS data, so this already shows that indeed it is needed to mention which data set is presented. We made this change for Fig. 2 (see pg. 11 in the

revised manuscript) and all relevant figures thereafter.

**18. Although the language is fluent and the manuscript smooth, at some points the language could be more formal (e.g. line 253: "The monsoon brings heavy rains which effectively washout aerosols").**

**Answer**: "The monsoon brings heavy rains which effectively washout aerosols" has changed to "The monsoon is accompanied by heavy rain resulting in the effective removal of aerosol particles by wet deposition" in the revised manuscript (see pg. 11 lines 340-341 in the revised manuscript).

**19. Throughout the manuscript, in the framework of the discussion of the different types of aerosols, the related sources, and atmospheric mass origin, phrases such as "likely", "maybe" and more are frequently used. To this end, of high value would be to add aerosol back trajectories cluster analysis, to strengthen the discussion, and avoid hypothesis used, especially in this extend.**

**Answer**: Thank you for this comment. The 48-h backward air mass trajectories over the three target regions are now described in the revised manuscript.

The text "Backward trajectories of the air masses arriving at the center of the three study areas BTH (38°N, 117°E), YRD (30.5°N, 119.5°E) and PRD (23°N, 113.5°E) were determined using HYSPLIT (https://www.ready.noaa.gov/HYSPLIT_traj.php). The air mass trajectories were determined for the arrival points at heights of 500 m, 1000m and 3000m, i.e. the centers of the height ranges with high frequency of occurrence of the different aerosol types as determined from CALIOP data (Sect. 3.3). The air mass back trajectories were determined over 48 hours, at steps of 6 hours." has been added in Sect. 2.2.4 with caption "2.2.4 Air mass trajectories" (see Sect. 2.2.4, lines 248-254 in the revised manuscript).

The text "In the above, the spatial distributions of the AOD and the vertical distributions of the aerosol types over the BTH, the YRD and the PRD were explained in terms of aerosol origin, local versus remote production and long-range transport, while also noting the CALIOP aerosol type classification method (Kim et al., 2018). To further illustrate the effect of transport and the differences between the three study regions, 48-h backward air mass trajectories for each region and arriving at 500 m, 1000 m and 3000 m were computed as described in Sec. 2.2.4, for every day in the period 2007-2020. These air mass trajectories were clustered by season and the results for each study region are presented in Figures S12-S14. The trajectories clearly show the differences between the three regions, and for each region between seasons, and also the arrival height. Air masses arriving in the BTH regions show the long-range transport from northerly and north-westerly directions, i.e. explaining the dust transport from the deserts such as Gobi and Taklamakan during all seasons except summer. During summer the origin of the 48-h air mass trajectories is relatively close to the BTH, especially due to the reduced transport form northwesterly directions. During summer, transport from south-southwesterly and south-easterly directions contributes more than during other seasons. During all seasons there is a rather strong local contribution, weakening with distance to the BTH over a distance of the order of 1000 km. Part of the air masses originate over the Bohai sea, explaining a marine

component in the aerosol types. The distribution of the air mass trajectories arriving at 1000 m is similar to that of the air masses arriving at 500 m, although some air masses go back over a somewhat longer distance. This also applies to air masses arriving at 3000 m, but the distances are substantially longer and there is a wider distribution over the directions, in particular during the autumn and winter when there are more contributions from westerly and southwesterly directions.

The 48-h back trajectories arriving at the YRD at 500 m are much shorter than for the BTH, with stronger contributions from easterly and northerly directions, including those over the East China Sea and the Yellow Sea. Air masses originating from northwesterly directions contribute during all seasons except summer. During summer the 48-h air mass trajectories originate from the east and south of China and over the oceans. In all seasons, only a very small fraction of the air masses originates from the west of China with Xinjiang (Taklamakan desert) and thus the observed dust aerosol type originates from the Gobi Desert in the north of China. Transport to the YRD from the north is an important factor, together with locally generated aerosol and regional transport from eastern and southern China and the East China Sea. This effect is much stronger for air masses arriving at 3000 m, with much longer trajectories and, like over the BTH, a wider distribution of directions. In particular, the contribution of components from westerly directions (SW-NW) is substantially larger.

For 48-h air mass trajectories arriving at the PRD at 500 m the distribution is quite different from those arriving at the BTH and YRD. The 48-h trajectories of the air masses arriving at the PRD are mostly shorter than over the other study areas and in the spring and summer from southerly directions over the South China Sea. In the autumn and winter the trajectories from the south are much shorter and there is a larger contribution from northerly and easterly directions (Central and East China). Also, in these seasons, a small fraction of the trajectories originates from northerly directions suggesting a possible contribution of dust aerosol originating from the Gobi Desert. In the winter season, a substantial fraction of the trajectories originates from South-East Asia, over longer distances than in other seasons. The distribution of the 48-h air mass trajectories arriving over the PRD at 1000 m is similar to that over air masses arriving at 500 m, also as regards the lengths of the trajectories. The distribution of the air mass trajectories arriving at 3000 m is similar to those arriving at 500 m and 1000 m during the summer, but during other seasons more trajectories originate from north-westerly directions, with distinct difference between spring, autumn and winter and go back further than at lower altitudes. The longer trajectories may result in different aerosol types, transported form other regions, than at lower levels, as also observed in the CALIOP data." has been added in in Sect. 3.6 with caption "3.6 Air mass trajectories and origin of aerosol over the three study regions" (see Sect. 3.6 lines 783-830 in the revised manuscript).

**20. In Figure 2 add annual figures and corresponding discussion.**

**Answer:** We have added annual figures in Figure S1 showing maps of the spatial distribution of AOD from MODIS and CALIOP, with statics information in Table S9

(see supplement file).

Figure S1 and Table S9 are discussed in the revised manuscript as a pre-amble of Sect. 3.2: "Maps showing the spatial variation of the annual mean AOD over the three study areas, derived from MODIS and CALIOP data and averaged over the whole study period (2007-2020), are presented in Fig. S1. Statistical information on these data is summarized in Table S9. Figure S1 shows that the spatial patterns of the MODIS and CALIOP AODs are similar. However, the CALIOP AOD is clearly smaller than that from MODIS, as quantitatively illustrated by the data in Table S9. Underestimation of the AOD by CALIOP has been reported and explained in the literature (cf. Kim et al., 2017, for an overview). It is noted that the comparison in Fig. S1 and Table S9 was made for all available samples and no selection was made based on collocation. Comparison of the maps in Fig. S1 clearly shows the much smaller number of samples in the CALIOP data, due to the much smaller coverage of CALIOP as a result of the smaller swath width and thus substantially smaller number of CALIOP overpasses (Table S9). Hence, the differences between the MODIS and CALIOP AOD are likely augmented by the highly non-uniform data sample and to the fundamentally different algorithms and operation of the sensors. This was also reported by de Leeuw et al. (2018). In view of these differences, the spatial variation and time series analysis is made using MODIS data, whereas the vertical information is provided from CALIOP observations." (see pg.12 lines 353-367 in the revised manuscript).

**21. Figure 3: use the CALIPSO official colors if possible, to the aerosol subtypes.**

**Answer**: Thank you for this suggestion, it will help readers familiarize with CALIOP to read the manuscript easier. So we made this change in all relevant figures in the revised manuscript and supplement.

**22. It would be of added value to the reader of the manuscript and to the manuscript itself to add a brief discussion and description on the CALIPSO aerosol classification algorithm (Kim et al. 2018) and to possible errors in the classification (Burton et al. 2013), since the aerosol subtyping is a cornerstone to this study.**

**Answer**: We made the following change in the revised manuscript (see pg.6-7 line 209-225 in the revised manuscript).

The text "The Version 4.10 CALIOP level 2 vertical feature mask (VFM) product provides the horizontal and vertical distributions of aerosol layers as well as aerosol types (Kim et al., 2018). The CALIOP sub-type detection scheme uses the input parameters - altitude, location, surface type, corrected depolarization ratio, and integrated attenuated backscatter measurements - to identify the aerosol types. Compared with Version 3.0, the Version 4.10 (V4) CALIOP level 2 contains substantial updates to aerosol subtyping algorithms and the following aerosol types are defined: clean marine (sea salt), clean continental (clean background), polluted continental/smoke (urban/industrial pollution), elevated smoke (biomass burning aerosol), dust (desert), polluted dust (dust mixed with anthropogenic aerosol such as

biomass burning smoke or urban pollution) and dusty marine (Kim et al., 2018). A limitation of identifying smoke layers according to altitude is that pollution lofted by convective processes or other vertical transport mechanisms can be misclassified as elevated smoke (Kim et al., 2018). This limitation needs to be kept in mind for the interpretation of the observations and where appropriate, will be mentioned. It is further noted that the CALIOP typing is done on integrated layers that are detected by a separate algorithm, which is not designed to detect differences in aerosol type. Smaller thresholds on depolarization and an attenuation-related depolarization bias can also affect the type classification. What's more, layer heights of contiguous aerosol layers of different types do not accurately reflect the boundaries between different aerosol types (Burton et al., 2013)." has been added in the revised manuscript (see pg.6-7 line 209-225).

**23. Lines 324-325: Include AOD limits of the different "moderately polluted", "polluted" and "heavily polluted" conditions. A histogram of the AOD values delineating the different categories would be nice also.**

**Answer**: We made the following change in the revised manuscript (see pg. 17 line 445-447).

The text "A histogram of CALIOP AOD values showing the different categories and corresponding number of cases for each region are reported in Fig. S2 and Table S10." has been added into 3.3.1 Section in the revised manuscript (see pg. 17 line 445-447 in the revised manuscript).

[Figure]

Figure S2. Histogram of number of CALIOP AOD values in each study area assigned to the AOD categories used to discriminate between moderately polluted (left), polluted (middle) and heavily polluted (right) during the period from 2007 to 2020 over the BTH, YRD and PRD. The AOD values along the x-axis are the maximum values for the different cases. The AOD ranges are provided in Table S10 and were selected such that the profiles over each region were divided into three equally sized subsets.

Table S10. AOD categories use to sub-divide the CALIOP observation in equally sized subsets for moderately polluted, polluted and heavily polluted conditions, based on the CALIOP AOD values over the three study regions. For each condition, the mean value and the range (minimum, maximum value) are shown together with the number of overpasses. All data in the period from 2007 to 2020 are included.

| CALIOP | | | | |
|---|---|---|---|---|
| AOD | | BTH | YRD | PRD |
| **Moderately polluted** | mean | 0.06 | 0.06 | 0.04 |
| | min | 0.0003 | 0.0002 | 0.0005 |
| | max | 0.13 | 0.14 | 0.10 |
| | nr of overpasses | 1440 | 1014 | 689 |
| **Polluted** | Mean | 0.24 | 0.26 | 0.19 |
| | Min | 0.13 | 0.14 | 0.10 |
| | Max | 0.38 | 0.40 | 0.32 |
| | nr of overpasses | 1442 | 923 | 669 |
| **Heavily polluted** | Mean | 0.79 | 0.76 | 0.69 |
| | Min | 0.38 | 0.40 | 0.32 |
| | Max | 2.99 | 2.98 | 2.97 |
| | nr of overpasses | 1231 | 813 | 615 |

**24. Figure 4: Please add variability-bars, and maybe "number of cases-used" to the right axes, to provide to the reader a degree of representativeness. Moreover please explain to the manuscript the feature of extinction coefficient increasing close to the surface (0 km).**

**Answer**: Adding variability-bars for the extinction coefficient profiles make the figure complex. Therefore we added Table S10 (see the answer to comment 23) with the "number of cases-used" for calculating extinction coefficient profiles in the three target regions. Due to the corrected methodology of calculating extinction coefficient profiles, the feature of extinction coefficient increasing close to the surface (0 km) has disappeared.

**25. FO is not explained clearly. Is it the "number of a specific aerosol subtype to the total number of aerosol", the "number of a specific aerosol subtype to the total number of aerosol including Clear-Air", or something else? For instance in the FO figures, the aerosols subtypes is between 0 and 1. Have the authors converted the FO to percentage? If not the FO is very unexpectedly/unphysically low.**

**Answer**: In the previous version of the manuscript, we calculate each aerosol type frequency of occurrence by dividing the number of CALIPSO measurements (including both clear air and aerosol) in the whole vertical layer. Through this calculation, the FO is very low. In the revised manuscript, we keep this calculating methodology. At the same time, we also calculate each aerosol type occurrence frequency by dividing the number of CALIPSO measurements (including both clear air and aerosol) within each vertical layer. Through this calculation, the FO is in the range from 0 to 1.

The text "Typically, aerosol type and optical properties vary with altitude. The frequency of occurrence (FO) of the different aerosol types can be calculated through two approaches: one approach is to calculate the frequency of occurrence of each aerosol type by dividing by the number of CALIPSO measurements (including both

clear air and aerosol) in the whole vertical layer; the other approach is to calculate the frequency of occurrence of each aerosol type by dividing the number of CALIPSO measurements (including both clear air and aerosol) within each vertical range. Here, the former definition is designated as All_FO (in %), the latter definition is designated as Layer_FO. It is noted that these profiles show the frequency of occurrence of each aerosol type, normalized to the sum of all aerosol types over the whole profile (All_FO) or each vertical layer (Layer_FO). Hence the FO only indicates a relative number, i.e. ratio of the number of times a certain aerosol type has been assigned by the VFM algorithm to the total number of times that any aerosol type was assigned. The vertical distribution of the All_FO of the different aerosol types during nighttime over the three regions during the spring, summer, autumn and winter, averaged over the years 2007-2020 are presented in Fig. 6. For comparison, similar aerosol type profiles determined using the Layer_FO approach are presented in Fig. S3. Annual mean vertical distributions of the All_FO and Layer_FO of different CALIOP aerosol types are provided in Figs. S4 and S5. The comparison of the aerosol type profiles derived by using the two approaches, shows the noisy character of the profiles resulting from the Layer_FO approach. For some aerosol types the profiles are in good agreement in the lower 2-3 km, for other they are not. At higher altitudes the profiles are often very different, with high FO values from the Layer_FO approach, which makes it hard to compare with values at lower altitude and provides unrealistic vertical distributions. Therefore, in the following we will focus on the vertical distribution of All_FO and, unless specified otherwise, referred to as FO. Profiles determined using the Layer_FO approach are provided in the Supplement, as they provide information on the contributions of different aerosol types as function of height, but are not discussed." has been added in the revised manuscript (see pg.18 line 467-490 in the revised manuscript).

**26. In Figure 5 please add the "Annual figures".**

**Answer**: We add annual figures in Figure S4 and S5 (see supplement file).

The text "Annual mean vertical distributions of the All_FO and Layer_FO of different CALIOP aerosol types are provided in Figs. S4 and S5." has been added in the revised manuscript (see pg.18 line 481-482).

**27. Lines 483-484: The use of Layer A, B, C, is not very convenient for the reader. Please consider the use of alternative ways of formalism.**

**Answer**: We made the following change in the revised manuscript. "height ranges" is used instead of "layers" since the retrieval is originally starting from distinct layers detected by CALIOP (see pg.19 line 495 in the revised manuscript).

**28. What I am missing in the manuscript is a connection between observations and physics. Observations are discussed, but the study does not go deeper. For example the authors could discuss that polluted dust and smoke are hydrophilic aerosols, in presence of high RH the act as effective CCN aerosols, releasing Latent Heat and contributing to instability, while dust aerosols act as effective IN**

**aerosols, having significant effects in higher altitude, … Please consider improving the manuscript in relation to physical interpretations of the outcomes.**

Answer: We have considerably changed the Introduction Section and now describe the aerosol properties and indicate the effects of composition on their physical properties and direct and indirect effects on climate. (Introduction, lines 39-90), followed by a paragraph providing an overview of previous studies relevant to the subject, touching such applications) (Introduction, lines 91-125). And the Sect. 3.7 Discussion (see pg. 35-36) also provides information between observations and physics and a quite long revised manuscript providing all the statistics.

**29. In the end, conclusion section, what is missing is a section of the way the observations can be used and their added value. Some examples for the authors could be effect on human health, transport, deposition, and more, to extend and discuss them.**

Answer: We made the following change in the revised manuscript (see pg.6 line 200-213 in the revised manuscript).

The text "The nature of aerosol effects on Earth's climate depends strongly on the aerosol vertical distribution. When absorbing aerosol is located above bright clouds, warming effects are amplified. The atmospheric lifetime of aerosol in the free troposphere is much longer than the boundary layer. Aerosol in the free troposphere is transported further away from its sources than at lower altitudes, which further affects the geographic pattern of aerosol impacts. The vertical distribution of tropospheric aerosol is especially valuable for evaluation of global aerosol models because it is a signature of the combined effects of aerosol emissions, the strength of vertical lifting and exchange, atmospheric transport patterns, and removal processes (Winker et al., 2013). The results from this study can be used to improve model assessment of the direct and indirect aerosol effects in eastern China (Wang et al., 2011; Wu et al., 2016). In addition, aerosol particles also play an adverse role on air quality and human health and bring about millions of premature deaths in the world (Chen et al., 2020). The integrated mass of dry particles (PM2.5) related to AOD is often used as an indicator for evaluating air quality and human health (van Donkelaar et al., 2016). The aerosol vertical distributions add value in air quality forecasting and human health research due to its relationship with AOD." has been added in the revised manuscript (see pg. 38 line 946-960).

[Figure]

Figure S1. Spatial distributions of annually averaged (a) MODIS AOD, plotted with a spatial resolution of 0.1° x 0.1°, (b) CALIOP AOD, spatial resolution 0.1° x 0.1° and (c) CALIOP AOD, spatial resolution 1o x 1o, during the period from 2007 to 2020 over the BTH (left), YRD (middle), and PRD (right). Note that the MODIS AOD is at 550 nm, whereas the CALIOP AOD is slightly smaller (532 nm). Also, there is a large difference in spatial coverage, with a MODIS swath width of 2330 km providing daily coverage in 1-2 days, as compared to the CALIOP footprint of 70m.

Table S9. Statistical information on the averaged MODIS and CALIOP AOD, plotted with a spatial resolution of 0.1° by 0.1° over the BTH, YRD and PRD, in Figure S1.

|  |  | MODIS | | | CALIOP | | |
| --- | --- | --- | --- | --- | --- | --- | --- |
|  |  | BTH | YRD | PRD | BTH | YRD | PRD |
| **AOD** | **Mean** | 0.56 | 0.55 | 0.41 | 0.38 | 0.40 | 0.33 |
|  | **Std** | 0.52 | 0.38 | 0.28 | 0.36 | 0.34 | 0.30 |
|  | **Min** | 0.00 | 0.00 | 0.00 | 0.0012 | 0.0015 | 0.0016 |
|  | **Max** | 3.00 | 3.00 | 2.98 | 2.92 | 2.79 | 2.85 |
|  | **Median** | 0.41 | 0.46 | 0.34 | 0.26 | 0.31 | 0.24 |
| **Nr of overpasses** |  | 9466 | 8202 | 6512 | 1682 | 1162 | 875 |

**3. Comments: Background information on the original satellite retrievals and their limitations is missing. Overinterpretation of results and even circular reasoning are the consequence. The interpretation of the appearance of aerosol types in the vertical column above the three regions requires the understanding of the decision tree for assigning an aerosol type to aerosol layers detected in the CALIOP signals (see Kim et al., 2018, Fig. 1). The aerosol subtype selection depends, e.g., on the surface type and the aerosol layer height. In the paper, atmospheric "findings" are discussed that actually originate from the CALIPSO retrieval input and threshold parameters. One should always keep in mind that the CALIPSO aerosol typing is a pre-condition for the L2 algorithms (to select a proper lidar ratio for the extinction retrieval) and is relying only on L1 data and auxiliary information.**

**Answer**: The following text has been added to the revised manuscript (see pg.6).

The text "The Version 4.10 CALIOP level 2 vertical feature mask (VFM) product provides the horizontal and vertical distributions of aerosol layers as well as aerosol types (Kim et al., 2018). The CALIOP sub-type detection scheme uses the input parameters - altitude, location, surface type, corrected depolarization ratio, and integrated attenuated backscatter measurements - to identify the aerosol types. Compared with Version 3.0, the Version 4.10 (V4) CALIOP level 2 contains substantial updates to aerosol subtyping algorithms and the following aerosol types are defined: clean marine (sea salt), clean continental (clean background), polluted continental/smoke (urban/industrial pollution), elevated smoke (biomass burning aerosol), dust (desert), polluted dust (dust mixed with anthropogenic aerosol such as biomass burning smoke or urban pollution) and dusty marine (Kim et al., 2018). A limitation of identifying smoke layers according to altitude is that pollution lofted by convective processes or other vertical transport mechanisms can be misclassified as elevated smoke (Kim et al., 2018). This limitation needs to be kept in mind for the interpretation of the observations and where appropriate, will be mentioned. It is further noted that the CALIOP typing is done on integrated layers that are detected by a separate algorithm, which is not designed to detect differences in aerosol type. Smaller thresholds on depolarization and an attenuation-related depolarization bias can also affect the type classification. What's more, layer heights of contiguous aerosol layers of different types do not accurately reflect the boundaries between different aerosol types (Burton et al., 2013)." has been added in the revised manuscript (see pg. 6-7 line 209-225).

**4. Comments: There is a contradiction in the discussion of CALIOP daytime vs. night-time data. First, it is stated that the study is restricted to night-time observations because of the lower SNR and corresponding biases at daytime (L165 ff.). Later, day- and night-time results are directly compared, and the differences are solely related to atmospheric processes, without considering any biases in the retrievals. A corresponding error estimation is not provided.**

We made the following change in the revised manuscript (see pg. 7 lines 226-237, pg.28 lines 696-710 and pg. 31-32 lines 766-776).

The text "Daytime signals can be affected by background sunlight and reduce the SNR (signal to noise ratio), resulting in a larger fraction of undetected aerosol layers during daytime than during nighttime, and underestimation of the CALIOP extinction coefficients and AOD which is larger during daytime than during nighttime (Kim et al., 2017). The larger fraction of undetected aerosol layers during daytime may also lead to underestimation of the frequency of occurrence (FO) of daytime aerosol types, especially in the upper level (Huang et al., 2013). An overview of the evaluation of the CALIOP AOD versus other measurements shows a low bias of the CALIOP AOD of the order of about 30%. Kim et al. (2018) shows that the CALIOP V4 AOD is still biased low over ocean but less than V3. Over land the V4 vs V3 improvement was not quantified because of the larger uncertainties in the MODIS AOD data which are used as reference. To avoid day/night differences in the CALIOP data, in this study only nighttime measurements at 532 nm were used to investigate the vertical distribution of aerosol types and extinction coefficients." was added into section 2.2.2 in the revised manuscript (see pg. 7 lines 226-237).

The text "However, as discussed in Sect. 2.2.2, the occurrence of undetected aerosol layers results in underestimation of the CALIOP extinction coefficients and AOD and the fraction of undetected aerosol layers is larger during daytime than during nighttime. Furthermore, aerosol in the boundary layer is relatively well detected as compared with aerosol near the top of the boundary layer (Kim et al., 2017) which may lead to distortion of the vertical profile. Hence day/night differences may occur between the vertical distributions of the aerosol type FO due to the CALIOP processing which affect the interpretation of day/night differences in the profiles.

Day/night differences in the vertical structure of aerosol properties are expected due to natural processes such as direct production or formation of secondary aerosol, transformation of aerosol particles in the atmospheric boundary layer, vertical mixing and transport from remote locations, wet and dry deposition. Below the observed day/night differences are briefly discussed based on consideration of such processes. Separation of these effects from those due to the CALIOP processing, and determination of their relative importance, are beyond the scope of the current study. This would require a study on the effects of biases in the retrievals, detection thresholds, noise and the influence of quality control flags." was added into section 3.5 in the revised manuscript (see pg.28 lines 696-710).

The text "Figure S11 shows the difference between the frequency of occurrence of aerosol layer top and base during nighttime and daytime observations (night minus day), for each season averaged over the years 2007 - 2020 over the three study regions. The higher frequency of occurrence of both aerosol layer top and base at higher altitudes (> 2 km) during the night may be caused by two effects. As discussed above, background sunlight reduces the SNR in the lidar signal which increases the fraction of undetected aerosol layers with respect to the fraction of undetected layers at night (Liu et al., 2009; Kim et al., 2017). On the other hand, the CALIPSO overpasses are in the early afternoon and after midnight but the diurnal occurrence of deep convection and precipitation reach their maximum in the late afternoon or early evening (Huang et al., 2013). Hence, compared with daytime, during the night a

higher frequency of occurrence of aerosol types at high altitudes is detected by CALIOP due to the deep convective activity." was also added into section 3.5 in the revised manuscript (see pg. 31-32 lines 766-776).

**5. Comments: It is unclear why the investigation is restricted to the period 2007-2015. Several more years of data are available until today.**

**Answer**: Thank you for this comment. We have followed your suggestion and included the whole period between 2007 and 2020. This required repeating the analysis for this whole data set, but the results were not significantly different. The inclusion of the full 2007-2020 data set is mentioned in the relevant sentences throughout the revised manuscript and all figures were changed/modified accordingly.

**Specific comments**

**Answer** to specific comments 1-3 and 5: Thank you for these comments. Indeed we have been quite generic and did not describe in detail the various aerosol effects, but chose to add references where these are described. However, to highlight the importance of the aerosol types and their vertical distributions, which are the subject of this paper, we have described the various processes in more detail, while keeping the references to their effects on climate. As a result, we have re-organized the first page of the Introduction and this way addressed your comments (see pg. 1-2 lines 39-88). The new text reads:

An aerosol is technically defined as a suspension of fine solid or liquid particles in a gas. In the atmosphere, the air is the gas and in atmospheric research the term aerosol commonly refers to the particulate component only (Seinfeld and Pandis, 1998). In this paper, aerosol is used as a generic term for the particulate component, whereas processes are described for aerosol particles, and a group of aerosol particles with specific properties is indicated by that property (e.g. "dust aerosol"). Aerosol particles are characterized by their diameter, chemical composition and shape (both are size-dependent) and the number of aerosol particles of each size is described by the particle size distribution. Each of these aerosol properties varies with time and space (Unger et al., 2008; Shindell et al., 2009). The chemical composition of an aerosol particle determines its hygroscopicity and thus the ability to take up or release water vapor in response to changes in relative humidity (RH). In supersaturated conditions, hygroscopic particles may be activated and become cloud condensation nuclei (CCN). At low temperatures aerosol particles can act as ice nuclei (Kanji et al., 2017). The chemical composition of the aerosol particles together with the amount of aerosol water determines the optical properties through the complex refractive index which is important for the scattering and absorption of solar radiation in the atmosphere. The effects of these processes on climate (see below) are determined by the amount and size of the aerosol particles and thus the particle size distribution. For instance, in the presence of large CCN concentrations, the amount of water vapour available is distributed over many cloud droplets which results in smaller sizes and larger cloud albedo (Twomey, 1974) and less precipitation (Rosenfeld et al., 2008). In the presence of high concentrations of aerosol particles, more solar radiation is scattered and

absorbed than in the presence of low concentrations, resulting in larger extinction and less radiation reaching the surface (Quan et al., 2014; Li et al., 2017; 2018).

Due to such processes, aerosol particles have an important effect on the Earth's climate, directly by the scattering and absorption of solar radiation and indirectly by modifying cloud properties such as the size and lifetime of cloud droplets, which in turn affect cloud albedo and precipitation (Albrecht, 1989; Twomey, 1974; Andreae et al., 2004; Rosenfeld et al., 2008). Aerosol indirect effects on climate are still poorly understood, much research is done on aerosol-cloud-precipitation interaction (Rosenfeld et al., 2014; Seinfeld et al., 2016; Zhou et al., 2016; Liu et al., 2017; Saponaro et al., 2017; Guo et al., 2018). As indicated above, aerosol direct and indirect effects are strongly influenced by aerosol composition (IPCC, 2013; Rosenfeld et al., 2008; Yang et al., 2016; Massie et al., 2016). In addition, the aerosol vertical distribution is an important factor (Heese et al., 2017; Zhao et al., 2018; Pan et al., 2019) which depends on local sources and vertical mixing together with long-range transport of aerosol generated elsewhere. Also, the aerosol altitude relative to cloud layers needs to be considered (Costantino and Breon, 2013; Wang et al., 2015; Liu et al., 2017; de Graaf et al., 2019). Such information can only be obtained by airborne measurements, or by using remote sensing which provides the data for the current study. However, the data obtained from the optical instruments used for remote sensing, either ground-based or aboard satellites, does not provide sufficient information to fully constrain the aerosol properties. In particular, aerosol composition is poorly constrained and therefore at best aerosol types are retrieved based on the limited number of degrees of freedom. In this study we used aerosol types derived from observations using the Cloud-Aerosol Lidar with Orthogonal Polarization (CALIOP) aboard the Cloud-Aerosol Lidar and Infrared Pathfinder Satellite Observations (CALIPSO) (Kim et al., 2018), see Sect. 2.2.2 for detail.

Because of the strong spatial variability of aerosol properties and their vertical variation, which are hard to determine from local measurements and sparsely distributed networks, satellites are often used to study effects of aerosol on climate. Satellite-based instruments provide the aerosol optical depth (AOD, the column-integrated aerosol extinction coefficient) at the available wavelengths, as the primary parameter. AOD is often used as a proxy for the aerosol loading and to assess the aerosol effect on radiation, clouds and precipitation (Luo et al., 2014; Tian et al., 2017; Zhao et al., 2018; Liu et al., 2017; 2018). This brief summary of aerosol properties important for climate and air quality studies shows that a systematic analysis of the temporal and spatial variations of aerosol concentrations, aerosol types and their vertical distribution is needed to better understand aerosol effects.

**1. Comments: L38, "Aerosol effects depend on particle size distribution": Very generic. Which effects concretely?**
**Answer:** See above specific comment: "Answer to specific comments 1-3 and 5".

**2. Comments: L40, "... important role in Earth's climate change": Very generic. How exactly do aerosol particles change the climate (and not just influence the**

**energy balance)?**

**Answer:** See above specific comment: "Answer to specific comments 1-3 and 5".

**3. Comments: L41, "…serve as cloud condensation nuclei": Why only condensation nuclei? What about ice nuclei?**

**Answer**: See above specific comment: "Answer to specific comments 1-3 and 5". We do not specifically address ice nuclei.

**4. Comments: L35 vs. L44 etc.: "Aerosol" is defined in singular (L35), but then often used in plural. What exactly do you mean with "aerosols" (aerosol particles or aerosol types or something else)?**

**Answer**: Thank you for this comment. Indeed the different terms were not consequently used and may lead to confusion. We have changed the first sentences of the Introduction to "An aerosol is technically defined as a suspension of fine solid or liquid particles in a gas. In the atmosphere, the air is the gas and in atmospheric research the term aerosol commonly refers to the particulate component only (Seinfeld and Pandis, 1998). In this paper, aerosol is used as a generic term for the particulate component, whereas processes are described for aerosol particles, and a group of aerosol particles with specific properties is indicated by that property (e.g. "dust aerosol")." (see pg. 1-2 lines 39-43).

**5. Comments: L48, "…aerosol indirect effects are strongly influenced by aerosol types": What about the direct effects? How do direct and indirect effects depend on the aerosol type?**

**Answer**: See above specific comment: "Answer to specific comments 1-3 and 5".

**6. Comments: L168, "This is further illustrated…": What is illustrated and where?**

**Answer**: This is illustrated in Sect. 3.5 and Fig. 11 in the revised manuscript. We have added a reference to Sect. 3.5 at the end of Sect. 2.2.2 (see pg.7 line 235-238in the revised manuscript).

The text "This is further illustrated with the comparison of vertical distributions of the FO of the aerosol types during day- and night-time overpasses. To avoid such problems, nighttime measurements at 532 nm were used in this study to investigate the vertical distribution of aerosol types and extinction coefficients. The vertical distribution of the frequencies of occurrence of CALIOP-derived aerosol types during nighttime are compared with those derived during daytime." has been changed to "To avoid day/night differences in the CALIOP data, in this study only nighttime measurements at 532 nm were used to investigate the vertical distribution of aerosol types and extinction coefficients. The vertical distribution of the frequencies of occurrence of CALIOP-derived aerosol types during nighttime are compared with those derived during daytime in Sect. 3.5.". (see pg.7 line 235-238 in the revised manuscript).

**7. Comments: L181 ff.: The description of data processing is too brief. What does "cloud-free pixels" exactly mean? To which instrument and which cloud-detection scheme does it refer? Does it only hold for MODIS? What is about CALIOP? How is aerosol-cloud discrimination considered (e.g., are aerosol layers in cloudy profiles included or are only fully cloud-free profiles considered)? How is the AOD from CALIOP calculated? What do the quality control flags mean, i.e., which lidar data are discarded? Are the AOD retrievals from imager and lidar equivalent (e.g., in terms of coverage) and how do the values compare?**

**Answer**: We made the following change in the revised manuscript. Cloud-free pixels means MODIS Level 2 products to be cloud-free. "Aerosol properties are only retrieved for strictly cloud-free pixels" has been changed to "Aerosol properties are only retrieved for strictly cloud-free pixels in the MODIS Level 2 products" (see pg. 8 lines 256). All of the CALIOP data include both cases: the aerosol layers in cloudy profiles and in only fully cloud-free profiles. "The CALIOP version 4 level 2 aerosol products from January 2007 to December 2020 are employed in this study. All CALIOP data include both cases: the aerosol layers in cloudy profiles and in fully cloud-free profiles." has been added into section 2.3 in the revised manuscript (see pg. 8 lines 262-264). The CALIOP AOD with quality control flags are also shown in the manuscript (see pg. 8 line 264-268). The comparison between AOD retrievals from imager and lidar are reported in the answer to General comment 2 (see Reply to comments RC2).

**8. Comments: L200 ff.: Does the description refer to MODIS or CALIOP AOD?**

**Answer**: The description refers to MODIS and we made this change in the revised manuscript (see pg. 8 line 282 and 283, pg. 9 line 297 and pg. 11 line 331).

**9. Comments: L227 ff.: "…larger boundary layer heights … allow for mixing over a deeper layer resulting in elevated AOD": Why? Usually, mixing leads to a dilution of aerosol in the BL while AOD remains constant.**

**Answer**: We have removed this sentence.

**10. Comments: L231 ff.: Trajectory analysis and source attribution are needed instead of speculations about the sources of high AOD. Aerosol removal processes like washout and deposition must be considered in the discussion as well.**

**Answer**: Thank you for this comment. The 48-h backward air mass trajectories over the three target regions are now described in the revised manuscript.

The text "Backward trajectories of the air masses arriving at the center of the three study areas BTH (38°N, 117°E), YRD (30.5°N, 119.5°E) and PRD (23°N, 113.5°E) were determined using HYSPLIT (https://www.ready.noaa.gov/HYSPLIT_traj.php). The air mass trajectories were determined for the arrival points at heights of 500 m, 1000m and 3000m, i.e. the centers of the height ranges with high frequency of occurrence of the different aerosol types as determined from CALIOP data (Sect. 3.3).

The air mass back trajectories were determined over 48 hours, at steps of 6 hours." has been added in Sect. 2.2.4 with caption "2.2.4 Air mass trajectories" (see Sect. 2.2.4, lines 248-254 in the revised manuscript).

The text "In the above, the spatial distributions of the AOD and the vertical distributions of the aerosol types over the BTH, the YRD and the PRD were explained in terms of aerosol origin, local versus remote production and long-range transport, while also noting the CALIOP aerosol type classification method (Kim et al., 2018). To further illustrate the effect of transport and the differences between the three study regions, 48-h backward air mass trajectories for each region and arriving at 500 m, 1000 m and 3000 m were computed as described in Sec. 2.2.4, for every day in the period 2007-2020. These air mass trajectories were clustered by season and the results for each study region are presented in Figures S12-S14. The trajectories clearly show the differences between the three regions, and for each region between seasons, and also the arrival height. Air masses arriving in the BTH regions show the long-range transport from northerly and north-westerly directions, i.e. explaining the dust transport from the deserts such as Gobi and Taklamakan during all seasons except summer. During summer the origin of the 48-h air mass trajectories is relatively close to the BTH, especially due to the reduced transport form northwesterly directions. During summer, transport from south-southwesterly and south-easterly directions contributes more than during other seasons. During all seasons there is a rather strong local contribution, weakening with distance to the BTH over a distance of the order of 1000 km. Part of the air masses originate over the Bohai sea, explaining a marine component in the aerosol types. The distribution of the air mass trajectories arriving at 1000 m is similar to that of the air masses arriving at 500 m, although some air masses go back over a somewhat longer distance. This also applies to air masses arriving at 3000 m, but the distances are substantially longer and there is a wider distribution over the directions, in particular during the autumn and winter when there are more contributions from westerly and southwesterly directions.

The 48-h back trajectories arriving at the YRD at 500 m are much shorter than for the BTH, with stronger contributions from easterly and northerly directions, including those over the East China Sea and the Yellow Sea. Air masses originating from northwesterly directions contribute during all seasons except summer. During summer the 48-h air mass trajectories originate from the east and south of China and over the oceans. In all seasons, only a very small fraction of the air masses originates from the west of China with Xinjiang (Taklamakan desert) and thus the observed dust aerosol type originates from the Gobi Desert in the north of China. Transport to the YRD from the north is an important factor, together with locally generated aerosol and regional transport from eastern and southern China and the East China Sea. This effect is much stronger for air masses arriving at 3000 m, with much longer trajectories and, like over the BTH, a wider distribution of directions. In particular, the contribution of components from westerly directions (SW-NW) is substantially larger.

For 48-h air mass trajectories arriving at the PRD at 500 m the distribution is quite different from those arriving at the BTH and YRD. The 48-h trajectories of the air

masses arriving at the PRD are mostly shorter than over the other study areas and in the spring and summer from southerly directions over the South China Sea. In the autumn and winter the trajectories from the south are much shorter and there is a larger contribution from northerly and easterly directions (Central and East China). Also, in these seasons, a small fraction of the trajectories originates from northerly directions suggesting a possible contribution of dust aerosol originating from the Gobi Desert. In the winter season, a substantial fraction of the trajectories originates from South-East Asia, over longer distances than in other seasons. The distribution of the 48-h air mass trajectories arriving over the PRD at 1000 m is similar to that over air masses arriving at 500 m, also as regards the lengths of the trajectories. The distribution of the air mass trajectories arriving at 3000 m is similar to those arriving at 500 m and 1000 m during the summer, but during other seasons more trajectories originate from north-westerly directions, with distinct difference between spring, autumn and winter and go back further than at lower altitudes. The longer trajectories may result in different aerosol types, transported form other regions, than at lower levels, as also observed in the CALIOP data." has been added in in Sect. 3.6 with caption "3.6 Air mass trajectories and origin of aerosol over the three study regions" (see Sect. 3.6 lines 783-830 in the revised manuscript).

Aerosol deposition and removal have each been mentioned several a few times throughout the manuscript, but not explicitly described because the current study does not provide info on these processes.
L150-152: "The monsoon influences aerosol transport and wet deposition (Liu et al., 2011; Luo et al., 2014), while in turn aerosol particles affect the distribution of precipitation and monsoon intensity (Li et al., 2016)."
L340-341: "The monsoon is accompanied by heavy rain resulting in the effective removal of aerosol particles by wet deposition and thus"
L653-654: "indicating that descending motion of air masses is conducive of the deposition and accumulation of aerosols in the lower atmospheric layers"
L703-706: "natural processes such as direct production or formation of secondary aerosol, transformation of aerosol particles in the atmospheric boundary layer, vertical mixing and transport from remote locations, wet and dry deposition."
L845: "which may be due to wet removal by precipitation"
L952-953: "combined effects of aerosol emissions, the strength of vertical lifting and exchange, atmospheric transport patterns, and removal processes"

**11. Comments: Fig. 1: What is the data source, MODIS or CALIOP? More statistical information is required (e.g., box plots). At least the standard deviation must be provided and explained for all panels.**
**Answer**: The data source for Fig.2 in the revised manuscript is MODIS/Aqua. And the statistical investigation regarding the range of values, standard deviation, median, min and max value has been added in the revised version: "The annual mean AOD averaged over the whole study period is smallest in the PRD with a value of 0.41±0.09 (annual mean ± standard deviation); over the BTH and the YRD the annual

mean AODs averaged over the study period have similar values of 0.56±0.07 and 0.55±0.09, respectively (See Supplement Table S1).", "Detailed statistics of the seasonally averaged MODIS AOD are provided in Table S2 of the Supplement." and "The monthly mean data for each year and in each region, statistics and the number of overpasses included in the monthly averaged MODIS AOD are provided in Tables S3-S8 of the Supplement." in the revised manuscript (see pg.9 line 292-296, line 303-304, and pg.12 line 345-347 in the revised manuscript).

**12. Comments: Fig. 2: What is the data source? What do white pixels mean? How can mean values close to 1.5 be explained, if all AOD values larger than 1.5 are discarded?**

**Answer**: The data source for Fig.3 in the revised manuscript is MODIS/Aqua. The spatial variation of the MODIS AOD in 0.1° x 0.1° over the three urban clusters, averaged over the seasons in the years from 2007 to 2020. The white pixels in Fig. 3 in the revised manuscript indicate that data are missing. In the previous version of manuscript (Fig. 2), MODIS AOD greater than 1.5 were discarded after averaging MODIS AOD in 0.1° x 0.1°. So there are mean values close to 1.5. We agree that this leads to a bias. Hence in the revised manuscript we decided to use upper AOD limits similar to those of CALIPSO, cases with original MODIS AOD greater than 3.0 were discarded at first before the following analysis (including temporal and spatial variation of MODIS AOD). (See also our answer to general comment 1) We reanalyzed all data and update the figures. This issue is shown throughout the revised manuscript (all the figures were changed/modified in this respect).

We have re-organized the text "The spatial distributions of the seasonal mean MODIS AOD over the three urban clusters, averaged over the years from 2007 to 2020, and plotted with a resolution of 0.1°x 0.1°, are shown in Fig. 3. In Fig. 3, some small areas occur where no data are available; these areas are left white. As mentioned in Sect. 2.3, MODIS data with AOD>3.0 were discarded. It is noted that aerosol retrieval over areas with very high AOD may not be successful due to problems with discrimination between high AOD and clouds. The AOD>3.0 threshold avoids confusing cloudy pixels as high AOD cases." (see pg. 12 line 369-374).

**13. Comments: L285 ff.: It is stated that the relative contribution of each aerosol type to the aerosol burden is calculated. However, for such an investigation it would be necessary to weight the aerosol type occurrence with the layer mean extinction. Using only the occurrence does not say anything about the contribution to the aerosol burden.**

**Answer**: Thank you for this comment. Indeed we have formulated this incorrectly and replaced "the relative contribution of each aerosol type to the aerosol burden" with "The relative frequency of occurrence (rFO) of each aerosol type in the atmospheric column" (see pg. 15 line 399-400).

**14. Comments: Sec. 3.2.2 and 3.3: See major comment no. 3. It is important to understand the CALIPSO typing scheme for the interpretation of the findings.**

**Answer**: Thank you again for this comment, we have studied again the relevant literature and added relevant descriptions incl. references. We refer to our response to General comment no.3, as well as to responses to specific comments to Sects. 3.2.2 and 3.3.

**15. Comments: L317, "The vertical profiles of the aerosol extinction coefficients describe the variation of the attenuation of the laser light…": Vice versa. The aerosol extinction coefficient profile is the atmospheric property that causes variations in the attenuation of laser light, which can thus be used to describe the extinction.**

**Answer**: We have changed the text to: "The aerosol extinction coefficient, i.e. the sum of the scattering and absorption by aerosol particles, varies with altitude above the surface due to changes in aerosol properties (see below). Extinction profiles, derived from the vertical variation of the lidar signal (at a wavelength of 532 nm) provide a measure for the vertical variation of aerosol concentrations, weighed by the optical properties of the aerosol particles." (see pg. 17 line 433-437).

**16. Comments: L321, "…modulated by the boundary layer": What does it mean?**

**Answer**: We have replaced this text with "The vertical distribution of the aerosol properties depends on meteorological conditions such as vertical mixing, boundary layer height and the relative humidity profile, as well as the origin of the aerosol (local or long-range transported at elevated levels)." (see pg. 17 line 437-439).

**17. Comments: L323, "…soothes the features and results in rather smooth profiles": What does it mean?**

**Answer**: We have replaced the text with "In this study, profiles clustered for certain conditions are averaged over the whole 14-year study period resulting in the loss of detail (such as varying boundary layer heights) and rather smooth profiles." (see pg. 17 line 439-441).

**18. Comments: L323 ff.: Which AOD is used and what are the concrete intervals? Please provide numbers!**

**Answer**: CALIOP AOD is used in this section, and the concrete intervals and overpass numbers of aerosol samples (after data screening) detected by CALIPSO are both reported in Figure S2 and Table S10 (see supplement file).

The text "A histogram of CALIOP AOD values showing the different categories and corresponding number of cases for each region are reported in Fig. S2 and Table S10." has been added into 3.3.1 Section in the revised manuscript (see pg. 17 line 445-447 in the revised manuscript).

[Figure]

Figure S2. Histogram of number of CALIOP AOD values in each study area assigned to the AOD categories used to discriminate between moderately polluted (left), polluted (middle) and heavily polluted (right) during the period from 2007 to 2020 over the BTH, YRD and PRD. The AOD values along the x-axis are the maximum values for the different cases. The AOD ranges are provided in Table S10 and were selected such that the profiles over each region were divided into three equally sized subsets.

Table S10. AOD categories use to sub-divide the CALIOP observation in equally sized subsets for moderately polluted, polluted and heavily polluted conditions, based on the CALIOP AOD values over the three study regions. For each condition, the mean value and the range (minimum, maximum value) are shown together with the number of overpasses. All data in the period from 2007 to 2020 are included.

| CALIOP | | | | |
|---|---|---|---|---|
| **AOD** | | **BTH** | **YRD** | **PRD** |
| **Moderately polluted** | mean | 0.06 | 0.06 | 0.04 |
| | min | 0.0003 | 0.0002 | 0.0005 |
| | max | 0.13 | 0.14 | 0.10 |
| | nr of overpasses | 1440 | 1014 | 689 |
| **Polluted** | Mean | 0.24 | 0.26 | 0.19 |
| | Min | 0.13 | 0.14 | 0.10 |
| | Max | 0.38 | 0.40 | 0.32 |
| | nr of overpasses | 1442 | 923 | 669 |
| **Heavily polluted** | Mean | 0.79 | 0.76 | 0.69 |
| | Min | 0.38 | 0.40 | 0.32 |
| | Max | 2.99 | 2.98 | 2.97 |
| | nr of overpasses | 1231 | 813 | 615 |

**19. Comments: L333 ff.: This discussion is strange (input = output).**
**Answer**: This sentence has been deleted.

**20. Comments: Fig. 4: Figure caption is wrong.**
**Answer**: The caption of Fig. 4 (now Fig. 5) has been changed to "Aerosol extinction coefficient profiles, averaged over the years 2007-2020, over the BTH (left), YRD (middle) and PRD (right), grouped in different CALIOP AOD ranges for moderately

polluted, polluted and heavily polluted conditions (see caption)."

**21. Comments: Fig. 4: Please provide the variance and the AOD values for each profile.**

Answer: See our response to comment 18.

**22. Comments: L355: The definition of "layers" in this context is a bit misleading, since the retrieval is originally starting from distinct layers detected by CALIOP. It would be better to speak of "height ranges" here. These height ranges should also be indicated in the figures, in order to guide the reader in the discussion. It should be discussed if and how the CALIPSO typing scheme artificially introduces the boundaries of these height ranges.**

Answer: We have followed your suggestion and have made the following change in the revised manuscript (see pg.16 line 420-421). "height ranges" is used instead of "layers" according to comments. These height ranges have also been indicated in the Figure 6 (see pg. 19 lines 504-509 and pg. 22 Figure 6). The definition for elevated smoke with tops higher than 2.5 km above ground level, and for dusty marine with tops lower than 2.5 km above ground level (i.e., a simple approximation of a region above the PBL) in CALIOP V4 (Kim et al., 2018) artificially introduces the boundaries of these height ranges, which are shown in the figures throughout the revised manuscript in this respect.

The text "It is noted that the definitions used in the CALIOP classification approach (Kim et al., 2018) for elevated smoke, with tops higher than 2.5 km above ground level, and for dusty marine, with tops lower than 2.5 km above ground level (i.e., a simple approximation of a region above the PBL) in CALIOP V4 (Kim et al., 2018) artificially introduces the boundaries of these height ranges." has been added into the revised manuscript (see pg. 19 line 500-504).

**23. Comments: L381 ff.: Please explain, under consideration of the CALIPSO typing scheme, where marine and dusty marine profiles come from. Also explain the occurrence of smoke in view of the typing scheme.**

Answer: We made the following change in the revised manuscript (see pg.18 line 499-503 and pg.18 line 492-497).

The text "A new dusty marine aerosol type is introduced in CALIOP V4, to identify mixtures of dust and marine aerosol. As the BTH is located to the west of the Bohai Sea, marine aerosol occurs most frequently around the coast, and dusty marine aerosol also occurs most frequently when dust settles into the marine boundary layer (MBL) as it approaches the BTH area." and "During the summer, the direct emission of aerosol particles and precursor gases (contributing to secondary formation of aerosol particles) from straw burning contributes to the high AOD over the BTH. The larger boundary layer height (BLH) in the summer allows for mixing over a deeper layer, which may promote the larger vertical extent of elevated smoke (see however Sect. 2.2.2)." and "A limitation of identifying smoke layers according to altitude is that pollution lofted by convective processes or other vertical transport mechanisms can be

misclassified as elevated smoke (Kim et al., 2018). This limitation needs to be kept in mind for the interpretation of the observations and where appropriate, will be mentioned." have been added in the revised manuscript (see pg. 20 line 537-541 and pg.18 line 531-535 and pg. 6 line 218-222).

**24. Comments: Fig. 5: Figure caption is wrong.**
**Answer**: We have changed the caption of Fig. 5 (now Fig. 6) to "Figure 6. Vertical distribution of the nighttime FO (i.e. All_FO as explained in the text) of different CALIOP aerosol types (see legend) by season, averaged over the years 2007-2020, over the BTH (left), YRD (middle) and PRD (right). The designation of different layers is illustrated for the PRD, autumn, see text." (see pg. 22).

**25. Comments: L437, "dusty marine aerosol": From Fig. 6, it seems that it is clean marine aerosol. Again, please consider the surface-dependent typing in the interpretation.**
**Answer**: Thank you for catching this mistake, we have changed in the revised manuscript (see pg.23 line 598-599).

**26. Comments: Fig. 6: Figure caption is wrong.**
**Answer**: We have changed the Fig. 6 (now Fig. 7) caption to "Figure 7. Vertical distribution of the nighttime FOs of different CALIOP aerosol types (see legend), averaged over the years 2007-2020, grouped in different CALIOP AOD ranges for moderately polluted (top), polluted (middle) and heavily polluted (bottom) conditions (see caption), over the BTH (left), YRD (middle) and PRD (right)." (see pg. 25).

**27. Comments: Sec. 3.4.1: What does the RH at 950 hPa have to do with the vertical distribution of aerosols up to 8 km height? This discussion is very misleading. RH analysis can only be done when the RH for each detected layer is considered. A trajectory analysis would be much more appropriate to support the discussion of air mass origin.**
**Answer**: Sect. 3.4.1 has been removed. The 48-h backward air mass trajectories over the three target regions are now described in the revised manuscript (see pg. 33 lines 783-830 in the revised manuscript and our response to the previous comment nr 10).

**28. Comments: Fig. 7: While the RH discussion is already strange, the provision of 2 decimal places is even more useless.**
**Answer**: Sect. 3.4.1 has been removed.

**29. Comments: Sec. 3.4.2: Horizontal transport (according to trajectory analysis) and limitations of the typing scheme need to be considered in the discussion.**
**Answer**: The 48-h backward air mass trajectories over the three target regions are now described in the revised manuscript (see pg.33 lines 783-830 in the revised manuscript and our response to the previous comment nr 10).
The limitations of the typing scheme have been considered throughout the revised

manuscript.

For example, the text "The occurrence of elevated smoke in layer C is consistent with the definition used in the CALIOP classification approach for elevated smoke, i.e. the layer with tops higher than 2.5 km above ground level (Kim et al., 2018)." has been added in the Sec.3.4.2 (see pg. 27 lines 681-683 in the revised manuscript).

The text "It is noted that the definitions used in the CALIOP classification approach (Kim et al., 2018) for elevated smoke, with tops higher than 2.5 km above ground level, and for dusty marine, with tops lower than 2.5 km above ground level (i.e., a simple approximation of a region above the PBL) in CALIOP V4 (Kim et al., 2018) artificially introduces the boundaries of these height ranges." has been added in the Sec.3.2.2 (see pg. 19 lines 500-504 in the revised manuscript).

The text "following the CALIOP classification" has been added in the Sec.3.7 (see pg. 35 lines 852 in the revised manuscript).

The text "Daily air mass back trajectories are provided for the whole study period and discussed to evaluate the different source regions for the three study areas, at three altitude ranges in which different aerosol types are assigned in the CALIOP approach." has been added in the Introduction (see pg. 4 lines 137-139 in the revised manuscript).

The text "The observed distributions of aerosol types are thus to a certain extend biased by the CALIOP classification approach, yet differences are observed. Below we also discuss long range transport, air mass trajectories showing transport pathways are presented in Sect. 3.6" has been added in the Sec.3.3.2 (see pg. 19 lines 521-523 in the revised manuscript).

**30. Comments: L533 ff.: Again, consider how smoke is assigned to a layer in the typing scheme and do not overinterpret the results. This discussion is mainly based on circular reasoning.**

**Answer**: We have discussed these issues in Sect. 2.2.2, where we added at pg.6 lines L218-221: "A limitation of identifying smoke layers according to altitude is that pollution lofted by convective processes or other vertical transport mechanisms can be misclassified as elevated smoke (Kim et al., 2018). This limitation needs to be kept in mind for the interpretation of the observations and where appropriate, will be mentioned." And we indeed have referred to this text a few times.

**31. Comments: Sec. 3.5: As mentioned in major comment no. 4, it is unclear how the detection limitations at daytime influence the obtained differences. Detection thresholds, noise, and the influence of the quality control flags must be investigated, before conclusions can be drawn from the findings.**

**Answer**: Please see our response to General comment 4.

**32. Comments: Fig. 10: Figure caption is wrong.**

**Answer**: The caption of Fig. 10 has been changed in the revised manuscript to "Figure 10. Vertical distribution of the FO of different CALIOP aerosol types (see legend) during daytime, for each season over the BTH (left), YRD (middle) and PRD (right), averaged over the years 2007-2020." (see pg. 30).

**33. Comments: L575 ff.: Discussion on the influence of the diurnal variation of emissions is missing.**

**Answer**: We have added the following text "Furthermore, anthropogenic activities result in stronger emissions from, e.g., domestic activities and traffic, of particulate matter, aerosol precursor gases and secondary formation by photochemical processes, during the day than during the night. This results in a diurnal cycle with substantial day/night differences of AOD and particulate matter (Lennartson et al., 2018), with higher aerosol concentrations during daytime." (see pg. 30 L731-735)

And: "Polluted continental aerosol is emitted, or formed from precursor gases, near the surface and its transport to higher elevations is prohibited by the temperature inversion at the top of the mixed layer. The formation from precursor gases often involves a photo-chemical reaction, i.e. requires the availability of solar radiation and thus occurs during daytime." (see pg. 31 L742-746)

**34. Comments: Fig. 11: Figure caption is wrong.**

**Answer**: The caption of Fig. 11 has been changed in the revised manuscript to "Figure 11. Differences between nighttime and daytime vertical distributions of the FO of CALIOP-derived aerosol types (see legend) (nighttime minus daytime) by season, averaged over the years 2007-2020, over the BTH (left), YRD (middle) and PRD (right)." (see pg. 33).

**35. Comments: Conclusions: Reconsider your conclusions in view of all the comments above.**

**Answer**: We have reconsidered our conclusions and modified where we thought it was needed, and added more text in response to your comments and those from the other reviewer. However, all detail is provided in Section 3 (and Sect. 2 where the typing approach is discussed based on literature) and the results are summarized in the Conclusion Section which therefore has been changed to Summary and Conclusions

**36. Comments: L654, "both flying on the Aqua satellite": Really?!**

**Answer**: We changed this in the revised manuscript to "the A-Train satellite constellation" (see pg.36 line 876-877).

**37. Comments: Acknowledgements, "We also thank the reviewers of this paper for their valuable comments which helped improve the manuscript.": Well, I do hope that this statement will become true.**

**Answer**: We have done a lot for the manuscript improvement, following your suggestions as well as those from the other reviewer, and we are really do hope that this statement will become true. Thank you for your valuable comments!

**Technical comments**
**1. Comments: L146, "…we use of the MODIS": delete "of"**
**Answer**: We made this change in the revised manuscript (see pg.6 line 193).

**2. Comments: L187: typo in index 532nm,unc**

**Answer**: We made this change in the revised manuscript (see pg.8 line 266).

**3. Comments: L265: North China Plain (not Plan)**

**Answer**: We made this change in the revised manuscript (see pg.12 line 377).

**4. Comments: L349: certain (not certainly)**

**Answer**: We made this change in the revised manuscript (see pg.18 line 476).

[revised manuscript text omitted]

Page 3, line 90-94: Text was changed to 'Ground-based remote sensing includes the

use of sun photometers which are part of networks such as AERONET (Zhang and Li, 2019), SONET (Zhang and Li, 2015; Li et al., 2019; Zhang et al., 2020), CARSNET (Che et al., 2015), hand-held sun photometers in the CARE-China network (Xin et al., 2015) and solar radiation measurements (Xu et al., 2015).'

Page 3, line 97-98: Text was added as: 'An analysis of global aerosol type as retrieved by MISR was presented by Kahn and Gaitley (2015).'

Page 3, line 102-104: Text was changed to 'Since the launch of CALIOP/CALIPSO in 2006 (Winker et al., 2009), the seasonal variations of aerosol types and the aerosol vertical distribution could be examined over large spatial scales, complementary to the local point measurements using ground-based lidars.'

Page 4, line 137-139: Text was added as: 'Daily air mass back trajectories are provided for the whole study period and discussed to evaluate the different source regions for the three study areas, at three altitude ranges in which different aerosol types are assigned in the CALIOP approach.'

Page 4, line 155-156: Text was added as: 'i.e. the Beijing-Tianjin-Hebei (BTH) area, the Yangtze River Delta (YRD) and the Pearl River Delta (PRD) (see Fig. 1).'

Page 5, line 167-169: Text was added as: 'Figure 1. Elevation map of Eastern China showing the study areas, i.e. Beijing-Tianjin-Hebei (BTH, 35.5°N-40.5°N and 113.5°E-120.5°E), the Yangtze River Delta (YRD, 28°N-33°N and 117°E-122°E) and the Pearl River Delta (PRD, 21.5°N-24.5°N and 111.5°E-115.5°E). These areas are indicated by the black rectangles. The elevation data is downloaded from the website https://search.earthdata.nasa.gov/ (last access: 20 May 2021).'

Page 6, line 201-202: Text was changed to 'CALIOP is a space-borne near-nadir dual-wavelength lidar (532 nm and 1064 nm) that provides high-resolution vertical profiles of aerosols and clouds.'

Page 6, line 209-237: Text was changed to 'The Version 4.10 CALIOP level 2 vertical feature mask (VFM) product provides the horizontal and vertical distributions of aerosol layers as well as aerosol types (Kim et al., 2018). The CALIOP sub-type detection scheme uses the input parameters - altitude, location, surface type, corrected depolarization ratio, and integrated attenuated backscatter measurements - to identify the aerosol types. Compared with Version 3.0, the Version 4.10 (V4) CALIOP level 2 contains substantial updates to aerosol subtyping algorithms and the following aerosol types are defined: clean marine (sea salt), clean continental (clean background), polluted continental/smoke (urban/industrial pollution), elevated smoke (biomass burning aerosol), dust (desert), polluted dust (dust mixed with anthropogenic aerosol such as biomass burning smoke or urban pollution) and dusty marine (Kim et al., 2018). A limitation of identifying smoke layers according to altitude is that pollution

lofted by convective processes or other vertical transport mechanisms can be misclassified as elevated smoke (Kim et al., 2018). This limitation needs to be kept in mind for the interpretation of the observations and where appropriate, will be mentioned. It is further noted that the CALIOP typing is done on integrated layers that are detected by a separate algorithm, which is not designed to detect differences in aerosol type. Smaller thresholds on depolarization and an attenuation-related depolarization bias can also affect the type classification. What's more, layer heights of contiguous aerosol layers of different types do not accurately reflect the boundaries between different aerosol types (Burton et al., 2013). Daytime signals can be affected by background sunlight and reduce the SNR (signal to noise ratio), resulting in a larger fraction of undetected aerosol layers during daytime than during nighttime, and underestimation of the CALIOP extinction coefficients and AOD which is larger during daytime than during nighttime (Kim et al., 2017). The larger fraction of undetected aerosol layers during daytime may also lead to underestimation of the frequency of occurrence (FO) of daytime aerosol types, especially in the upper level (Huang et al., 2013). An overview of the evaluation of the CALIOP AOD versus other measurements shows a low bias of the CALIOP AOD of the order of about 30%. Kim et al. (2018) shows that the CALIOP V4 AOD is still biased low over ocean but less than V3. Over land the V4 vs V3 improvement was not quantified because of the larger uncertainties in the MODIS AOD data which are used as reference. To avoid day/night differences in the CALIOP data, in this study only nighttime measurements at 532 nm were used to investigate the vertical distribution of aerosol types and extinction coefficients.'

Page 7, line 241-246: Text was changed to 'Meteorological parameters are available for the whole world, with different spatial resolutions, every six hours, from the daily ERA Interim Reanalysis (http://apps.ecmwf.int/datasets/data/interim-full-daily/; last access: 20 May 2021). Daily temperatures at the 1000 hPa and 700 hPa levels and pressure vertical velocity (PVV) at the 750 hPa level on 0.125°×0.125° grids are used with the closest collocation with the CALIOP (nighttime) overpass time (18:00 UTC) over the study area.'

Page 7, line 248-254: Text was added as: 'Backward trajectories of the air masses arriving at the center of the three study areas BTH (38°N, 117°E), YRD (30.5°N, 119.5°E) and PRD (23°N, 113.5°E) were determined using HYSPLIT (https://www.ready.noaa.gov/HYSPLIT_traj.php; last access: 20 May 2021) and GDAS meteorological data (ftp://arlftp.arlhq.noaa.gov/pub/archives/gdas1/; last access: 20 May 2021). The air mass trajectories were determined for the arrival points at heights of 500 m, 1000 m and 3000 m, i.e. the centers of the height ranges with high frequency of occurrence of the different aerosol types as determined from CALIOP data (Sect. 3.3). The air mass back trajectories were determined over 48 hours, at steps of 6 hours.'

Page 8, line 256-258: Text was changed to 'Aerosol properties are only retrieved for

strictly cloud-free pixels in the MODIS Level 2 products, as determined using a cloud-detection scheme. Through a sensitive cloud detection scheme, the MODIS aerosol algorithm could minimize cloud contamination (Martins et al., 2002).'

Page 8, line 260-264: Text was changed to 'To avoid such problems and use upper AOD limits similar to those of CALIPSO, cases with MODIS AOD greater than 3.0 were discarded in the analysis.
The CALIOP version 4 level 2 aerosol products from January 2007 to December 2020 are employed in this study. All CALIOP data include both cases: the aerosol layers in cloudy profiles and in fully cloud-free profiles.'

Page 8, line 268-273: Text was added as: 'The aerosol extinction vertical profiles used in this study were selected following similar quality control procedures: (1) $0 <= AOD_{532nm} <= 3.0$; (2) $-100 <= CAD\_Score <= -20$; (3) $Ext\_QC = 0, 1$; and (4) $0 < AOD_{532nm,unc} /AOD_{532nm} <= 100\%$, and (5) extinction coefficients with uncertainty of 99.99 km-1 in the profile are rejected.'

Page 8, line 279-280: Text was changed to 'here, lower tropospheric stability (LTS) and pressure vertical velocity (PVV) are considered.'

Page 8, line 288-290: Text was changed to 'However, the AOD peaked in 2011 over the YRD, in 2012 over the BTH and PRD. After these years the AOD decreased until the end of the study period, but slower during the last 3-4 years.'

Page 8-9, line 292-296: Text was added as: 'The annual mean AOD averaged over the whole study period is smallest in the PRD with a value of 0.41±0.09 (annual mean±standard deviation); over the BTH and the YRD the annual mean AODs averaged over the study period have similar values of 0.56±0.07 and 0.55±0.09, respectively (See Supplement Table S1).'

Page 9, line 303-304: Text was added as: 'Detailed statistics of the seasonally averaged MODIS AOD are provided in Table S2 of the Supplement.'

Page 9, line 323-326: Text was added as: 'The effects of long-range transport, such as that of desert dust in the BTH, the biomass burning over the PRD in the spring and the westerly winds in the autumn and the northerly winds in the winter over the three areas are confirmed by air mass back trajectories presented and discussed in Sect. 3.6.'

Page 11, line 329-330: Text was changed to 'Figure 2. Annually (a), seasonally (b) and monthly (c) averaged MODIS AOD over the three study regions. The data for the three regions are color-coded.'

Page 11, line 331-334: Text was changed to 'The monthly mean MODIS AOD over

the three regions, averaged over the 14-years 2007-2020, is presented in Fig. 2(c). Figure 2(c) shows that the largest differences between the regions occur from May to August. The summer AOD peaks in the BTH and in the YRD clearly occur in June (AOD of 0.79 and 0.84, respectively), with a fast decline thereafter.'

Page 11, line 340-341, 'The monsoon brings heavy rains which effectively washout aerosols, resulting in' was changed to 'The monsoon is accompanied by heavy rain resulting in the effective removal of aerosol particles by wet deposition and thus'.

Page 12, line 345-347: Text was added as: 'The monthly mean data for each year and in each region, statistics and the number of overpasses included in the monthly averaged MODIS AOD are provided in Tables S3-S8 of the Supplement.'

Page 12, line 348-351: Text was added as: 'In summary, the data in Fig. 2 show that AOD differences between the PRD and the other two study areas are largest during the summer months, by a factor of about 2. This difference is the main reason for the lower annual mean AOD over the PRD. During some months (March-April) the AOD is similar in all three study areas.'

Page 12, line 353-367: Text was added as: 'Maps showing the spatial variation of the annual mean AOD over the three study areas, derived from MODIS and CALIOP data and averaged over the whole study period (2007-2020), are presented in Fig. S1. Statistical information on these data is summarized in Table S9. Figure S1 shows that the spatial patterns of the MODIS and CALIOP AODs are similar. However, the CALIOP AOD is clearly smaller than that from MODIS, as quantitatively illustrated by the data in Table S9. Underestimation of the AOD by CALIOP has been reported and explained in the literature (cf. Kim et al., 2017, for an overview). It is noted that the comparison in Fig. S1 and Table S9 was made for all available samples and no selection was made based on collocation. Comparison of the maps in Fig. S1 clearly shows the much smaller number of samples in the CALIOP data, due to the much smaller coverage of CALIOP as a result of the smaller swath width and thus substantially smaller number of CALIOP overpasses (Table S9). Hence, the differences between the MODIS and CALIOP AOD are likely augmented by the highly non-uniform data sample and to the fundamentally different algorithms and operation of the sensors. This was also reported by de Leeuw et al. (2018). In view of these differences, the spatial variation and time series analysis is made using MODIS data, whereas the vertical information is provided from CALIOP observations.'

Page 12, line 369-376: Text was changed to 'The spatial distributions of the seasonal mean MODIS AOD over the three urban clusters, averaged over the years from 2007 to 2020, and plotted with a resolution of 0.1°x 0.1°, are shown in Fig. 3. In Fig. 3, some small areas occur where no data are available; these areas are left white. As mentioned in Sect. 2.3, MODIS data with AOD>3.0 were discarded. It is noted that aerosol retrieval over areas with very high AOD may not be successful due to

problems with discrimination between high AOD and clouds. The AOD>3.0 threshold avoids confusing cloudy pixels as high AOD cases. The spatial patterns over each of the three regions are similar in all seasons, but the AOD values vary from season to season.'

Page 15, line 395-397: Text was changed to 'Figure 3. Spatial distributions of seasonally mean MODIS AOD over the BTH (left column), the YRD (middle column), and the PRD (right column), for spring (MAM), summer (JJA), autumn (SON), and winter (DJF) (top to bottom rows), averaged over the study period from 2007 to 2020.'

Page 15, line 398-401: Text was changed to 'The relative frequency of occurrence (rFO) of each aerosol type in the atmospheric column over each of the three study areas'

Page 15-16, line 403-404: Text was changed to 'Over the BTH, polluted dust is the most dominant aerosol type, with an rFO of 45%. The rFO of dust aerosol is 28%.'

Page 16, line 414-418: Text was changed to 'The aerosol composition over the PRD is substantially different from that over the BTH and YRD, with an rFO of elevated smoke aerosol of 30%. The rFOs of polluted dust and polluted continental aerosol are 17% and 26%, respectively. In contrast to the other two regions, clean marine aerosol has a substantial FO (13%) over the PRD and for dust it is only 3%. The rFO of clean continental aerosol is higher (6%) over the PRD than the other two regions.'

Page 16, line 422-425: Text was changed to 'The rFOs of clean marine, polluted continental, clean continental and elevated smoke aerosol are lowest over the BTH and highest over the PRD. In contrast, polluted dust and dust have the largest FOs over the BTH whereas their rFOs over the PRD are small.'

Page 16, line 425-426: Text was added as: 'Transport pathways, i.e. air mass back trajectories, are presented and discussed in Sect. 3.6.'

Page 16, line 429-430: Text was changed to 'Figure 4. Relative frequencies of occurrence of different CALIOP aerosol types over the BTH (left), YRD (middle) and PRD (right), averaged over the time period 2007-2020.'

Page 17, line 433-451: Text was changed to 'The aerosol extinction coefficient, i.e. the sum of the scattering and absorption by aerosol particles, varies with altitude above the surface due to changes in aerosol properties (see below). Extinction profiles, derived from the vertical variation of the lidar signal (at a wavelength of 532 nm) provide a measure for the vertical variation of aerosol concentrations, weighed by the optical properties of the aerosol particles. The vertical distribution of the aerosol properties depends on meteorological conditions such as vertical mixing, boundary

layer height and the relative humidity profile, as well as the origin of the aerosol (local or long-range transported at elevated levels). In this study, profiles clustered for certain conditions are averaged over the whole 14-year study period resulting in the loss of detail (such as varying boundary layer heights) and rather smooth profiles. Figure 5 shows nighttime aerosol extinction coefficient profiles averaged over each of the three regions during moderately polluted, polluted and heavily polluted conditions. This distinction was made based on the CALIOP AOD (obtained by integration of the extinction coefficient profiles over the tropospheric column) which was used to divide the profiles into three equally sized subsets. A histogram of CALIOP AOD values showing the different categories and corresponding number of cases for each region are reported in Fig. S2 and Table S10. The annual mean extinction coefficient profiles were calculated following procedures discussed in Amiridis et al. (2013) and Tackett et al., (2018). The mean of the quality-assured extinction coefficient profiles was first calculated at the overpass level - based on L2 profiles per overpass. Then seasonal and annual profiles were calculated using the mean profiles for all overpasses.'

Page 18, line 463-465: Text was changed to 'Figure 5. Aerosol extinction coefficient profiles, averaged over the years 2007-2020, over the BTH (left), YRD (middle) and PRD (right), grouped in different CALIOP AOD ranges for moderately polluted, polluted and heavily polluted conditions (see caption).'

Page 18, line 468-490: Text was changed to 'Typically, aerosol type and optical properties vary with altitude. The frequency of occurrence (FO) of the different aerosol types can be calculated through two approaches: one approach is to calculate the frequency of occurrence of each aerosol type by dividing by the number of CALIPSO measurements (including both clear air and aerosol) in the whole vertical layer; the other approach is to calculate the frequency of occurrence of each aerosol type by dividing the number of CALIPSO measurements (including both clear air and aerosol) within each vertical range. Here, the former definition is designated as All_FO (in %), the latter definition is designated as Layer_FO. It is noted that these profiles show the frequency of occurrence of each aerosol type, normalized to the sum of all aerosol types over the whole profile (All_FO) or each vertical layer (Layer_FO). Hence the FO only indicates a relative number, i.e. ratio of the number of times a certain aerosol type has been assigned by the VFM algorithm to the total number of times that any aerosol type was assigned. The vertical distribution of the All_FO of the different aerosol types during nighttime over the three regions during the spring, summer, autumn and winter, averaged over the years 2007-2020 are presented in Fig. 6. For comparison, similar aerosol type profiles determined using the Layer_FO approach are presented in Fig. S3. Annual mean vertical distributions of the All_FO and Layer_FO of different CALIOP aerosol types are provided in Figs. S4 and S5. The comparison of the aerosol type profiles derived by using the two approaches, shows the noisy character of the profiles resulting from the Layer_FO approach. For some aerosol types the profiles are in good agreement in the lower 2-3 km, for other they are not. At higher altitudes the profiles are often very different, with high FO

values from the Layer_FO approach, which makes it hard to compare with values at lower altitude and provides unrealistic vertical distributions. Therefore, in the following we will focus on the vertical distribution of All_FO and, unless specified otherwise, referred to as FO. Profiles determined using the Layer_FO approach are provided in the Supplement, as they provide information on the contributions of different aerosol types as function of height, but are not discussed.'

Page 19, line 494-498: Text was added as: 'Therefore we denote the three aerosol height ranges A, B and C. Range A extends from the surface to about 2 km and does not have a distinct maximum. Range B is interspersed with Range A and extends from the surface, where the FO is very small, to about 3 km, with a distinct maximum at about 1 km. Range C extends from about 1.5 km to 4-5 km with a distinct FO peak around 3 km.'

Page 19, line 498-504: Text was added as: 'In spring (MAM), range C may extend to 6 km. In addition, some aerosol types may occur over the whole column without a distinct layering (e.g. dust aerosol over the BTH and the YRD in the spring (MAM)) which is denoted as range D. It is noted that the definitions used in the CALIOP classification approach (Kim et al., 2018) for elevated smoke, with tops higher than 2.5 km above ground level, and for dusty marine, with tops lower than 2.5 km above ground level (i.e., a simple approximation of a region above the PBL) in CALIOP V4 (Kim et al., 2018) artificially introduces the boundaries of these height ranges.'

Page 19, line 518-523: Text was added as :'Note that the explanation of the vertical distribution of the aerosol types focuses on physical processes, whereas, as mentioned above, the CALIOP aerosol type classification results from the consideration of statistics on the occurrence of certain aerosol types (Kim et al., 2018). The observed distributions of aerosol types are thus to a certain extend biased by the CALIOP classification approach, yet differences are observed. Below we also discuss long range transport, air mass trajectories showing transport pathways are presented in Sect. 3.6.'

Page 19-20, line 531-535: Text was added as: 'During the summer, the direct emission of aerosol paticles and precursor gases (contributing to secondary formation of aerosol particles) from straw burning contributes to the high AOD over the BTH. The larger boundary layer height (BLH) in the summer allows for mixing over a deeper layer, which may promote the larger vertical extent of elevated smoke (see however Sect. 2.2.2).'

Page 20, line 537-541: Text was added as: 'A new dusty marine aerosol type is introduced in CALIOP V4, to identify mixtures of dust and marine aerosol. As the BTH is located to the west of the Bohai Sea, marine aerosol occurs most frequently around the coast, and dusty marine aerosol also occurs most frequently when dust settles into the marine boundary layer (MBL) as it approaches the BTH area.'

Page 22, line 578-580: Text was changed to 'Figure 6. Vertical distribution of the nighttime FO (i.e. All_FO as explained in the text) of different CALIOP aerosol types (see legend) by season, averaged over the years 2007-2020, over the BTH (left), YRD (middle) and PRD (right). The designation of different layers is illustrated for the PRD, autumn, see text.'

Page 23, line 598-599: Text was changed to 'Over the PRD, the aerosol in moderately polluted conditions is dominated by clean marine aerosol in range A reflecting the influence of the ocean south of the PRD.'

Page 24, line 624-626: Text was added as: 'The vertical distributions of the frequency of occurrences for different aerosol types, derived using the Layer_FO approach, sub-divided into different CALIOP AOD ranges with equally sized subsets, are presented in Fig. S6.'

Page 25, line 629-632: Text was changed to 'Figure 7. Vertical distribution of the nighttime FOs of different CALIOP aerosol types (see legend), averaged over the years 2007-2020, grouped in different CALIOP AOD ranges for moderately polluted (top), polluted (middle) and heavily polluted (bottom) conditions (see caption), over the BTH (left), YRD (middle) and PRD (right).'

Page 26-27, line 661-664: Text was changed to 'Figure 8. Vertical distributions of the nighttime FOs of different aerosol types (see legend) over the YRD, averaged over the years 2007-2020. The profiles are stratified by pressure vertical velocity (PVV), as a measure for the strength of vertical mixing (see text), at 750 hPa: i.e. for PVV<0 (left) and for PVV>0 (right).'

Page 27, line 676-677: Text was changed to 'The peak FOs of polluted continental and polluted dust aerosol in layer B are larger and occur at somewhat lower altitude when the atmosphere becomes more stable.'

Page 27, line 681-686: Text was added as: 'The occurrence of elevated smoke in layer C is consistent with the definition used in the CALIOP classification approach for elevated smoke, i.e. the layer with tops higher than 2.5 km above ground level (Kim et al., 2018). The data in Fig. 9 show that the FO of elevated smoke aerosol in range C decreases when the atmosphere becomes more stable. In contrast, the peak FOs of polluted dust and dust aerosol above 2 km gradually increases with the increase of LTS.'

Page 28, line 689-691: Text was changed to 'Figure 9. Vertical distributions of the nighttime FOs of different aerosol types (see legend) for data stratified by unstable atmosphere (left), neutral stable atmosphere (middle) and stable atmosphere (right)

over the YRD, averaged over the years 2007-2020.'

Page 28, line 695-710: Text was added as: 'These differences will be discussed based on difference plots (night - day) of FO vertical distributions. However, as discussed in Sect. 2.2.2, the occurrence of undetected aerosol layers results in underestimation of the CALIOP extinction coefficients and AOD and the fraction of undetected aerosol layers is larger during daytime than during nighttime. Furthermore, aerosol in the boundary layer is relatively well detected as compared with aerosol near the top of the boundary layer (Kim et al., 2017) which may lead to distortion of the vertical profile. Hence day/night differences may occur between the vertical distributions of the aerosol type FO due to the CALIOP processing which affect the interpretation of day/night differences in the profiles.
Day/night differences in the vertical structure of aerosol properties are expected due to natural processes such as direct production or formation of secondary aerosol, transformation of aerosol particles in the atmospheric boundary layer, vertical mixing and transport from remote locations, wet and dry deposition. Below the observed day/night differences are briefly discussed based on consideration of such processes. Separation of these effects from those due to the CALIOP processing, and determination of their relative importance, are beyond the scope of the current study. This would require a study on the effects of biases in the retrievals, detection thresholds, noise and the influence of quality control flags.'

Page 30, line 725-726: Text was changed to 'Figure 10. Vertical distribution of the FO of different CALIOP aerosol types (see legend) during daytime, for each season over the BTH (left), YRD (middle) and PRD (right), averaged over the years 2007-2020.'

Page 30-31, line 731-735: Text was added as: 'Furthermore, anthropogenic activities result in stronger emissions from, e.g., domestic activities and traffic, of particulate matter, aerosol precursor gases and secondary formation by photochemical processes, during the day than during the night. This results in a diurnal cycle with substantial day/night differences of AOD and particulate matter (Lennartson et al., 2018), with higher aerosol concentrations during daytime.'

Page 31, line 742-746: Text was added as: 'Polluted continental aerosol is emitted, or formed from precursor gases, near the surface and its transport to higher elevations is prohibited by the temperature inversion at the top of the mixed layer. The formation from precursor gases often involves a photo-chemical reaction, i.e. requires the availability of solar radiation and thus occurs during daytime.'

[revised manuscript text omitted]

Page 35, line 856-858: Text was changed to 'which may be due to the long-range transport of dust aerosol by westerly and northerly winds from dust generated from the Taklamakan and Gobi deserts in the west and north of China (see Sect. 3.6).'

Page 36, line 887-890: Text was changed to 'On a monthly scale, the summer AOD peaks over the BTH (0.79) and the YRD (0.84) occur in June. In contrast, over the PRD the two AOD peaks are observed, one in March (0.68) and a weaker one in October (0.41), whereas, much lower AOD values occur in the summer with a clear minimum in July.'

Page 37, line 902-904: Text was changed to 'Elevated smoke (see comment in Sect. 2.2.2 on CALIOP classification of elevated smoke) is the second dominant aerosol type above 2 km and polluted continental is the second dominant aerosol type below 2 km in the summer.'

Page 37-38, line 933-960: Text was changed to 'Air mass trajectories show the differences in the origin of the aerosol observed over the three study areas. The distributions the 48-h air mass trajectories during the four seasons show substantial differences between the directions from which air masses are transported to the three study areas, and thus the origin of the aerosol. These air mass distributions vary by season, in particular during the summer they are much different from the distributions in other seasons. The air mass trajectory distributions also vary with height, not only as regards the length of the trajectory but also as regards their origin. Hence, the aerosol types may vary with height, as observed, due to different origins of the aerosol observed at different heights. It is noticed that the CALIOP aerosol type classification method influences the observations and introduces uncertainties, but do not lead to contradiction in the interpretation.

In summary, the aerosol properties, aerosol types and vertical profiles in different AOD and meteorological conditions over three representative regions over China were described, using synergetic use of aerosol products from active and passive sensors. Air mass trajectories were used to explain the transport pathways to the three study areas. The nature of aerosol effects on Earth's climate depends strongly on the aerosol vertical distribution. When absorbing aerosol is located above bright clouds, warming effects are amplified. The atmospheric lifetime of aerosol in the free troposphere is much longer than the boundary layer. Aerosol in the free troposphere is transported further away from its sources than at lower altitudes, which further affects the geographic pattern of aerosol impacts. The vertical distribution of tropospheric aerosol is especially valuable for evaluation of global aerosol models because it is a signature of the combined effects of aerosol emissions, the strength of vertical lifting and exchange, atmospheric transport patterns, and removal processes (Winker et al., 2013). The results from this study can be used to improve model assessment of the direct and indirect aerosol effects in eastern China (Wang et al., 2011; Wu et al., 2016). In addition, aerosol particles also play an adverse role on air quality and human health and bring about millions of premature deaths in the world (Chen et al., 2020). The integrated mass of dry particles (PM2.5) related to AOD is often used as an indicator for evaluating air quality and human health (van Donkelaar et al., 2016). The aerosol vertical distributions add value in air quality forecasting and human health research due to its relationship with AOD.'

---

## Author Response (AR2)

**Reply to comments by Reviewer#3 on "Multi-dimensional satellite observations of aerosol properties and aerosol types over three major urban clusters in eastern China"**

July 13, 2021

We thank the reviewer for the thorough reading of the manuscript and the thoughtful comments which are helpful not only for this manuscript but also for our future research. Our replies to all comments are shown below in red.

**Comments**

**1. The words in some subfigures overlaped the plots,e.g., SON and DJF in Fig.10.**

**Answer**: Thank you for this suggestion, it will help readers to read the manuscript easier. So we made this change in all relevant figures in the revised manuscript and supplement.

**2. Verification and correction of CALIPSO-based aerosol products is key to atmospheric environment and climate change. However, CALIPSO often cannot detect the full profile of aerosol for the low instrument sensitivity near the surface. Note that CALIOP may lose detection capability, if attenuated backscatter signal of aerosol is below $2\sim4\times10^{-4}$ $km^{-1}$ $sr^{-1}$ km. Particularly, the aerosol profile near the surface (below 1.5 km) always have higher uncertainties, and may contributed more errors to CALIPSO AOD. In recent decades, China has undergone rapid economic growth with high aerosol concentrations significantly increased over China, especially in PBL, implying that all the aforementioned large uncertainties in PBL should be discussed in section 2.3 or section 3.3.3.**

**Answer**: We made the following change in the revised manuscript (see pg. 8 line 264-269 in the revised manuscript).

The text "CALIOP often cannot detect the full profile of aerosol due to the low instrument sensitivity near the surface, i.e. CALIOP may lose detection capability when the attenuated aerosol backscatter signal is smaller than $2\sim4\times10^{-4}$ $km^{-1}$ $sr^{-1}$ km (Winker et al., 2009; Huang et al., 2013). In particular, the aerosol profile near the surface (below 1.5 km) has high uncertainties which may increase the error in the CALIPSO AOD (Guo et al., 2016a). The uncertainties can be constrained through data screening to some degree." has been added in section 2.3 in the revised manuscript (see pg. 8 line 264-269).

**3. In addition, some words are not uniform in section title ,e.g., setction titles 3.6, 3.7 and 4.**

**Answer**: We made the following changes in the revised manuscript. The title "3.6 Air mass trajectories and origin of aerosol over the three study regions" has been changed to "3.6 Air mass back trajectories and origin of aerosol over the three study regions", the title "3.7 Discussion" has been changed to "4 Discussion", and the title "4 Summary and Conclusions" has been changed to "5 Summary and Conclusions" in the revised manuscript.

References

Guo, J., Liu, H., Wang, F., Huang, J., Xia, F., Lou, M., Wu, Y., Jiang, J. H., Xie, T., Zhaxi, Y., and Yung, Y. L.: Three dimensional structure of aerosol in China: a perspective from multi-satellite observations, Atmos. Res., 178-179, 580-589, 2016a.

Huang, L., Jiang, J. H., Tackett, J. L., Su, H., Fu, R.: Seasonal and diurnal variations of aerosol extinction profile and type distribution from CALIPSO 5-year observations. Journal of Geophysical research: Atmospheres, vol. 118, 4572-4596, doi: 10.1002/jgrd.50407, 2013.

Winker, D. M., Vaughan, M. A., Omar, A., Hu, Y. X., Powell, K. A., Liu, Z. Y., Hunt, W. H., and Young, S. A.: Overview of the CALIPSO Mission and CALIOP Data Processing Algorithms, J. Atmos. Ocean. Tech., 26, 2310-2323, doi:10.1175/2009JTECHA1281.1, 2009.